# Chronic activation of endothelial MAPK disrupts hematopoiesis via NFKB dependent inflammatory stress reversible by SCGF

Pradeep Ramalingam[1], Michael G. Poulos[2], Elisa Lazzari[2], Michael C. Gutkin[2], David Lopez[1], Christopher C. Kloss[1], Michael J. Crowley[1], Lizabeth Katsnelson[1], Ana G. Freire[2], Matthew B. Greenblatt[3], Christopher Y. Park[4] & Jason M. Butler[2,5*]

Inflammatory signals arising from the microenvironment have emerged as critical regulators of hematopoietic stem cell (HSC) function during diverse processes including embryonic development, infectious diseases, and myelosuppressive injuries caused by irradiation and chemotherapy. However, the contributions of cellular subsets within the microenvironment that elicit niche-driven inflammation remain poorly understood. Here, we identify endothelial cells as a crucial component in driving bone marrow (BM) inflammation and HSC dysfunction observed following myelosuppression. We demonstrate that sustained activation of endothelial MAPK causes NF-κB-dependent inflammatory stress response within the BM, leading to significant HSC dysfunction including loss of engraftment ability and a myeloid-biased output. These phenotypes are resolved upon inhibition of endothelial NF-κB signaling. We identify SCGF as a niche-derived factor that suppresses BM inflammation and enhances hematopoietic recovery following myelosuppression. Our findings demonstrate that chronic endothelial inflammation adversely impacts niche activity and HSC function which is reversible upon suppression of inflammation.

[1] Department of Medicine, Division of Regenerative Medicine, Weill Cornell Medical College, New York, NY 10065, USA. [2] Center for Discovery and Innovation, Hackensack University Medical Center, Nutley, NJ 07110, USA. [3] Pathology and Laboratory Medicine, Weill Cornell Medical College, New York, NY 10021, USA. [4] Department of Pathology, New York University Langone Health, School of Medicine, New York, NY 10016, USA. [5] Molecular Oncology Program, Georgetown University, Washington, DC 20057, USA. *email: jason.butler@hmh-cdi.org

An outstanding question in stem cell biology is whether chronic inflammation within tissue-specific microenvironments impairs the ability of supportive niche cells to appropriately nurture their cognate stem cells[1–3]. In the context of hematopoiesis, inflammatory signals play key roles during diverse processes including embryonic specification of the hematopoietic stem cell (HSC) during development, emergency granulopoiesis during infections, and hematopoietic regeneration following transplantation[4–8]. On the other hand, sustained inflammation has been proposed as a key driver of aging-associated hematopoietic defects including loss of HSC self-renewal ability, myeloid-biased differentiation, and a predisposition towards leukemias[9–12]. Growing evidence indicates that the crosstalk between hematopoietic cells and their supportive niche cells initiates and sustains chronic inflammation within the bone marrow (BM), although their precise contributions in this process remain unclear[9–12]. Within the BM, endothelial cells (ECs) have been established as an integral component of the HSC-supportive perivascular niche and express a diverse array of HSC-regulatory factors[13–24]. Modulation of signaling pathways within ECs have been shown to impact HSC self-renewal and lineage-commitment decisions[15,21,22]. Additionally, ECs play a key role during chronic inflammation[24,25] and have emerged as an important source of niche-derived inflammatory signals within the BM, including interleukin-1 (IL-1) and G-CSF[10,26]. Sustained endothelial inflammation has been implicated in the initiation of myeloproliferative diseases through the expression of G-CSF and tumor necrosis factor-α (TNFα)[27]. However, signaling pathways mediating chronic endothelial inflammation within the BM microenvironment that impact niche activity and HSC function remain poorly understood.

Nuclear factor-κB (NF-κB) and MAPK are the principal signaling pathways regulating chronic inflammatory responses within ECs[25]. However, their role in modulating inflammation within the BM endothelial niche and the concomitant impact on HSC function remains unexplored. Prior work from our group has shown that suppression of NF-κB signaling within ECs enhances steady-state hematopoiesis as well as regeneration following myelosuppression, in part by decreasing pro-inflammatory cytokines[21]. Recent reports suggest that endothelial MAPK likely plays an essential role during inflammatory processes including LPS-induced granulopoiesis and chronic vascular inflammation-associated atherosclerosis[28,29]. Utilizing an ex vivo niche model system, we have previously demonstrated that endothelial MAPK activation drives myeloid-biased differentiation of co-cultured HSCs at the expense of their self-renewal, features that are suggestive of an inflammatory stress[15]. Here, we sought to determine whether endothelial MAPK activation impacts BM inflammation, niche activity, and HSC function.

## Results

### Endothelial MAPK activation impairs HSC function and hematopoiesis.

To examine whether endothelial MAPK activation affects hematopoiesis, we generated a mouse model wherein MAPK signaling is constitutively activated within ECs of adult mice. Mice carrying a *Rosa26* Stop/Floxed MEK1DD cassette (an inducible S218D/S222D MAPKK1 mutant that renders ERK-MAPK signaling constitutively active) were crossed to a tamoxifen-inducible *cre* transgenic mouse under the control of the adult EC-specific VE-cadherin promoter (*Cdh5(PAC)-creERT2*) to generate *CDH5-MAPK* mice. To activate MAPK signaling in ECs, 6- to 10-week-old male and female mice were maintained on tamoxifen-impregnated feed (250 mg/kg) for 4 weeks and were allowed to recover for 4 weeks before experimental analysis. *CDH5-MAPK* mice displayed decreased BM cellularity and a decline in the frequency and absolute numbers of immunophenotypically defined HSCs (defined as cKIT+Lineage$^{Neg}$ CD41−SCA1+ CD150+CD48$^{Neg}$), as well as hematopoietic stem and progenitor cells (HSPCs) including KLS cells (cKIT+Lineage$^{Neg}$ SCA1+), multipotent progenitors (MPPs; cKIT+Lineage$^{Neg}$ SCA1+ CD150 $^{Neg}$CD48$^{Neg}$), and hematopoietic progenitor cell subsets (HPC-1 and HPC-2 defined as cKIT+Lineage$^{Neg}$ SCA1+ CD150 $^{Neg}$CD48+ and cKIT+Lineage$^{Neg}$ SCA1+ CD150+CD48+, respectively), as compared to their littermate controls (Fig. 1a–d, Supplementary Fig. 1a, Source Data). The decline in HSPC frequency in *CDH5-MAPK* mice manifested as a functional loss of progenitor activity by methylcellulose-based colony assays (Fig. 1e). Competitive BM transplantation revealed that BM cells from *CDH5-MAPK* mice displayed diminished long-term engraftment and a significant myeloid-biased peripheral blood output (Fig. 1f, g). Limiting dilution transplantation assays confirmed that endothelial MAPK activation significantly reduced the frequency of bona fide long-term HSCs (LT-HSCs) that are able to give rise to stable (>4 months; >1% CD45.2 engraftment), multi-lineage engraftment (Fig. 1h, i). Cell-cycle analysis demonstrated that HSCs and HSPCs from *CDH5-MAPK* mice displayed a loss of quiescence and increased apoptosis as compared to their littermate controls (Fig. 1j, k, Supplementary Fig. 1b–f). Taken together, these data demonstrate that chronic activation of endothelial MAPK adversely impacts steady-state hematopoiesis and HSC function.

### Endothelial MAPK drives an NF-kB dependent inflammatory stress response.

The hematopoietic defects observed in *CDH5-MAPK* mice suggest that constitutive MAPK activation likely affects the integrity of the BM endothelial niche. Immunofluorescence analysis of the BM confirmed that MAPK activation led to disruption of the endothelial network, including an increase in vascular dilatation (Fig. 2a). Analysis of vascular integrity by Evan's Blue assay revealed that *CDH5-MAPK* mice develop a significant increase in BM vascular leakiness, indicative of a loss of vascular integrity (Fig. 2b–d). Notably, vascular dilation and enhanced leakiness are hallmarks of an inflammatory stress[30]. Plasma proteome analysis of *CDH5-MAPK* mice demonstrated significantly increased levels of inflammatory mediators, including sICAM, VCAM, and IL1b (Fig. 2e, Supplementary Table 1, Supplementary Data 1). Ingenuity Pathway Analysis of the differentially expressed proteins revealed that "Inflammatory Response" was the most significantly enriched disease process in *CDH5-MAPK* mice (P value $1.3 \times 10^{-13}$, Fisher's exact test, and activation z-score 2.32) (Fig. 2f), indicating that endothelial MAPK activation leads to an inflammatory stress response. It is well established that NF-κB signaling plays pivotal roles in driving inflammatory responses, and recent reports indicate that endothelial MAPK activation can lead to inflammation via downstream activation of canonical NF-κB signaling[28,29,31,32]. To verify this possibility, we performed immunoblot analysis on BM-derived ECs (BMECs) of *CDH5-MAPK* mice which confirmed an increase in MEK1DD driven ERK1/2 phosphorylation (Fig. 2g, h) and revealed a modest but consistent increase in p65 phosphorylation with no significant changes in total IκBα levels. These features are indicative of sustained activation of NF-κB signaling wherein endogenous feedback mechanisms increase the synthesis of total IκBα levels[33–35]. Quantification of nuclear p65 levels by immunofluorescence analysis demonstrated an increase in nuclear p65 within BMECs of *CDH5-MAPK* mice, confirming activation of NF-κB signaling downstream of endothelial MAPK activation[36] (Fig. 2i, j). Collectively, these findings suggested that increased NF-κB signaling within ECs of *CDH5-MAPK* mice drives an inflammatory stress response leading to vascular defects.

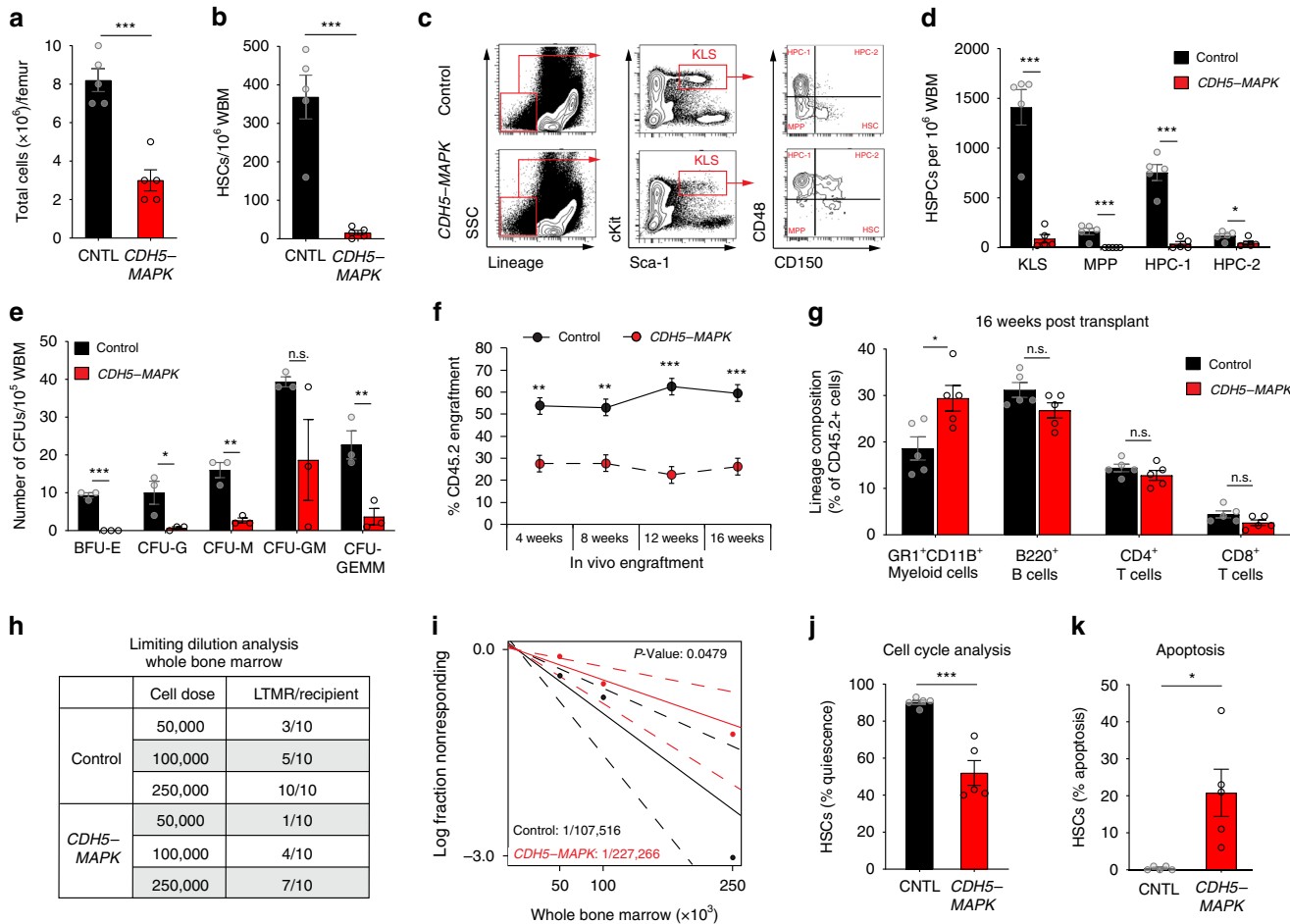

**Fig. 1** *CDH5-MAPK* mice manifest HSC and hematopoietic defects. **a** Total cells per femur ($n = 5$ mice/cohort). **b** Frequency of phenotypic HSCs per $10^6$ femur cells assessed by flow cytometry ($n = 5$ mice/cohort). **c** Representative contour plots demonstrating gating strategy for quantification of BM HSC and HSPC frequency by flow cytometry. **d** Frequency of phenotypic HSPCs per $10^6$ femur cells assessed by flow cytometry ($n = 5$ mice/cohort). **e** Methylcellulose-based progenitor assay. Bar graphs indicate number of CFUs per $10^5$ WBM ($n = 3$ mice/cohort). **f, g** Competitive repopulation assays assessing total CD45.2$^+$ cell engraftment and CD45.2$^+$ lineage distribution over 4 months ($n = 10$ recipients/cohort; $n = 5$ donors/cohort). For evaluating competitive repopulation, $5 \times 10^5$ donor WBM cells (CD45.2) were transplanted with $5 \times 10^5$ competitor WBM cells (CD45.1) into pre-conditioned CD45.1 recipient mice. **h** Table representing the number of recipients that were positive for long term, multi-lineage reconstitution (LTMR) following whole-bone-marrow-limiting dilution transplantation assay ($n = 10$ recipients/cohort per cell dose; $n = 5$ donors/cohort). **i** Line graph displaying estimates of HSC frequency in the indicated genotypes with dashed lines representing 95% confidence intervals. Stem cell frequency and significance were determined using Extreme Limiting Dilution Analysis (ELDA). **j** Cell-cycle analysis of HSCs in the indicated genotypes by flow cytometry ($n = 5$ mice/cohort). Gating strategy described in Supplementary Fig. 1b. **k** Quantification of HSC apoptosis in the indicated genotypes by flow cytometry ($n = 5$ mice/cohort). Gating strategy described in Supplementary Fig. 1b. Error bars represent sample mean ± SEM. Statistical significance was determined using two-tailed unpaired Student's *t*-test. *$P \leq 0.05$; **$P < 0.01$; ***$P < 0.001$; $^{n.s.}P > 0.05$.

**Endothelial NF-κB inhibition resolves vascular defects in *CDH5-MAPK* mice.** We next determined whether suppression of NF-κB signaling within ECs of *CDH5-MAPK* mice is sufficient to restore their vascular defects. To this end, we transduced BMECs derived from control and *CDH5-MAPK* mice with a lentivirus expressing a dominant negative IκBα$^{S32A/S36A}$ super-suppressor (IkB-SS) construct that sequesters NF-kB (p65/p50) in the cytoplasm[37,38]. Immunoblot analysis confirmed the expression of IkB-SS transgene and revealed no significant alterations in ERK1/2 or p65 phosphorylation levels due to transgene expression (Fig. 3a, b). Immunofluorescence analysis confirmed that expression of IkB-SS decreased p65 nuclear translocation in BMECs of *CDH5-MAPK* mice (Fig. 3c, d). These findings suggested that increased NF-κB signaling within ECs of *CDH5-MAPK* mice in vivo could be suppressed by EC-specific expression of IkB-SS. To test this hypothesis, we crossed *CDH5-MAPK* mice with *Tie2.IkB-SS* mice (*CDH5-MAPK::IkB* mice) wherein

the IkB-SS transgene is selectively expressed within ECs of adult mice[39]. Analysis of NF-κB regulated target gene expression within in vivo BMECs (defined as CD45-Ter119-CD31+VEcadherin+) of *CDH5-MAPK* mice revealed increased levels of NF-kB signaling targets, including the pro-inflammatory cytokines and chemokines *Il1a*, *Il1b*, *Cxcl1*, *Cxcl3*, *Ccl12*, and *Ccl22* (Fig. 3e–f, Supplementary Table 2). Importantly, BMECs derived from *CDH5-MAPK::IkB* mice demonstrated an overall decrease in expression of NF-kB target genes indicating that IkB-SS transgene expression within BMECs of *CDH5-MAPK* mice decreased their NF-κB signaling (Fig. 3e–f, Supplementary Table 2). Furthermore, immunofluorescence analysis of the BM endothelial niche demonstrated a normalization of the vascular dilation in *CDH5-MAPK::IkB* mice, confirming that inhibition of endothelial NF-κB signaling is sufficient to restore BM vascular integrity in *CDH5-MAPK* mice (Fig. 3g). Collectively, these findings confirmed that the detrimental effects of endothelial MAPK activation on the BM

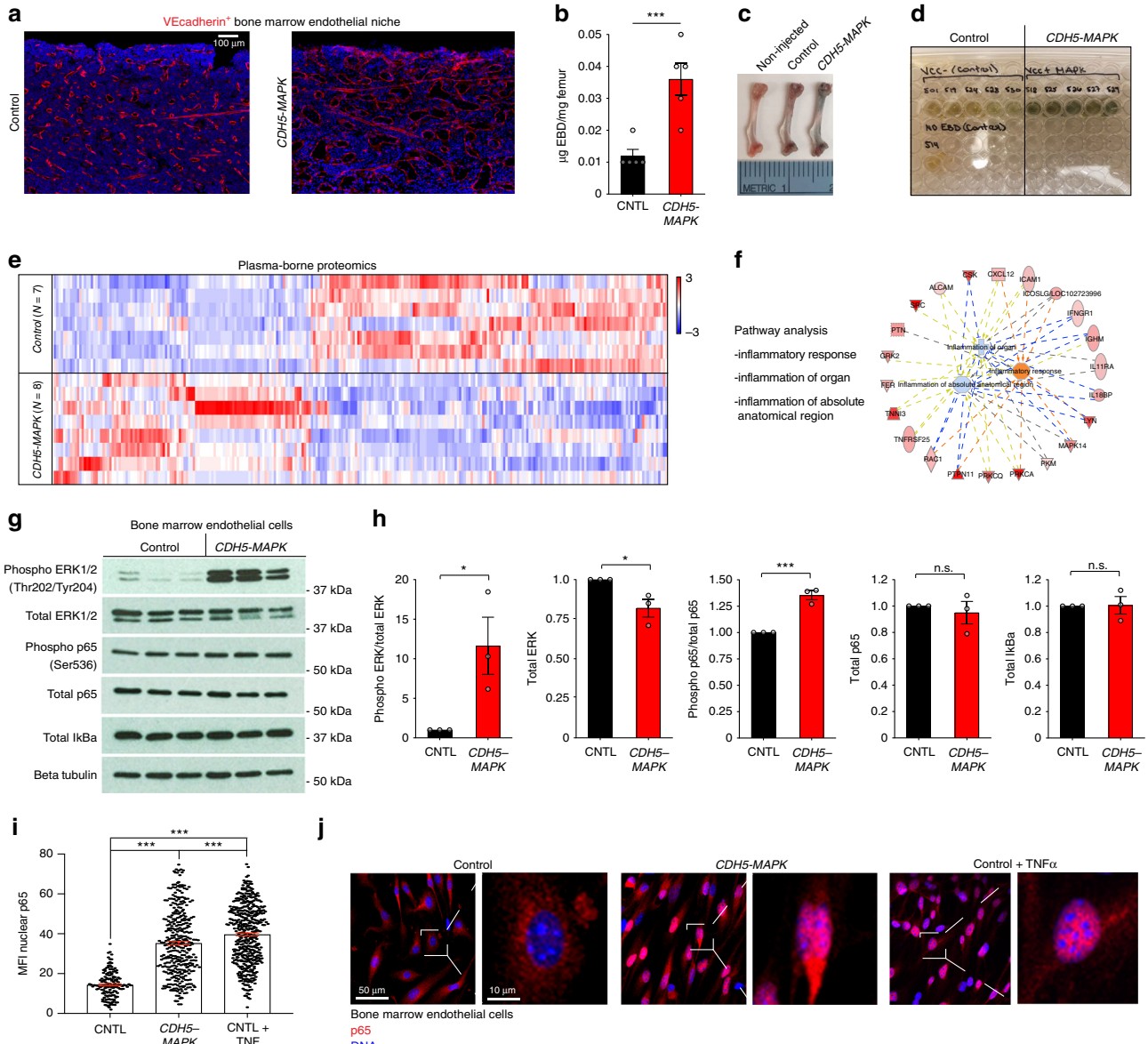

**Fig. 2 CDH5-MAPK mice display systemic and BM-localized inflammation. a** Representative immunofluorescence images of femurs intravitally labeled with a vascular-specific CD144/VE-cadherin antibody (red) demonstrating vascular dilatation in *CDH5-MAPK* mice. **b** Quantification of Evan's Blue Dye (EBD) extravasation (*n* = 5 mice/cohort). **c** Representative images of femurs isolated from mice injected with EBD. **d** Microtiter plate demonstrating intensity of extracted EBD for each sample. Non-injected controls were used to determine baselines. **e** Heatmap of 242 differentially expressed proteins in plasma of *CDH5-MAPK* mice identified by proteomic analysis (*n* = 7 control and *n* = 8 *CDH5-MAPK* mice). Color scales represent relative protein abundance reflecting mean fluorescence intensities of SomaLogic aptamer-based ELISA. Raw data included in Supplementary Data 1 and Source Data. **f** Ingenuity Pathway Analysis of differentially expressed proteins demonstrating that inflammatory responses are over-represented in *CDH5-MAPK* mice. **g**, **h** Immunoblot analysis of BMECs isolated from *CDH5-MAPK* mice demonstrating that MEK1DD expression in BMECs results in an increase in ERK1/2 and p65 phosphorylation (*n* = 3 biological replicates per genotype). **i**, **j** Representative immunofluorescence images and quantification demonstrating increased levels of nuclear p65 in BMECs derived from *CDH5-MAPK* mice as compared to controls. Control BMECs treated with TNFα (10 ng/mL for 15 min) were used as a positive control for the assay. Each dot within the bar graph represents nuclear p65 staining intensity per individual cell. Error bars represent sample mean ± SEM. Statistical significance was determined using two-tailed unpaired Student's *t*-test. *$P \leq 0.05$; ***$P < 0.001$; n.s.$P > 0.05$.

vascular niche are mediated primarily by downstream activation of endothelial NF-κB-dependent inflammatory stress.

**Endothelial NF-κB inhibition restores HSC activity in CDH5-MAPK mice.** We next determined whether restoration of BM endothelial niche integrity in *CDH5-MAPK::IkB* mice resulted in a functional recovery of HSCs and the hematopoietic system. Analysis of *CDH5-MAPK::IkB* mice demonstrated a restoration of BM cellularity and frequency of phenotypic HSCs and HSPCs (Fig. 4a,

b, Supplementary Fig. 2a). The phenotypic recovery of progenitors within BM of *CDH5-MAPK::IkB* mice was also reflected in their progenitor activity in colony assays (Fig. 4c). Additionally, HSC functionality assayed by competitive BM transplantations demonstrated a complete recovery of long-term engraftment potential and a reversal of myeloid-biased differentiation in *CDH5-MAPK::IkB* mice (Fig. 4d, e, Supplementary Fig. 2b). WBM cells derived from *CDH5-MAPK::IkB* mice were also able to maintain their serial repopulation and multi-lineage reconstitution abilities during

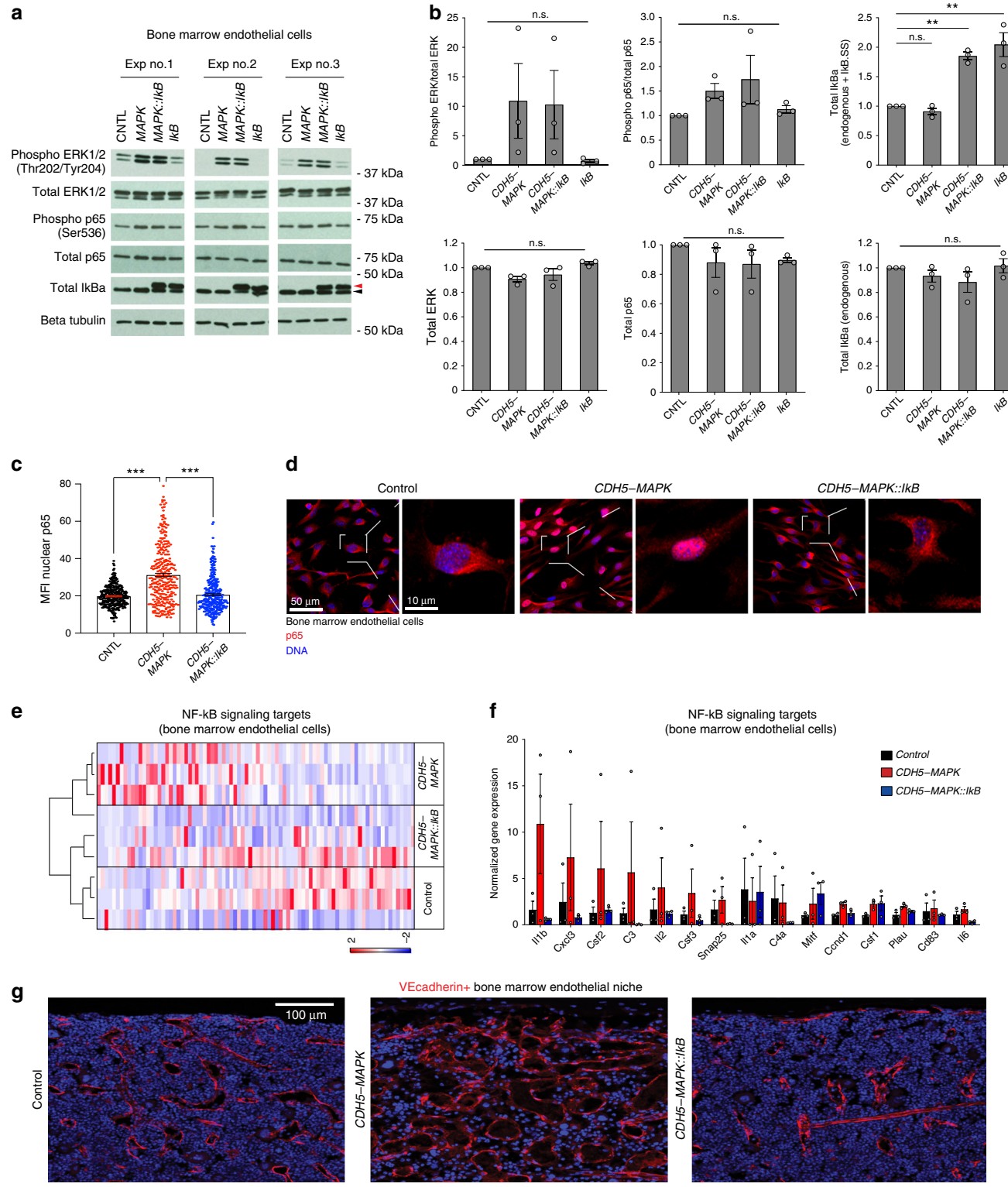

secondary transplantation assays (Supplementary Fig. 2c, d). Moreover, limiting dilution BM transplantation confirmed the restoration in frequency of bona fide LT-HSCs that were able to give rise to long-term multi-lineage engraftment in *CDH5-MAPK::IkB* mice (Fig. 4f, g). Under homeostatic conditions, HSCs predominantly reside in a perivascular niche wherein they receive instructive cues from ECs and perivascular niche cells that regulate their quiescence and self-renewal. To determine whether inflammation within BMECs of *CDH5-MAPK* disrupted HSC–niche

interactions, we performed whole-mount immunofluorescence imaging of HSCs which revealed an increase in average distance of HSCs from the vasculature (Fig. 4h, i). Notably, *CDH5-MAPK::IkB* mice demonstrated a restoration of HSC proximity to the vascular niche (Fig. 4h, i). These data demonstrate that rescue of BM endothelial integrity in *CDH5-MAPK::IkB* mice resulted in a normalization of HSC–niche interactions and a restoration of HSC function. To determine whether inflammation associated with endothelial MAPK activation influences post-myelosuppressive

**Fig. 3 Endothelial NF-κB inhibition resolves endothelial inflammation and restores vascular integrity in *CDH5-MAPK* mice. a, b** Immunoblot analysis and quantification demonstrating that expression of IkB-SS in BMECs isolated from *CDH5-MAPK* mice does not affect ERK1/2 or p65 phosphorylation. Black arrowhead represents endogenous IκBα whereas red arrowhead represents IkB-SS transgene. **c, d** Representative immunofluorescence images and quantification demonstrating increased levels of nuclear p65 in BMECs derived from *CDH5-MAPK* mice as compared to controls. Note that expression of IkB-SS in BMECs isolated from *CDH5-MAPK* mice (*CDH5-MAPK::IkB*) decreases nuclear p65 levels. Each dot within the bar graph represents nuclear p65 staining intensity per individual cell. **e, f** Heatmap and bar graph demonstrating increased expression of NF-κB-dependent inflammatory genes within BMECs of *CDH5-MAPK* mice. Crossing *CDH5-MAPK* with *Tie2.IkB-SS* mice (*CDH5-MAPK::IkB*) resulted in decreased expression of NF-κB-dependent target genes. Expression of Actb was used for normalization. Dendrograms represent unsupervised hierarchical clustering of the entire dataset ($n = 3$ mice/cohort). Color scales represent relative mRNA abundance based on normalized gene expression. Raw data for Heatmaps included in Supplementary Table 2 and Source Data. **g** Representative immunofluorescence images of femurs intravitally labeled with a vascular-specific CD144/VE-cadherin antibody (red) demonstrating that suppression of NF-κB signaling within endothelial cells of *CDH5-MAPK* mice resolves their vascular dilatation. Error bars represent sample mean ± SEM. One-way ANOVA for multiple comparisons and Tukey's correction was performed to determine significance. $^{**}P < 0.01$; $^{***}P < 0.001$; $^{n.s.}P > 0.05$.

hematopoietic reconstitution, we subjected *CDH5-MAPK, Tie2.IkB-SS*, and *CDH5-MAPK::IkB* mice and littermate controls to a sublethal myelosuppressive (650 Rad) injury. Peripheral blood analysis revealed that *CDH5-MAPK* mice displayed a significant delay in hematopoietic recovery, indicating that sustained endothelial MAPK activation is deleterious for recovery following myelosuppression (Fig. 4j). However, EC-specific NF-κB inhibition in *CDH5-MAPK* mice protected white blood cell, neutrophil, red blood cell, and platelet loss following irradiation (Fig. 4j). Notably, the rate of hematopoietic recovery in *CDH5-MAPK::IkB* mice is similar to the recovery observed in *Tie2.IkB-SS* mice, which have been previously demonstrated to display a robust protection of the hematopoietic system following myelosuppressive injury[21]. These data suggest that endothelial inflammation in *CDH5-MAPK* mice delays hematopoietic recovery whereas EC-specific NF-κB inhibition in *CDH5-MAPK* mice protects their hematopoietic compartment and enhances recovery following myelosuppressive injury.

**Endothelial inflammation impairs hematopoietic progenitor activity.** Along with HSCs, the vascular niche within the BM plays a crucial role in maintaining a diverse array of lineage-committed hematopoietic progenitors that sustain steady-state peripheral blood output[40–42]. Moreover, the vascular niche within the spleen has been shown to be a vital component for extramedullary hematopoiesis[43]. We next sought to determine the effects of endothelial MAPK activation on hematopoietic progenitors within the BM and spleen. *CDH5-MAPK* mice displayed a decline in BM multipotent progenitors (MPPs), common lymphoid progenitors (CLP), common myeloid progenitors (CMPs), granulocyte/macrophage progenitors (GMPs), megakaryocyte/erythroid progenitors (MEPs), and B cell progenitor subsets (sIgM-B220+ B cells, Pre Pro B cells, Pro B cells, and Pre B cells) which was functionally reflected in their decreased peripheral blood counts (Fig. 5a, b, d, Supplementary Fig. 2e, f). Hematopoietic cells within BM of *CDH5-MAPK* mice also manifested a myeloid-biased output (increased *percentage* of CD11b+GR1+ cells within CD45+ BM cells), likely mediated by the effects of inflammatory cytokines that promote myeloid-biased differentiation of HSCs (Fig. 5c). The lineage-skewing in the BM was also reflected in their peripheral blood lineage composition (Fig. 5e). Collectively, these findings indicate that endothelial MAPK activation resulted in decreased blood counts due to loss of HSC and progenitor activity and was associated with a myeloid-biased output. Importantly, the BM and peripheral blood defects observed in *CDH5-MAPK* mice were completely restored in *CDH5-MAPK::IkB* mice. We next determined whether impaired hematopoiesis within BM of *CDH5-MAPK* mice could result in HSPC mobilization and extramedullary hematopoiesis. Peripheral blood analysis did not reveal significant differences in circulating KLS HSPCs demonstrating a lack of

HSPC mobilization in *CDH5-MAPK* mice (Fig. 5f). Analysis of the spleen in *CDH5-MAPK* mice revealed significant decreases in size and cellularity along with a decrease in frequency of HSCs and hematopoietic subsets, indicating an absence of extramedullary hematopoiesis (Fig. 5g–k). Lineage composition of CD45+ cells in the spleen revealed a myeloid bias (CD11b+GR1+ cells), confirming that endothelial MAPK-dependent inflammation adversely impacts niche activity and hematopoiesis within the spleen (Fig. 5l). Notably, *CDH5-MAPK::IkB* mice demonstrated a restoration of the defects observed in the spleen of *CDH5-MAPK* mice (Fig. 5g–l). These data demonstrate that *CDH5-MAPK* mice manifest hematopoietic defects, both in primary (bone marrow) and secondary (spleen) hematopoietic organs, and that suppression of endothelial NF-κB signaling overrides the hematopoietic defects.

**Endothelial MAPK activation drives an inflammatory stress response within the bone marrow.** The HSC and hematopoietic defects along with the vascular dilation within BM of *CDH5-MAPK* mice (Figs. 1 and 2) raised the possibility that downstream endothelial NF-κB activation induces a generalized inflammatory stress response within the BM microenvironment. In support of this idea, RT-qPCR analysis revealed an overall upregulation of NF-kB target genes in hematopoietic cells (CD45+), stromal cells (CD45-Ter119-CD31-VEcadherin-) as well as in unfractionated whole-bone marrow (WBM) cells of *CDH5-MAPK* mice (Supplementary Fig. 3a–f, Supplemetary Table 2, Source data), demonstrating that endothelial MAPK activation drives a generalized inflammatory response within the BM. The striking increase in expression of NF-kB target genes within the WBM of *CDH5-MAPK* mice (Supplementary Fig. 3e, f) raised the possibility of promiscuous expression of MEK1DD transgene within hematopoietic cells of *CDH5-MAPK* mice. To verify this possibility, we evaluated the fidelity of transgene expression in *CDH5-MAPK* mice by using the endogenous *Rosa26:eGFP* reporter system to track *cre*-mediated recombination (Fig. 6). GFP expression was strictly confined to ECs within the bone marrow with no detectable expression in any of the hematopoietic subsets analyzed including HSCs as well as myeloid cells, B cells, and T cells (Fig. 6a–e, Supplementary Fig. 4a, b). Flow cytometric analysis demonstrated that tamoxifen administration resulted in activation of endothelial MAPK signaling in vivo (Fig. 6c), and also confirmed the fidelity of the GFP reporter system to track recombination efficiency (Fig. 6d). Moreover, RT-PCR analysis of FACS-sorted cells demonstrated that *cre* expression was restricted to ECs and not detected in hematopoietic cells of *CDH5-MAPK* mice (Fig. 6f, g). These findings confirm previous reports that *Cdh5(PAC)-creERT2* mice demonstrate faithful endothelium-restricted expression in adult mice, with no off-target expression in HSCs or hematopoietic cells[44–47]. We also confirmed that

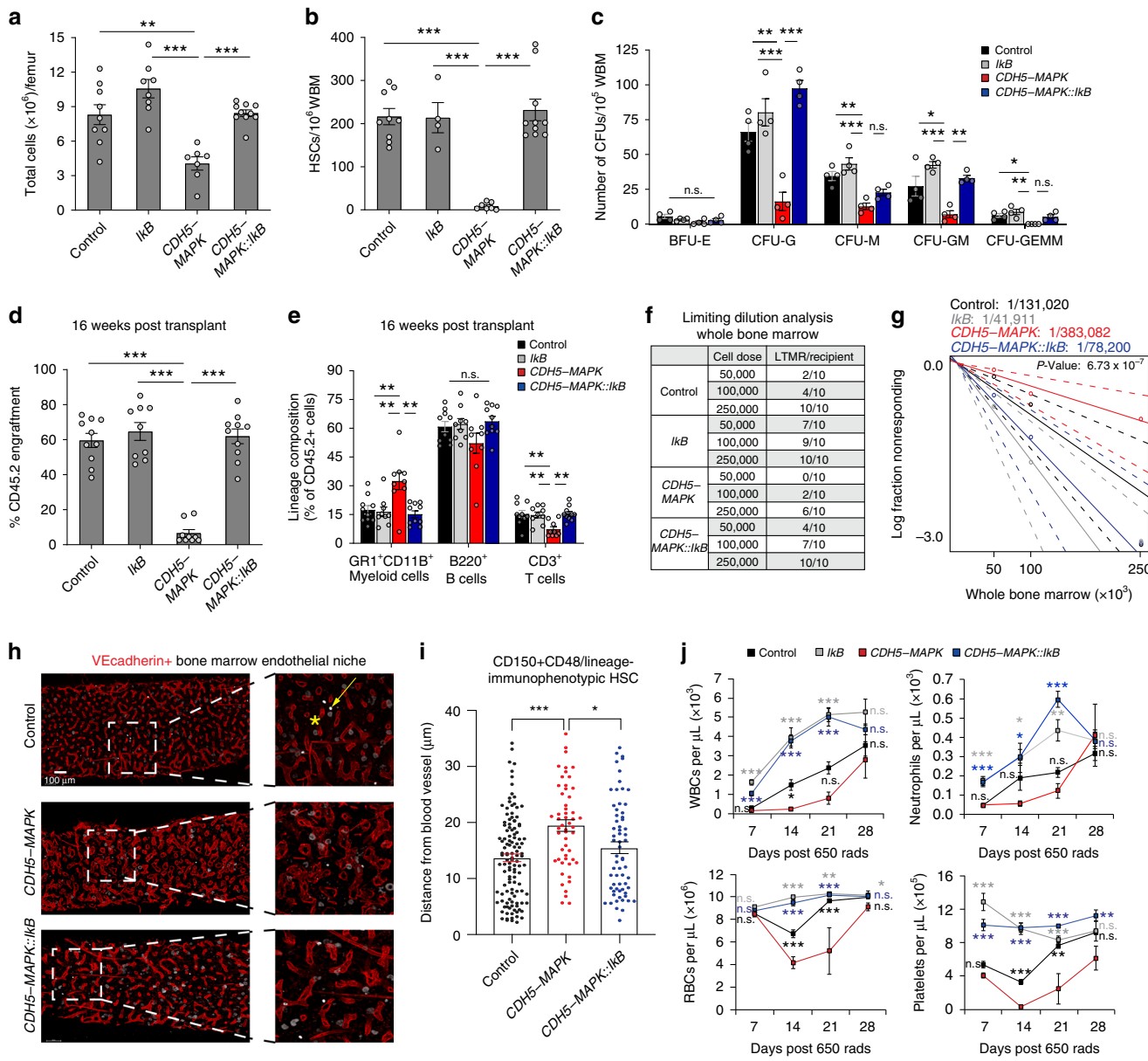

**Fig. 4 Endothelial NF-κB inhibition restores HSC activity in *CDH5-MAPK* mice. a** Total cells per femur (*n* = 7–10 mice/cohort). **b** Frequency of phenotypic HSCs per 10⁶ femur cells assessed by flow cytometry (*n* = 7–10 mice/cohort). Gating strategy described in Fig. 1b. **c** Methylcellulose-based progenitor assay. Bar graphs indicate number of CFUs per 10⁵ WBM (*n* = 4 mice/cohort). **d, e** Competitive repopulation assays assessing total CD45.2⁺ cell engraftment and CD45.2⁺ lineage distribution 4 months post-transplantation. For evaluating competitive repopulation, 5 × 10⁵ donor WBM cells (CD45.2) were transplanted with 5 × 10⁵ competitor WBM cells (CD45.1) into pre-conditioned CD45.1 recipient mice (*n* = 9–10 recipients/cohort; *n* = 5 donors per cohort). Gating strategy described in Supplementary Fig. 2b. **f** Table representing the number of recipients that were positive for long term, multi-lineage reconstitution (LTMR) following whole-bone-marrow-limiting dilution transplantation assay (*n* = 10 recipients/cohort per cell dose, *n* = 5 donors per cohort). **g** Log fraction plot of limiting dilution analysis indicates that *CDH5-MAPK::IkB* mice display an increase in frequency of HSCs capable of LTMR as compared to *CDH5-MAPK* mice. Dashed lines indicated 95% confidence intervals. Stem cell frequency and significance were determined using Extreme Limiting Dilution Analysis (ELDA). **h, i** Representative whole-mount immunofluorescence images of femurs and quantification of HSC distance from blood vessels of mice intravitally labeled with a vascular-specific CD144/VE-cadherin antibody (red) demonstrating that endothelial MAPK activation disrupts interactions of HSCs (white) with the vascular niche that are restored in *CDH5-MAPK::IkB* mice. Yellow arrowhead denotes a typical HSC (defined as Lineage^neg^CD48^neg^CD150^bright^) located in close proximity to a sinusoidal vessel. Yellow asterisk denotes a megakaryocyte. Each dot within the bar graph represents the distance of an individual HSC from its nearest blood vessel (*n* = 3 mice/cohort). **j** Time course of peripheral blood recovery following irradiation (650 rad) including red blood cells (RBCs). White blood cells (WBCs), platelets, and neutrophils (*n* = 6–10 mice/cohort). The colors of significance asterisks represent comparisons of the indicated genotypes with *CDH5-MAPK* mice. Results demonstrate a myeloprotective effect in *CDH5-MAPK::IkB* mice, indistinguishable from *Tie2.IkB-SS* mice. Error bars represent sample mean ± SEM. One-way ANOVA for multiple comparisons and Tukey's correction was performed to determine significance. *P ≤ 0.05; **P < 0.01; ***P < 0.001; ^n.s.^P > 0.05.

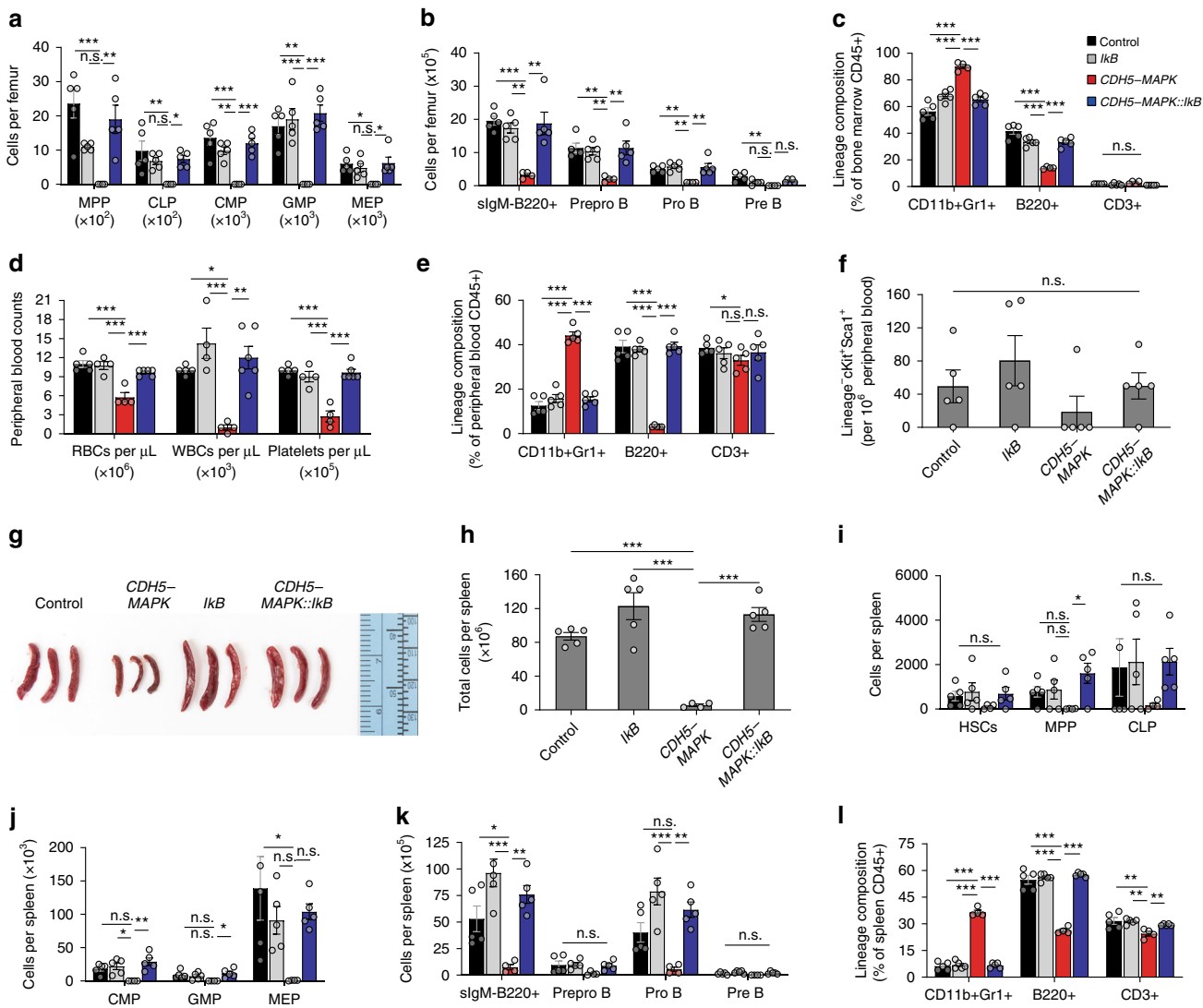

**Fig. 5 Endothelial NF-κB inhibition restores hematopoietic progenitor activity in *CDH5-MAPK* mice. a, b** Total cells per femur of the indicated hematopoietic progenitors estimated by flow cytometry ($n = 4-5$ mice/cohort). Gating strategy described in Supplementary Fig. 2e, f. **c** Lineage composition of CD45+ cells within BM ($n = 4-5$ mice/cohort). **d** Steady-state peripheral blood counts ($n = 4-6$ mice/cohort). **e** Lineage composition of CD45+ cells within peripheral blood ($n = 5$ mice/cohort). **f** HSPC frequency in peripheral blood ($n = 5$ mice/cohort). **g** Gross images of spleen from the indicated genotypes. **h** Spleen cellularity in the indicated genotypes ($n = 4-5$ mice/cohort). **i–k** Total cells per spleen of the indicated hematopoietic progenitors ($n = 4-5$ mice/cohort). **l** Lineage composition of CD45+ cells within spleen ($n = 4-5$ mice/cohort). Note that crossing *CDH5-MAPK* with *Tie2. IkB-SS* mice (*CDH5-MAPK::IkB*) restores the hematopoietic and HSPC attributes of *CDH5-MAPK* mice to control levels. Error bars represent sample mean ± SEM. One-way ANOVA for multiple comparisons and Tukey's correction was performed to determine significance. *$P \leq 0.05$; **$P < 0.01$; ***$P < 0.001$; n.s.$P > 0.05$.

expression of the IkB-SS transgene in *Tie2.IkB-SS* mice was restricted to ECs with no detectable expression in hematopoietic cells (Fig. 6f, g). The lack of *Tie2*-driven transgene expression in adult hematopoietic cells is consistent with previous reports[48–51]. Collectively, these observations confirm that HSC and hematopoietic defects in *CDH5-MAPK* mice are exclusively mediated from an endothelial NF-κB-dependent inflammatory stress within the BM. Importantly, endothelial-specific suppression of NF-κB signaling in *CDH5-MAPK::IkB* mice resolved the BM inflammation as indicated by unsupervised hierarchical clustering as well as the overall decrease in expression of NF-kB signaling targets (Supplementary Fig. 3a–f, Supplementary Table 2). Collectively, these results highlight that ECs, despite being a rare population within the BM (Fig. 6b), have a profound impact on HSC function and hematopoiesis during inflammatory stress.

**Endothelial NF-κB inhibition resolves inflammation-induced hypoxic injury.** The mechanisms by which BM inflammation impacts niche activity and HSC function remain poorly understood. Chronic inflammation is known to cause organ damage by inducing tissue hypoxia[52–54]. Furthermore, increased generation of reactive oxygen species (ROS) at sites of inflammation leads to endothelial dysfunction, vascular leakiness, and tissue injury[55]. Importantly, excessive ROS and hypoxia have been shown to adversely impact HSPC function by promoting loss of quiescence and exhaustion[56–58]. To this end, we examined the oxygenation status and ROS levels of HSPCs and BM niche cells (Fig. 7). HSPCs of *CDH5-MAPK* mice demonstrated a significant increase in hypoxia and ROS levels along with a loss of quiescence and increased apoptosis (Fig. 7a–d, Supplementary Fig. 5a, b). Notably, both BMECs and BM stromal cells also displayed increased

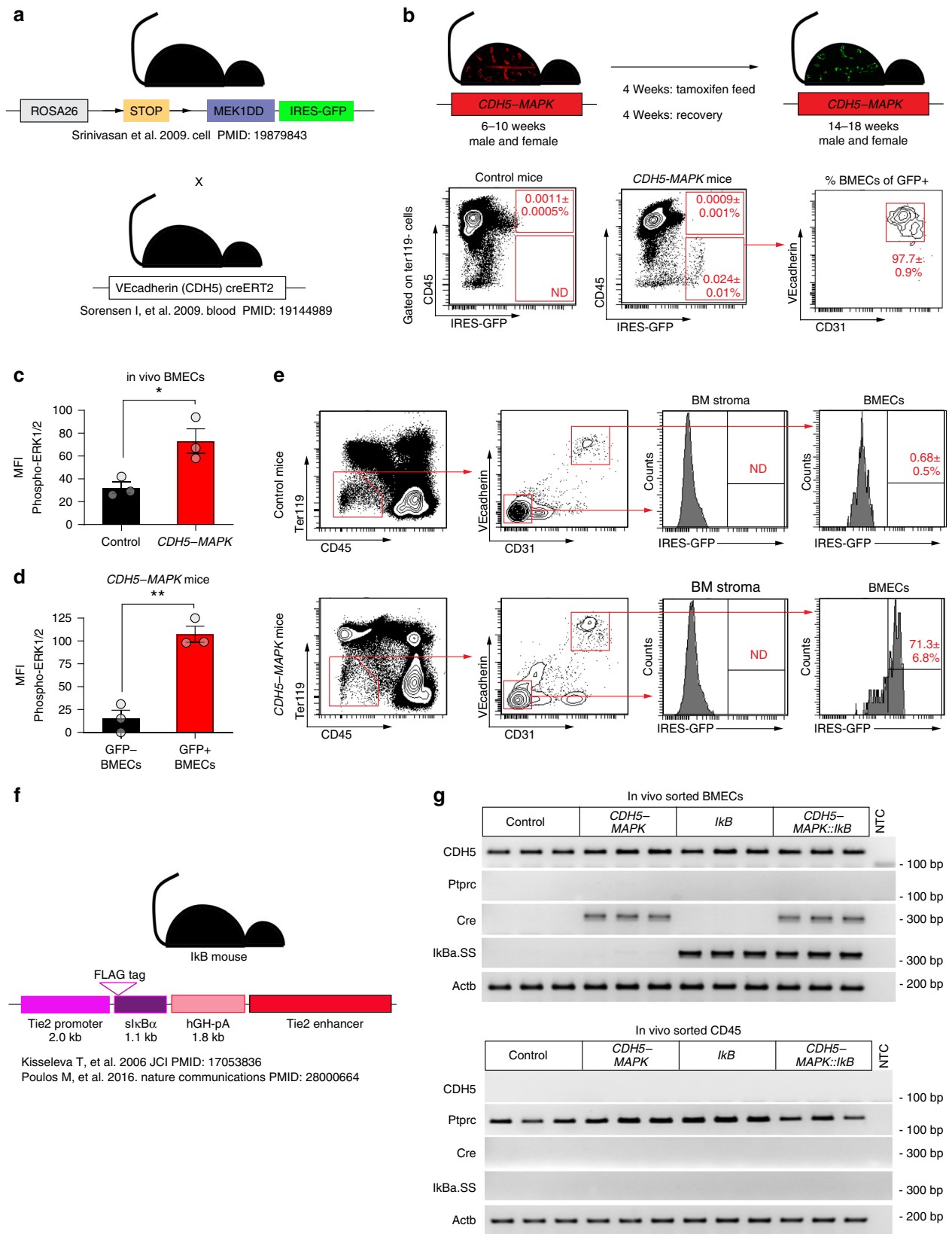

hypoxia (Fig. 7e), indicating that hypoxic injury induced by inflammation impacted both HSPCs and niche cells. Interestingly, niche cells did not display significant changes in ROS levels (Fig. 7f), indicative of a either a higher ROS detoxification ability or an impaired production in response to inflammatory stress.

However, BM stromal cells demonstrated a loss of cycling and increased apoptosis in CDH5-MAPK mice (Fig. 7g, h). Contrarily, the quiescent nature of BMECs observed under homeostasis was disrupted in CDH5-MAPK mice likely due to the direct effect of MAPK-driven cell-cycle entry (Fig. 7g). The alterations in

**Fig. 6 *CDH5-MAPK* and *Tie2.IkB-SS* mice demonstrate endothelial-specific expression of transgenes. a** Schematic describing breeding strategy for generation of *CDH5-MAPK* mice. **b** Schematic describing Tamoxifen regimen before experimental analysis. Representative flow cytometry contour plots demonstrating GFP expression in whole-bone-marrow cells of *CDH5-MAPK* mice. Numbers indicate average frequency of cells in the indicated quadrants as the percentage of total BM cells ± SEM (n = 4–5 mice/cohort). Note that GFP expression is detected exclusively in cells within the CD45− fraction of whole-bone-marrow cells and demonstrate surface expression of endothelial markers. **c** Analysis of phospho-ERK1/2 expression by flow cytometry confirms in vivo activation of the MAPK pathway in BMECs of *CDH5-MAPK* mice following Tamoxifen administration (n = 3 mice/cohort). **d** GFP+ BMECs as compared to GFP− BMECs within *CDH5-MAPK* mice demonstrate increased phospho-ERK1/2 expression by flow cytometry, confirming the fidelity of GFP reporter for tracking *cre*-mediated recombination in vivo (n = 3 mice/cohort). **e** Endothelial cells within the BM of *CDH5-MAPK* mice (defined as CD45− Ter119− CD31+ VEcadherin+) demonstrate *cre*-mediated recombination whereas stromal cells (defined as CD45− Ter119− CD31− VEcadherin−) do not (n = 4 mice/cohort). **f** Schematic describing the *Tie2.IkB-SS* mouse model. **g** Agarose gel electrophoresis image of RT-PCR amplicons for the indicated genes using RNA isolated from FACS-sorted endothelial cells and CD45+ hematopoietic cells in the indicated genotypes (n = 3 mice/cohort). Note that IkB-SS transgene is expressed in endothelial cells and shows no detectable expression in hematopoietic cells. Also note the expression of *cre* transgene in endothelial cells of *CDH5-MAPK* mice and no detectable expression in hematopoietic cells. NTC denotes "No Template Control". Sort purity was confirmed using expression of *Cdh5* (for endothelial cells) and *Ptprc* (for hematopoietic cells). Error bars represent sample mean ± SEM. Statistical significance was determined using two-tailed unpaired Student's *t*-test. *P ≤ 0.05; **P < 0.01.

cell-cycle status of niche cells in *CDH5-MAPK* mice were reflected in their BM cellularity wherein BMECs displayed an increase in absolute numbers whereas stromal cells were reduced (Fig. 7j, k). Notably, the defects observed in niche cells of *CDH5-MAPK* mice were restored in *CDH5-MAPK::IkB* mice (Fig. 7e–h).

BM niche cells, including endothelium and various BM stromal subsets, are known to play critical roles in HSC maintenance by expressing pro-HSC factors such as KitL and SDF1 (ref. [23]). To determine whether endothelial MAPK activation altered the levels of these HSC-regulatory factors, we assessed their expression in candidate BM niche cells by RT-qPCR. However, no significant alterations in their expression were observed in either BMECs or BM stromal cells among the genotypes (Fig. 7j, l). Within the BM stromal population, Lepr+ cells (which include Nestin+ and CXCL12-abundant reticular cells) have been shown to be an important source of KitL and SDF1 for HSC maintenance[59]. SDF1 is also known to be expressed by CD45−Ter119−CD31−Sca1−CD51+ BM osteoblast cells[60]. Analysis of BM Lepr+ cells and osteoblasts did not reveal significant changes in their cellularity or in their expression of HSC-regulatory factors in *CDH5-MAPK* mice (Supplementary Fig. 6a–e, g, h). However, both Lepr+ cells and osteoblasts of *CDH5-MAPK* mice displayed an increased expression of NF-κB regulated target genes similar to the other BM cellular subsets, which was suppressed upon inhibition of endothelial NF-κB signaling (Supplementary Fig. 6f, i). Collectively, these results demonstrate that the niche and HSPC defects observed in *CDH5-MAPK* mice correlate with inflammation-induced alterations in ROS levels and hypoxia, and that inhibition of NF-κB signaling within ECs of *CDH5-MAPK* resolves these defects.

To identify pro-inflammatory genes that mediate HSPC and niche defects in *CDH5-MAPK* mice, we surveyed the qPCR-array data (Supplementary Table 2) for genes that showed increased expression within the BM endothelial, hematopoietic, and stromal compartments; *Il1b*, *Csf1*, *Cdkn1a*, and *Csf2* were significantly upregulated in all three cellular subsets upon endothelial MAPK activation (Fig. 7m). Il1b and Csf1 have been shown to directly impact HSC function and promote myeloid-biased differentiation[10,61]. Chronic Il1 exposure has also been shown to cause enhanced HSC cycling and exhaustion[10]. The hematopoietic defects observed in *CDH5-MAPK* mice including increased HSPC cycling, impaired repopulating ability, and a myeloid-biased differentiation suggest that Il1b and Csf1 possibly mediate HSPC defects in *CDH5-MAPK* mice, and their suppression in *CDH5-MAPK::IkB* mice likely restores hematopoietic function. To verify this, we performed RT-qPCR analysis and confirmed that inhibition of endothelial NF-κB signaling in *CDH5-MAPK* mice decreased the expression of *Il1b* within

BMECs, stromal cells, and hematopoietic cells, while *Csf1* expression was decreased in stromal cells and hematopoietic cells (Fig. 7n–p). Notably, the decrease in endothelial *Il1b* expression correlated with a significant down-regulation of inflammation within the unfractionated WBM cells of *CDH5-MAPK::IkB* mice (Fig. 7q, Supplementary Fig. 3e), suggesting that endothelial *Il1b* likely plays a key role in mediating BM inflammation in *CDH5-MAPK* mice. However, the diverse changes observed in the plasma proteome of *CDH5-MAPK* mice suggest that chronic inflammation possibly involves a balance of multiple pro- and anti-inflammatory mediators that regulate inflammatory responses and HSC function (Fig. 2e, f).

**SCGF suppresses BM inflammation and restores HSC function in *CDH5-MAPK* mice.** Given that crossing *CDH5-MAPK* mice to *Tie2.IkB-SS* mice resolved their inflammation and restored vascular and hematopoietic defects, we utilized these models to screen for candidate proteins that might regulate HSC function during inflammation. To this end, we performed a proteomic analysis on plasma derived from *Tie2.IkB-SS* mice and identified 82 proteins that were differentially expressed as compared to their littermate controls (Supplementary Table 3, Supplementary Data 1). We hypothesized that a potential pro-hematopoietic protein would display opposing trends in *CDH5-MAPK* mice as compared to *Tie2.IkB-SS* mice. Using this approach, we identified 18 candidate factors that were significantly altered and inversely correlated (i.e. down in *CDH5-MAPK* mice, up in *Tie2.IkB-SS* and vice versa) (Supplementary Fig. 7a, b). Among these, Clec11a/stem cell growth factor-α (SCGF) was the most significantly downregulated protein in *CDH5-MAPK* mice (Supplementary Fig. 7c). SCGF has recently been identified as a potential rejuvenation factor for restoration of bone formation in aged mice[62]. Although SCGF has been reported to be dispensable for steady-state hematopoiesis, plasma levels of SCGF have been reported to be downregulated in patients with severe malarial anemia and decreased levels of plasma SCGF correlated with poor hematopoietic recovery following bone marrow transplantation, indicating that SCGF could play key roles during stress hematopoiesis[62–64]. We confirmed the specificity of the SCGF aptamer and validated the observed decrease of plasma SCGF in *CDH5-MAPK* mice (Supplemntary Fig. 7d). Notably, *CDH5-MAPK::IkB* mice displayed a restoration of their plasma SCGF levels, further indicating that SCGF could be a potential pro-hematopoietic factor that promotes recovery in *CDH5-MAPK::IkB* mice. (Supplementary Fig. 7e, f). To determine if SCGF can restore hematopoietic defects in *CDH5-MAPK* mice, we subcutaneously infused 4 μg of SCGF for five consecutive days and analyzed phenotypic and functional attributes of their hematopoietic

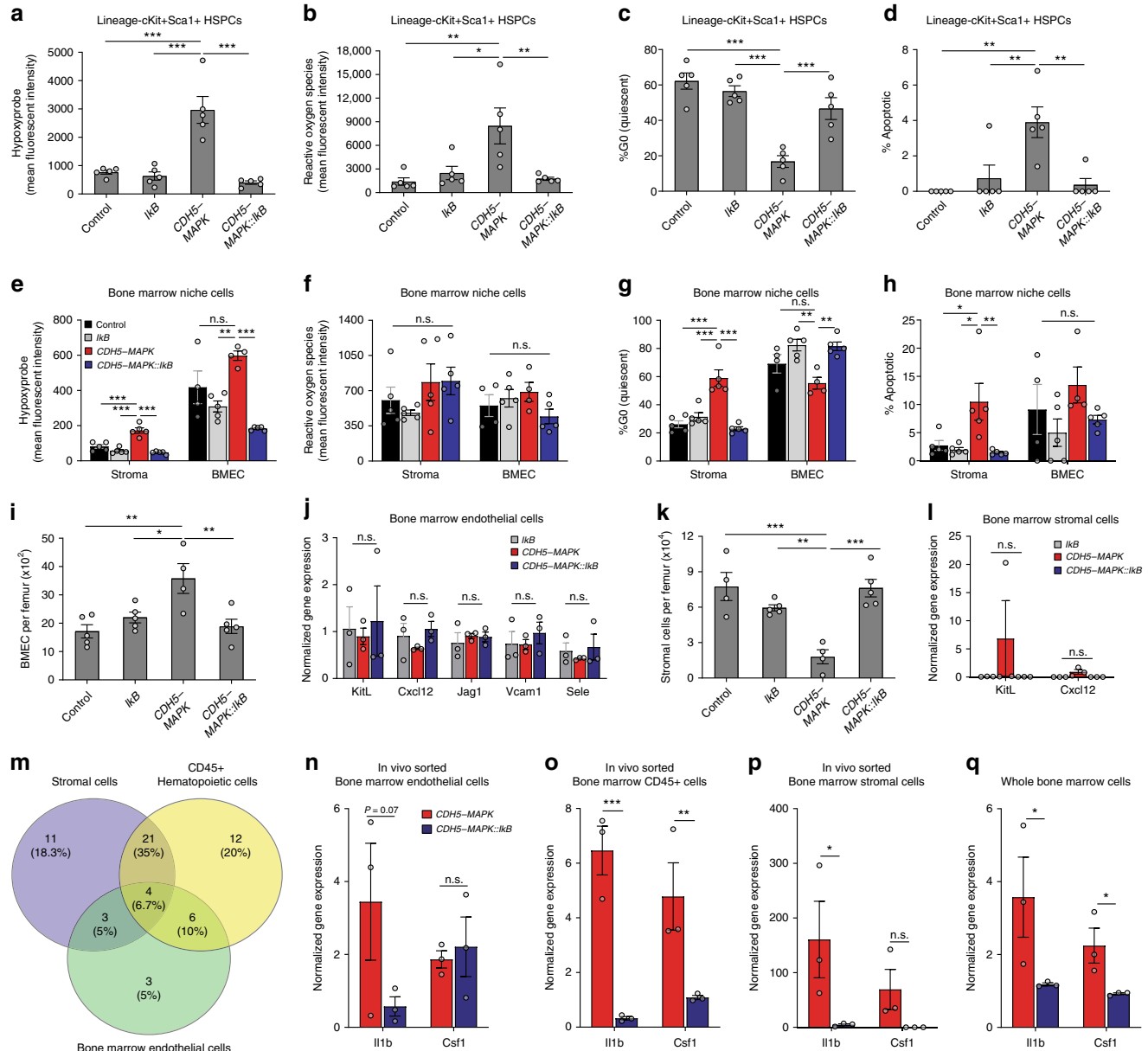

**Fig. 7 Endothelial NF-κB inhibition rescues hypoxic injury of HSPCs and BM niche cells. a** Estimation of oxygenation status in BM HSPCs based on quantification of hypoxyprobe by flow cytometry ($n = 5$ mice/cohort). Gating strategy described in Supplementary Fig. 5a. **b** Quantification of ROS levels in HSPCs by flow cytometry-based quantification of CellROX orange ($n = 5$ mice/cohort). Note that *CDH5-MAPK* HSPCs display increased hypoxia and ROS levels that are resolved by crossing with *Tie2.IkB-SS* mice. Gating strategy described in Supplementary Fig. 5b. **c** Cell-cycle analysis of HSPCs based on Ki67-Hoechst staining by flow cytometry ($n = 5$ mice/cohort). **d** Quantification of apoptosis in HSPCs by quantification of the percentage of cells in the sub-G0/G1 phase by flow cytometry ($n = 5$ mice/cohort). HSPCs from *CDH5-MAPK* mice display a loss of quiescence and increased apoptosis that is resolved upon suppression of their endothelial NF-κB signaling. **e–h** Quantification of hypoxia, ROS levels, quiescence, and apoptosis in the indicated BM niche cells by flow cytometry ($n = 4$–5 mice/cohort). **i, k** Estimation of BMECs and stromal cells per femur by flow cytometry ($n = 4$–5 mice/cohort). **j, l** Expression of pro-HSC paracrine factors in FACS-sorted BM niche cells by RT-qPCR ($n = 3$ mice/cohort). *Actb* was used for normalization. **m** Identification of commonly upregulated NF-κB target genes from qPCR-array data in BM stromal cells, hematopoietic cells, and ECs of *CDH5-MAPK* mice. **n–q** RT-qPCR confirmation of *Il1b* and *Csf1* expression in the indicated cell types ($n = 3$ mice/cohort). *Actb* was used for normalization. Notice that inhibition of endothelial NF-κB signaling significantly suppresses the expression of *Il1b* and *Csf1* within the whole bone marrow of *CDH5-MAPK* mice. Error bars represent sample mean ± SEM. One-way ANOVA for multiple comparisons and Tukey's correction was performed to determine significance. *$P \leq 0.05$; **$P < 0.01$; ***$P < 0.001$; n.s.$P > 0.05$.

system 24 h following the last injection (Supplementary Fig. 8a). SCGF infusion into littermate control mice confirmed that SCGF did not affect steady-state hematopoiesis as previously reported[62] (Fig. 8, Supplementary Fig. 8). However, infusion of SCGF into *CDH5-MAPK* mice had profound effects on their hematopoiesis (Fig. 8a–e, Supplementary Fig. 8a–e). SCGF infusion resulted in a

modest increase in BM cellularity (Supplementary Fig. 8b) and a significant increase in the frequency of phenotypic HSCs and HSPCs in *CDH5-MAPK* mice (Fig. 8a, b; Supplementary Fig. 8c). The increase in HSPC frequency was reflected in the enhanced colony-forming ability of BM cells isolated from SCGF-treated *CDH5-MAPK* mice (Fig. 8c). SCGF infusion also resolved the

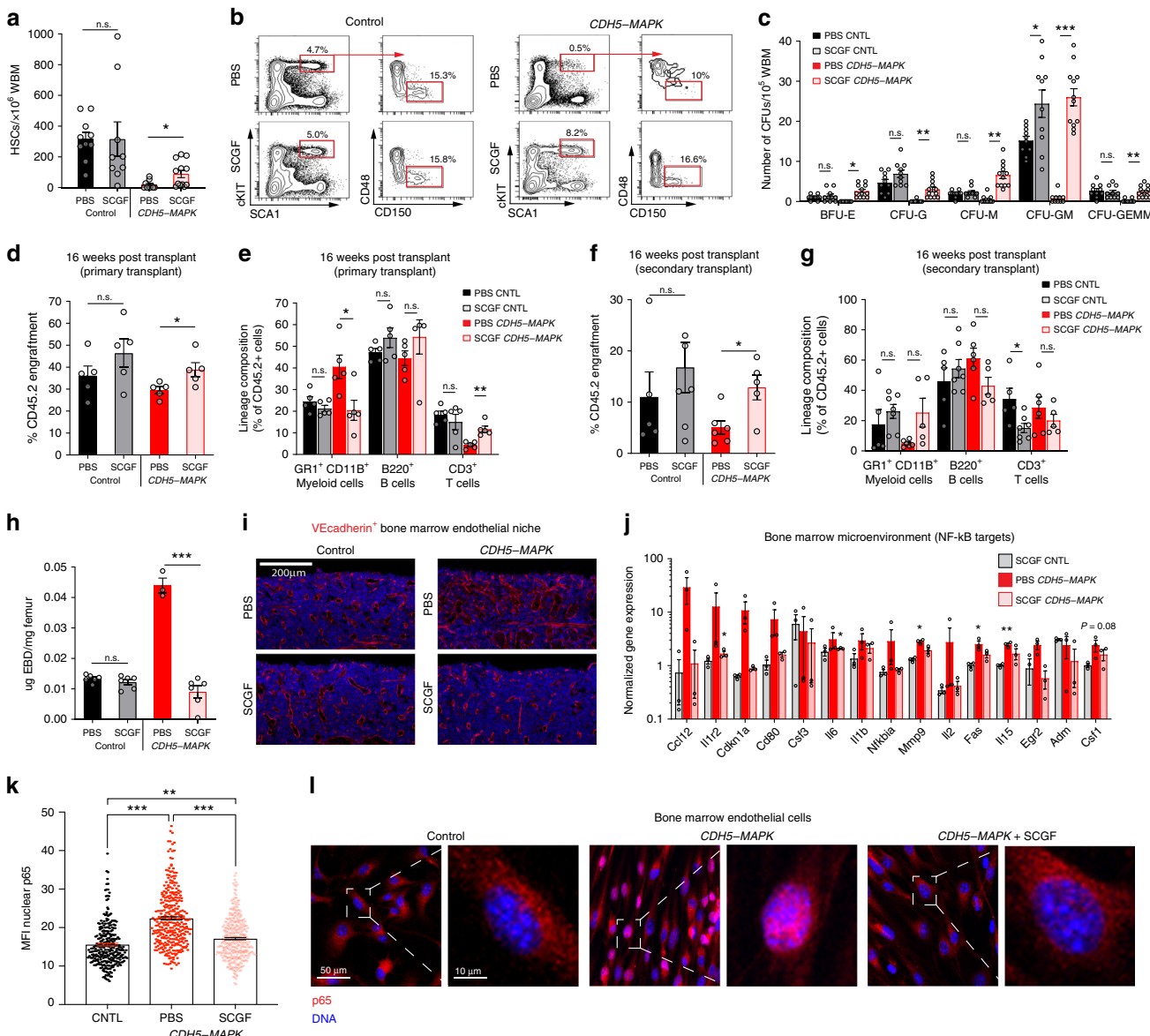

**Fig. 8 SCGF infusion resolves hematopoietic and vascular defects in *CDH5-MAPK* mice. a** Frequency of phenotypic HSCs per 10⁶ WBM (*n* = 9–10 mice/cohort). **b** Representative contour plots demonstrating that SCGF infusion restores phenotypic HSC defects observed in *CDH5-MAPK* mice. **c** Methylcellulose-based progenitor assay (*n* = 5 mice/cohort). **d, e** Competitive repopulation assay assessing total CD45.2⁺ cell engraftment and CD45.2⁺ lineage distribution 4 months post-transplantation (*n* = 5 recipients/cohort; *n* = 5 donors per cohort). For competitive repopulation assays, 5 × 10⁵ donor WBM cells (CD45.2) were transplanted along with 5 × 10⁵ competitor WBM cells (CD45.1) into pre-conditioned CD45.1 recipient mice. Note that donor cells derived from SCGF-treated *CDH5-MAPK* mice displayed a significant increase in engraftment efficiency with the resolution of the myeloid bias and increase in lymphoid output. Also, note that SCGF does not significantly impact the hematopoietic function or phenotype of control mice. **f, g** Secondary transplantation assay wherein WBM cells from long-term engrafted primary recipients were isolated and transplanted into pre-conditioned CD45.1 recipient mice (2 × 10⁶ donor WBM cells per recipient) (*n* = 5 recipients/cohort; *n* = 5 donors per cohort). **h** Analysis of BM vascular leakiness by Evan's Blue Dye (EBD) extravasation reveals that SCGF infusion significantly reduces vascular leakiness of *CDH5-MAPK* mice (*n* = 3–5 mice/cohort). **i** Representative immunofluorescence images of femurs intravitally labeled with a vascular-specific VECAD antibody (red) demonstrating reversal of vascular dilatation in *CDH5-MAPK* mice treated with SCGF. **j** Normalized gene expression of NF-κB target genes within the BM microenvironment of indicated genotypes as compared to PBS-treated control mice. B2m was used for normalization (*n* = 3 mice/cohort). **k, l** Representative immunofluorescence images and quantification demonstrating decreased levels of nuclear p65 in BMECs derived from *CDH5-MAPK* mice treated with SCGF. Error bars represent sample mean ± SEM. Statistical significance was determined using two-tailed unpaired Student's *t*-test for pairwise comparisons (**a**–**j**) and one-way ANOVA for multiple comparisons (**k**). *P ≤ 0.05; **P < 0.01; ***P < 0.001; n.s.P > 0.05.

peripheral blood myeloid bias in *CDH5-MAPK* mice and restored their blood counts (Supplementary Fig. 8d, e). Moreover, competitive BM transplantation demonstrated that SCGF infusion into *CDH5-MAPK* mice restored their long-term engraftment potential with a significant decrease in myeloid bias (Fig. 8d, e). The increased engraftment potential of BM cells derived from

SCGF-treated *CDH5-MAPK* mice was also maintained during secondary transplantation assays indicating a preservation of serial repopulation ability (Fig. 8f, g). Interestingly, the myeloid-biased output of *CDH5-MAPK* mice observed in primary transplantations was resolved in secondary transplantations indicating that inflammation-induced lineage-skewing of HSCs is not

permanent and is reversible upon exposure to a wild-type BM microenvironment during serial transplantations (Fig. 8f, g).

Given that SCGF knockout mice display normal hematopoietic parameters and accelerated bone loss[62], and the lack of discernible effects on hematopoiesis in SCGF-infused control mice (Fig. 8a–g), it is likely that the hematopoietic recovery observed in SCGF-infused *CDH5-MAPK* mice is possibly mediated by its effect on the vascular niche. Analysis of the endothelial niche revealed that infusion of SCGF resolved the vascular dilation and suppressed vascular leakiness within the BM microenvironment of *CDH5-MAPK* mice (Fig. 8h, i). The vascular recovery mediated by SCGF infusion was also associated with an overall decrease in expression of NF-kB target genes within the BM of *CDH5-MAPK* mice (Fig. 8j). The striking vascular recovery mediated by SCGF infusion raised the possibility that SCGF directly acts on the endothelium of *CDH5-MAPK* mice to suppress their NF-kB signaling. Immuno-fluorescence analysis demonstrated that SCGF treatment resulted in decreased nuclear p65 levels within BMECs isolated from *CDH5-MAPK* mice confirming a direct effect of SCGF on ECs (Fig. 8k, l). Taken together, these data indicate that infusion of SCGF into *CDH5-MAPK* mice suppresses endothelial inflammation and restores vascular integrity, which leads to a recovery of their hematopoietic system.

Since SCGF has been shown to promote osteogenesis, it is likely that the decrease in plasma SCGF levels along with the BM inflammation observed in *CDH5-MAPK* mice could result in osteopenia. *CDH5-MAPK* mice indeed displayed an overall decrease in trabecular bone volume, trabecular numbers, and thickness demonstrating that endothelial MAPK activation has a deleterious impact on bone health (Supplementary Fig. 9a–d). We next set forth to determine if SCGF could restore bone formation in *CDH5-MAPK* mice. Similar to its effects on hematopoiesis, SCGF did not affect bone formation in control mice (Supplementary Fig. 9a–d). However, infusion of SCGF caused a significant increase in trabecular bone volume and trabecular numbers and thickness in *CDH5-MAPK* mice, confirming its role in promoting osteogenesis (Supplementary Fig. 9a–d). Notably, SCGF expression was absent in hematopoietic cells and BMECs, and was primarily expressed in BM stromal cells including BM Lepr+ and osteoblastic stromal subsets (Supplementary Fig. 9e). Analysis of SCGF expression in total stromal cells, Lepr+ cells, and osteoblasts from BM of *Tie2-IkB-SS*, *CDH5-MAPK*, and *CDH5-MAPK::IkB* mice, however, revealed no significant changes in mRNA expression (Supplementary Fig. 9f), indicating that decreased plasma SCGF in *CDH5-MAPK* mice is not due to transcriptional alterations. It is known that cytokines mediating inflammatory responses can be regulated at the translational level and a recent report demonstrated that *Il1b* regulates the secretory response of chondrocytes by regulating translation[65,66]. Given that SCGF is a secreted protein and appears to regulate inflammatory responses, it is likely that it might be subject to translational regulation. However, the most likely explanation for decreased plasma SCGF in *CDH5-MAPK* mice appears to be due to the overall decrease in BM stromal cell numbers (the cells producing plasma SCGF) in *CDH5-MAPK* mice due to their apoptosis. (Fig. 7h, k). Collectively, these data indicate that SCGF infusion restores hematopoietic defects observed in *CDH5-MAPK* mice by restoring vascular integrity, resolving BM inflammation and improving bone health, thus suggesting that SCGF might play key roles in mediating hematopoietic recovery under stress situations.

**SCGF enhances hematopoietic regeneration following myelo-suppressive injury**. Myelosuppressive insults have been shown to

adversely impact the endothelial niche resulting in a loss of vascular integrity and delayed hematopoietic recovery[13,67]. In particular, ionizing radiation is known to activate NF-kB signaling within the endothelium leading to inflammation and endothelial dysfunction[68]. Given that SCGF infusion resolves vascular and hematopoietic defects in *CDH5-MAPK* mice, we sought to determine whether SCGF can enhance hematopoietic recovery following myelosuppressive stress. Wild-type mice were given a myelosuppressive dose of irradiation (650 rad) and infused every other day with either 0.5, 1, or 2 μg of SCGF for a total of seven injections starting at Day +1 post-irradiation and hematopoietic recovery was assessed for 21 days (Supplementary Fig. 10a). The dose–response experiment indicated that infusion of 2 μg of SCGF resulted in a significantly enhanced recovery of white blood cells, red blood cells, and platelets, confirming that SCGF enhances hematopoietic recovery following myelosuppressive stress (Supplementary Fig. 10a). Utilizing this strategy, we sought to test whether SCGF can improve hematopoietic recovery and preserve HSPC activity in both control and *CDH5-MAPK* mice (Supplementary Fig. 10b). We found that infusion of SCGF improved hematopoietic recovery following 650 rad of myelo-suppressive irradiation in both control and *CDH5-MAPK* mice (Fig. 9a, b). Immunofluorescence analysis of femoral BM sections at Day 28 post-irradiation revealed that SCGF infusion resulted in preservation of vascular integrity in both control as well as *CDH5-MAPK* mice (Fig. 9c, d). Evaluation of BM cellularity and HSC frequency after 28 days following myelosuppressive injury revealed that there was a significant increase in BM cellularity in both control and *CDH5-MAPK* mice treated with SCGF (Fig. 9e), while no significant changes were observed in phenotypic HSC frequency (Fig. 9f). Given that HSC frequency has been shown to poorly correlate with engraftment potential under conditions of inflammatory stress[69], we set forth to determine if SCGF infusion enhances HSC functionality following myelosuppression. To this end, we performed a competitive BM transplant on day 28 post-irradiation, wherein $2.5 \times 10^6$ donor WBM cells from *CDH5-MAPK* mice or littermate controls (treated with phosphate-buffered saline (PBS) or SCGF) were transplanted along with $5 \times 10^5$ CD45.1 competitor WBM cells into lethally irradiated (950 rads) CD45.1 mice (Fig. 9g, h). Analysis of CD45.2 cell engraft-ment 4 months post-transplant revealed that infusion of SCGF significantly enhanced long-term engraftment potential and multi-lineage reconstitution abilities for hematopoietic cells derived from both control and *CDH5-MAPK* mice (Fig. 9g, h). SCGF infusion was able to maintain the serial repopulation ability of hematopoietic cells derived from *CDH5-MAPK* mice during secondary transplantation assays (Fig. 9g, h), demonstrating that SCGF preserves HSC functionality in *CDH5-MAPK* mice both at steady state (Fig. 8d–g) and following myelosuppressive injury (Fig. 9g–j). Although SCGF infusion did not impact steady-state hematopoiesis in control mice (Fig. 8), these data show that SCGF preserves HSC function in control mice following myelosup-pressive injury (Fig. 9g, h), thereby uncovering an undefined role for SCGF during hematopoietic stress. Collectively, these findings demonstrate that SCGF has potent therapeutic properties that not only enhances vascular and hematopoietic recovery under an inflammatory stress, but also significantly improves the func-tionality of the HSC following myelosuppressive insult to the hematopoietic system.

## Discussion

The direct effects of specific inflammatory cytokines on HSC function have been extensively investigated[70]. However, the impact of chronic inflammation on HSC-supportive niche cells within the BM microenvironment remains poorly understood due to the

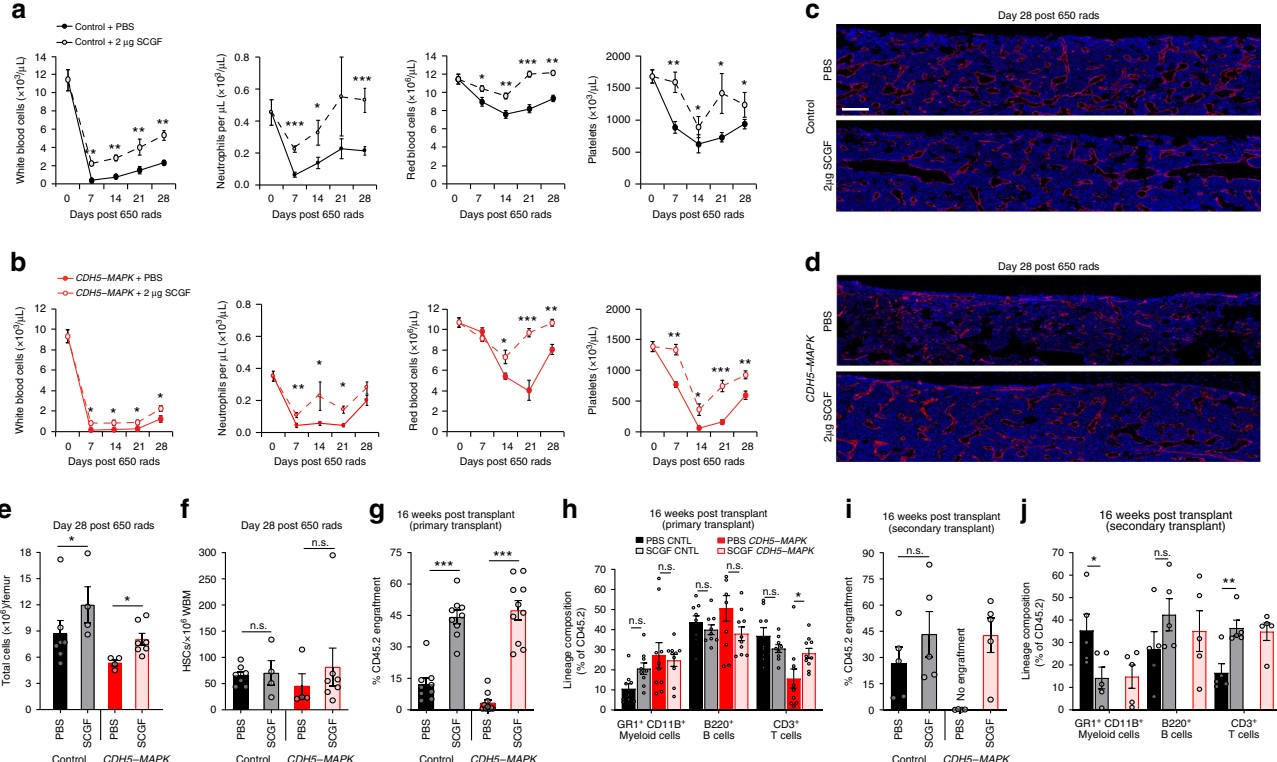

**Fig. 9 SCGF enhances hematopoietic regeneration following myelosuppressive injury. a, b** Following 650 rad of irradiation, 2 μg of SCGF was infused on alternate days into control and *CDH5-MAPK* mice for a total of seven injections starting at Day +1 and hematopoietic recovery was assessed for 28 days. SCGF infusion promoted a significant increase in white blood cell, neutrophil, red blood cell, and platelet recovery at indicated time points in **a** control mice (*n* = 6–7 mice/cohort) as well as **b** *CDH5-MAPK* mice (*n* = 7–8 mice/cohort). **c, d** Representative immunofluorescence images of femurs intravitally labeled with a vascular-specific CD144/VEcadherin antibody (red) at 28 days following irradiation demonstrating that SCGF infusion improved vascular recovery in both **c** control mice as well as **d** *CDH5-MAPK* mice. **e** Total cells per femur (*n* = 4–7 mice/cohort). **f** Frequency of phenotypic HSCs per $10^6$ femur cells assessed by flow cytometry (*n* = 4–7 mice/cohort). **g, h** Competitive repopulation assays assessing total CD45.2+ cell engraftment and CD45.2+ lineage distribution of donor WBM cells. Donor cells were isolated 28 days post-irradiation from PBS/SCGF-treated control and *CDH5-MAPK* mice. $2.5 \times 10^6$ donor WBM cells (CD45.2) were transplanted with $5 \times 10^5$ competitor WBM cells (CD45.1) into pre-conditioned CD45.1 recipient mice (*n* = 9–10 recipients/cohort; *n* = 5 donors per cohort). **i, j** Secondary transplantation assay wherein WBM cells from long-term engrafted primary recipients were isolated and transplanted into pre-conditioned CD45.1 recipient mice. $2 \times 10^6$ donor WBM cells were transplanted per secondary recipient (*n* = 4–5 recipients/cohort; *n* = 5 donors per cohort). Error bars represent sample mean ± SEM. Statistical significance was determined using two-tailed unpaired Student's *t*-test. *$P \leq 0.05$; **$P < 0.01$; ***$P < 0.001$; n.s.$P > 0.05$.

paucity of model systems that recapitulate microenvironment-derived inflammation. In this study, we demonstrate that sustained inflammation within the BM endothelial niche adversely impacts HSC function resulting from altered oxygenation status, ROS levels, and pro-inflammatory cytokine milieu within the BM micro-environment. Activation of MAPK signaling selectively within the endothelium of adult mice drives an NF-kB dependent inflammatory stress response within the BM microenvironment including HSPCs and multiple niche cells, highlighting the essential role of endothelium during chronic inflammation (Fig. 10). Moreover, inflammatory stress resulting from endothelial MAPK activation caused a significant impairment in hematopoietic recovery and HSC functionality following myelosuppressive injury. Importantly, resolution of inflammation by genetic (suppression of endothelial NF-kB) or pharmacological (SCGF infusion) means was able to resolve all of the hematopoietic and HSC defects observed due to endothelial MAPK activation, indicating that stem cell dysfunction induced by inflammation within tissue-specific microenvironments is reversible.

It has become increasingly clear that the NF-κB and MAPK pathways play essential roles  not only in hematopoiesis, but also in modulating inflammatory responses within the microvascular endothelium[25,32,71]. Within ECs, NF-κB serves as a master

regulator of a vast repertoire of pro-inflammatory cytokines[72]. In addition to the established roles of endothelial NF-κB signaling in launching immune responses against invading pathogens, it is also activated following injuries such as irradiation, leading to chronic vascular inflammation, tissue damage, and organ dysfunction[68,73]. Interestingly, NF-kB signaling within blood vessels remains activated for several years following radiation therapy, leading to sustained expression of pro-inflammatory cytokines[74]. Studies have suggested that inhibiting NF-κB may be beneficial in protecting against myeloablative therapy, graft rejection, and graft-versus-host disease by decreasing the quantity of cytokines secreted by the graft[32]. Indeed, we have recently demonstrated that inhibition of the canonical NF-κB signaling pathway specifically in ECs has a profound impact on enhancing both steady-state hematopoiesis as well as regeneration following irradiation induced myelosuppression, in part by decreasing pro-inflammatory cytokines[21]. Recent studies have begun to illuminate the role of MAPK signaling during regeneration of the hematopoietic system, in particular when patients are exposed to moderate to high doses of total-body irradiation[75]. The delay in hematopoietic recovery following radiation injury has been attributed to increased MAPK signaling[76]. Chronic endothelial MAPK activation has been shown to cause increased vascular

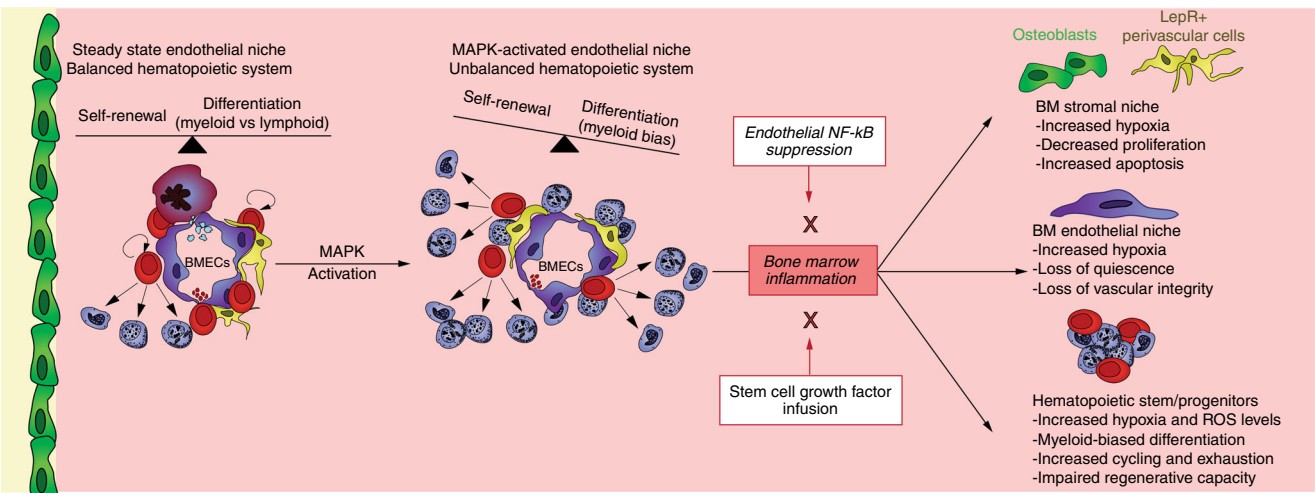

**Fig. 10 Schematic describing the impact of inflammation on BM niche cells and HSPCs.** Endothelial MAPK activation drives an NF-kB-dependent inflammatory stress response within the bone marrow leading to functional defects in the vascular niche and HSPCs. Suppression of inflammation by inhibition of endothelial NF-kB or infusion of SCGF restores vascular integrity, resolves HSPC and niche defects, and augments post-myelosuppressive hematopoietic recovery.

permeability, a hallmark of vascular dysfunction particularly following inflammation-induced injury[77,78]. Growing evidence suggests that MAPK activation in ECs results in increased vascular inflammation and endothelial dysfunction[29]. Collectively, these studies indicate that chronic inflammation within the endothelium might involve the interplay between both MAPK and NF-kB signaling pathways.

In the present study, we demonstrate that crosstalk between ERK-MAPK and NF-κB pathways regulate the outcomes of chronic endothelial inflammation within the BM and its resultant impact on niche activity and HSC function. The myeloid-biased output of HSCs observed in *CDH5-MAPK* mice illustrates the impact of chronic vascular inflammation on HSC function and highlights the potential of sustained niche-driven inflammation to influence aging-associated HSC phenotypes including predisposition towards myeloid neoplasms. Importantly, the complete rescue of hematopoietic defects observed in *CDH5-MAPK* mice upon endothelial NF-kB inhibition allows the opportunity to utilize these genetic models to derive testable hypotheses for interrogating stem cell niche interactions during chronic inflammation and to identify factors like SCGF that resolve inflammation-associated HSC and niche defects. SCGF/Clec11a is a member of the C-type lectin proteins belonging to the Tetranectin family[79]. Recent studies have highlighted the significant roles played by C-type lectins in the context of immunity, inflammation, and a wide array of physiological processes[79]. Although SCGF did not impact steady-state hematopoiesis in control mice, infusion of SCGF into *CDH5-MAPK* mice had tremendous benefits to the phenotypic and functional hematopoietic attributes indicating that SCGF might play key roles in mediating hematopoietic recovery under stress situations. The ability of SCGF to enhance post-myelosuppressive hematopoietic recovery in both control as well as *CDH5-MAPK* mice confirms its role as a rejuvenation factor during stress hematopoiesis. Considering the impact of SCGF in suppressing BM inflammation, restoring vascular integrity, promoting myelosuppressive recovery as well as its osteogenic properties, the identification of its gene regulatory mechanisms, cognate receptor/s, and downstream signaling pathways are exciting future directions. These studies will become important to understand the precise molecular mechanisms by which SCGF enhances hematopoietic regeneration and to develop treatment strategies directed towards

protecting the hematopoietic system and the BM endothelial niche following myelosuppressive therapies.

## Methods

**Animals.** C57BL/6J (CD45.2; stock no. 000664), B6.SJL-*Ptprc*^a *Pepc*^b/BoyJ (CD45.1; stock no. 002014), and C57BL/6-*Gt(ROSA)26Sor*^tm(Map2k1*,EGFP)Rsky/J[80] (*Mapk*^fl/fl) (stock no. 012352) mice were purchased from The Jackson Laboratory (Bar Harbor, ME). *Cdh5(PAC)-creERT2* mice (C57BL6 background) were obtained from Ralf H. Adams at The Max Planck Institute for Molecular Biomedicine[81]. *Tie2.IkB-SS* mice (C57BL6 background) were obtained from Jan Kitajewski at Columbia University[39]. *Lepr-cre* mice (C57BL6 background) were obtained from Sean J. Morrison at the University of Texas Southwestern Medical Center[82]. *Mapk*^fl/fl, *Cdh5(PAC)-creERT2*, *Lepr-cre*, and *Tie2.IκB-SS* mice were bred and maintained on a C57BL/6J (CD45.2) genetic background. All mice were housed in Positive Individual Ventilation (PIV) cages with HEPA-filtered air exchange (Thoren Caging Systems, Inc.) and maintained on Pico Lab Rodent Diet 20 (Lab Diet 5053) and water ad libitum. To induce *Cdh5(PAC)-creERT2*-mediated recombination, Tamoxifen (Sigma-Aldrich T5648) solubilized in sunflower oil (Sigma-Aldrich S5007) was administered via intraperitoneal injection (150 mg/kg body weight) at a dose of 30 mg/mL for three consecutive days at 8–12 weeks of age or fed Custom Teklad 2020 Feed supplemented with 0.025% w/w tamoxifen (Envigo) ad libitum at 6–10 weeks of age for four consecutive weeks. Age-matched *cre*-negative littermate mice also underwent the same tamoxifen induction regimen and were utilized as controls. Mice were allowed to recover for 4 weeks post-tamoxifen induction prior to experimental analysis. All mice were maintained in specific-pathogen-free housing. Total-body γ-irradiation (TBI) was administered from a ^137Cs source at doses indicated in the subsequent methodology. Irradiated recipients were given PicoLab Mouse 20 antibiotic feed (0.025% Trimethoprim and 0.124% Sulfamethoxazole; LabDiet) 24 h prior to irradiation and subsequently maintained for four weeks. All experiments were conducted in accordance with the Association for Assessment and Accreditation of Laboratory Animal Care, Intl. (AAALAC) and National Institutes of Health (NIH) Office of Laboratory Animal Welfare (OLAW) guidelines and under the protocols approved by Institutional Animal Care and Use Committee (IACUC) at the Center for Discovery and Innovation and Weill Cornell Medical College.

**Buffers and media.** Magnetic activated cell sorting (MACS) buffer: PBS without $Ca^{2+}/Mg^{2+}$ (pH 7.2) (Corning 21-040-CV) containing 0.5% W/V bovine serum albumin (BSA; Fisher Scientific BP1605) and 2 mM EDTA (Corning 46-034-CI).

Digestion buffer: 1× Hanks Balanced Salt Solution (Life Technologies 14065) containing 20 mM HEPES (Corning 25-060-CI), 2.5 mg/mL Collagenase A (Roche 11088793001), and 1 unit/mL Dispase II (Roche 04942078001).

Endothelial growth medium: 1:1 ratio of low-glucose DMEM (ThermoFisher Scientific 11885-084) and Ham's F-12 (Corning 10-080), supplemented with 20% heat-inactivated FBS (Denville Scientific FB5002-H), 1% antibiotic–antimycotic (Corning 30-004-CI), 1% non-essential amino acids (Corning 25-025-CI), 10 mM HEPES (Corning 25-060-CI), 100 μg/mL heparin (Sigma-Aldrich H3149), and 50 μg/mL EC growth supplement (Alfa Aesar BT-203)].

**Hematopoietic, BMEC, and stromal cell quantification**. To quantify total hematopoietic cells, femurs were gently crushed with a mortar and pestle and enzymatically disassociated for 15 min at 37 °C in Digestion Buffer following which cell suspensions were filtered (40 μm; Corning 352340) and washed in MACS buffer. Viable cell numbers were quantified using a hemocytometer with trypan blue (Life Technologies) exclusion. To quantify HSPCs in the BM, femurs and tibiae were flushed using a 26G × 1/2 needle with MACS buffer. To quantify splenic HSPCs, spleens were gently crushed and filtered over a 40 μm filter to obtain single-cell suspensions. To quantify BMECs, total BM stromal cells, BM Lepr+ cells, and BM osteoblasts, femurs were gently crushed with a mortar and pestle and enzymatically disassociated for 15 min at 37 °C in Digestion buffer following which cell suspensions were filtered (40 μm; Corning 352340) and washed in MACS buffer. Cells were surface stained using fluorochrome-conjugated antibodies as per the manufacturer recommendations. Cell populations were analyzed using flow cytometry.

**Flow cytometry**. Prior to cell surface staining, $F_c$ receptors were blocked using an antibody against CD16/32 (93; Biolegend) in MACS buffer for 10 min at 4 °C. For CMP/GMP/MEP staining, samples were blocked with 10% normal rat serum for 10 min at 4 °C. Blocked samples were subsequently stained with fluorochrome-conjugated antibodies in MACS buffer for 30 min at 4 °C as described. Samples stained with biotinylated anti-Lepr antibody were washed and stained with Streptavidin-conjugated fluorochromes for 15 min at 4 °C. Stained cells were washed in MACS buffer and fixed in 1% paraformaldehyde (PFA) in PBS (pH 7.2) with 2 mM EDTA. Sample data were collected and analyzed using an LSR II SORP (BD Biosciences) or Fortessa (BD Biosciences) with FACS DIVA 8.0.1 software (BD Biosciences). Gates were established using unstained controls and standard fluorescence minus one strategies. List of antibody clones utilized for flow cytometry is included in Supplementary Table 4.

Cell populations for flow cytometry and FACS sorting were defined as follows:

| HSC | Lineage (Ter119/CD11b/GR1/B220/CD3)−CD41−cKIT+ SCA1+CD48− CD150+ | Gating strategy in Fig. 1c |
|---|---|---|
| KLS | Lineage−cKIT+SCA1+ | Gating strategy in Fig. 1c |
| MPP | Lineage−cKIT+SCA1+ CD48−CD150− | Gating strategy in Fig. 1c |
| HPC-1 | Lineage−cKIT+SCA1+CD48+CD150− | Gating strategy in Fig. 1c |
| HPC-2 | Lineage−cKIT+SCA1+CD48+CD150+ | Gating strategy in Fig. 1c |
| CLP | Lineage−cKIT^low SCA1^low FLT3+IL7Rα+ | Gating strategy in Supplementary Fig. 2e |
| CMP | Lineage−cKIT+SCA1−CD34+CD16/32− | Gating strategy in Supplementary Fig. 2e |
| GMP | Lineage−cKIT+SCA1−CD34+CD16/32+ | Gating strategy in Supplementary Fig. 2e |
| MEP | Lineage−cKIT+SCA1−CD34−CD16/32− | Gating strategy in Supplementary Fig. 2e |
| Pre Pro B | sIgM−B220+CD43+CD24− | Gating strategy in Supplementary Fig. 2f |
| Pro B | sIgM−B220+CD43+CD24+ | Gating strategy in Supplementary Fig. 2f |
| Pre B | sIgM−B220+CD43−CD24+ | Gating strategy in Supplementary Fig. 2f |
| Myeloid | CD45+CD11B+GR1+ | Gating strategy in Supplementary Fig. 2b |
| B cells | CD45+B220+ | Gating strategy in Supplementary Fig. 2b |
| T cells | CD45+CD3+ | Gating strategy in Supplementary Fig. 2b |
| BMECs | CD45−Ter119−CD31+VEcadherin+ | Gating strategy in Supplementary Fig. 6a |
| BM stromal cells | CD45−Ter119−CD31−VEcadherin− | Gating strategy in Supplementary Fig. 6a |
| BM Lepr+ cells | CD45−Ter119−CD31−Lepr+ | Gating strategy in Supplementary Fig. 6b |
| BM osteoblasts | CD45−Ter119−CD31−SCA1−CD51+ | Gating strategy in Supplementary Fig. 6c |

**Progenitor activity**. Colony-forming units (CFUs) in semi-solid methylcellulose were quantified to assess hematopoietic progenitor activity. WBM was flushed from femurs and tibiae using a 26G × 1/2 needle with MACS buffer. Viable cell counts were determined with a hemocytometer using Trypan Blue (Life Technologies). WBM cells ($5 \times 10^4$ cells/well) were plated in duplicate in Methocult GF M3434 methylcellulose (StemCell Technologies) according to the manufacturer's

suggestions. Colonies were scored for phenotypic CFU-GEMM, CFU-GM, CFU-G, CFU-M, and BFU-E colonies using a SZX16 stereo-microscope (Olympus).

**Competitive transplantation and limiting dilutions**. Adult CD45.1 recipient mice were pre-conditioned with lethal irradiation (950 rad) 16 h prior to transplantation. WBM was isolated from femurs by gently crushing with a mortar and pestle and was enzymatically disassociated for 15 min at 37 °C in Digestion buffer following which cell suspensions were filtered (40 μm; Corning 352340) and washed in MACS to obtain single-cell suspensions. Viable cell numbers were quantified using a hemocytometer with Trypan Blue (Life Technologies) for live/dead exclusion. For competitive transplantation experiments at steady state (1:1 ratio), $5 \times 10^5$ donor WBM cells (CD45.2) were transplanted with $5 \times 10^5$ competitor WBM cells (CD45.1) via retro-orbital sinus injections into CD45.1 recipient mice pre-conditioned with myeloablative irradiation (950 rads). For competitive transplantation experiments following myelosuppressive injury (5:1 ratio), $2.5 \times 10^6$ donor WBM cells (CD45.2) were transplanted with $5 \times 10^5$ competitor WBM cells (CD45.1). Retro-orbital sinus bleeds using 75 mm heparinized glass capillary tubes (Kimble-Chase) were used to assess multi-lineage hematopoietic engraftment. Peripheral blood was depleted of red blood cells using RBC Lysis Buffer (Biolegend 420301) and stained with fluorochrome-conjugated antibodies according to the manufacturer's recommendations. Hematopoietic engraftment antibody panel includes CD45.1 (A20; Biolegend), CD45.2 (104; Biolegend), and TER119 (TER119; Biolegend). Multi-lineage engraftment panels include CD45.2 (104; Biolegend), GR1 (RB6-8C5; Biolegend), CD11B (M1/70; Biolegend), B220 (RA3-6B2; Biolegend), CD3 (17A2; Biolegend), CD4 (GK1.5; Biolegend), and CD8 (53-6.7; Biolegend). For limiting dilution analysis, indicated numbers of WBM were non-competitively transplanted via retro-orbital sinus injections into pre-conditioned CD45.1 recipient mice. Percent negative responding/dead mice were monitored for a four-month post-transplant period. Multi-lineage hematopoietic engraftment in surviving mice was confirmed by flow cytometry in red blood cell (RBC)-lysed peripheral blood using antibodies raised against CD45.2 (104; Biolegend), GR1 (RB6-8C5; Biolegend), CD11B (M1/70; Biolegend), B220 (RA3-6B2; Biolegend), and CD3 (17A2; Biolegend). HSC frequency and statistical significance was calculated using Extreme Limiting Dilution Analysis (ELDA) software (http://bioinf.wehi.edu.au/software/elda/)[83].

**Vascular permeability**. Bone marrow vascular integrity was examined as follows. 0.5% w/v Evans Blue Dye (Sigma-Aldrich E2129) in PBS (pH 7.2) was injected via tail vein at 25 mg dye/kg total body weight. Three hours post-injection, mice were sacrificed via cervical dislocation and cardiac perfused with 10 mL PBS (pH 7.2). Femurs were crushed in a mortar and pestle with 600 μL formamide and incubated at 55 °C overnight. Extractions were briefly vortexed and centrifuged at $16,000 \times g$ for 5 min at room temperature. Supernatant was removed and absorbance (Abs) was measured at 620 and 740 nm. Sample Abs was corrected for heme-containing proteins $[Abs_{620} - (1.426 \times Abs_{740} + 0.03)]$ and blanked using non-injected controls [corrected sample $Abs_{620}$ − corrected non-injected control $Abs_{620}$]. Evan's blue dye extravasation was calculated using a standard curve and normalized to femur weight.

**Immunohistochemistry**. To label the vasculature, mice were intravenously administered 25 μg of Alexa Fluor 647-conjugated CD144/VEcadherin antibody (Clone BV13; Biolegend) via retro-orbital sinus injections. Animals were sacrificed 10 min post-injection and cardiac perfused with 10 mL PBS (pH 7.2). Femurs were fixed overnight in 4% PFA in PBS (pH 7.2), decalcified in 10% EDTA for 72 h at room temperature, cryopreserved in 30% sucrose for 48 h at 4 °C, and embedded in 50% optimal cutting temperature (OCT) and 50% sucrose. Longitudinal femur sections (12 μm) were cut using a CM 3050S Cryostat (Leica), counterstained with 1 μg/mL 4-6,diamidino-2-phenylindole (DAPI) (Biolegend), and mounted using Prolong Gold anti-fade solution (Life Technologies). Sections were imaged on a LSM 710 confocal microscope (Zeiss).

**Whole-mount immunofluorescence**. Mice were intravenously administered 25 μg of Alexa Fluor 647-conjugated CD144/VEcadherin antibody (Clone BV13; Biolegend) via retro-orbital sinus injections. After 10 min, mice were euthanized and cardiac perfused with 4% PFA following which femurs were isolated, stripped of muscle and connective tissue, and fixed in 4% PFA for 30 min at room temperature. Bones were washed in 1× PBS 3 × 5 min and cryopreserved in 15% sucrose for 24 h at 4 °C, and further cryopreserved in 30% sucrose for 24 h at 4 °C. Bones were then embedded in a 50% OCT and 50% sucrose solution and flash frozen in liquid nitrogen. Bones were shaved longitudinally on a Leica CM 3050S cryostat in order to fully expose the bone marrow cavity for antibody penetration. Shaved bones were unmounted and washed 3 × 5 min in 1× PBS until OCT was completely melted. Exposed bones were blocked for 2 h at room temperature in blocking buffer [20% normal goat serum (Jackson Laboratories) in 1× PBS containing 0.5% Triton X-100], protected from light. Bones were then stained with fluorochrome-conjugated primary antibodies (Supplementary Table 5) diluted in blocking buffer by immersion incubation in 1.5 mL Eppendorf microcentrifuge tubes for 48 h at 4 °C. Bones were washed 3 × 10 min in 1× PBS. Forty-micrometer Z-stack images were acquired on a Nikon C2 confocal laser scanning microscope. HSCs were

defined by Lineage[neg]CD48[neg]CD150[bright] cells and their distance relative to the nearest vascular cell (VE-cadherin/CD144+) was measured for quantification.

**Proteomic analysis.** Plasma proteome analysis was performed at SomaLogic (Boulder, CO) using the aptamer-based SomaScan platform[84]. To generate plasma, mice were bled via the retro-orbital sinus using 75 mm heparinized glass capillary tubes (Kimble-Chase) into EDTA containing microcentrifuge tubes (5 μM final concentration). Whole blood was centrifuged at $2200 \times g$ for 15 min at room temperature and plasma was collected and stored at $-80\,^\circ C$. Cryopreserved mouse plasma-EDTA samples were analyzed using the SomaScan Assay 1.1K platform and proteomic data are presented as relative fluorescent units (RFUs). To identify differentially expressed proteins, two-sided Student's $t$-test was performed with the threshold of significance set at $P \leq 0.075$ using SomaSuite Software (SomaLogic). Core analysis was performed on the entire dataset ($P$ value cut-off $\leq 0.05$, Fisher's exact test) using Ingenuity Pathway Analysis (Qiagen) to identify biological processes that are significantly enriched in *CDH5-MAPK* mice. (https://www.qiagenbioinformatics.com/products/ingenuity-pathway-analysis/).

**Lentivirus.** Myristoylated-Akt1 (*myrAkt1*) lentivirus was generated by co-transfecting *pCCL-myrAkt1* backbone[15] (13 μg) with RRE (5 μg), REV (2.5 μg), and VSV-G (3 μg) packaging plasmids on a 10 cm dish of 80% confluent 293T/17 cells (ATCC CRL-11268) using Lipofectamine 2000 (ThermoFisher Scientific 12566-014) according to the manufacturer's suggestions. Forty-eight hours post-transfection, supernatants were processed using a Lenti-X Concentrator (ClonTech 631232) according to the manufacturer's suggestions. Precipitated *myrAkt1* lentivirus was resuspended in 0.5 mL TNE Buffer (50 mM Tris pH 8.0, 1 mM EDTA, 130 mM NaCl), aliquoted, and stored at $-80\,^\circ C$. Viral titers were determined using Lenti-X p24 Rapid Titer Kit (ClonTech 632200). *IkB-SS* expressing lentiviral vector was generated by sub-cloning the sequence of human IκBα super-suppressor (Addgene #15264) into pLVX Puro Vector (Clontech #632164). *IkB-SS* lentivirus was generated by co-transfecting pLVX-Puro-IkB-SS vector with RRE, REV, and VSV-G packaging plasmids in 293T cells as described earlier. pLVX Puro Vector (Clontech #632164) was utilized to generate the "Puro-empty" lentivirus.

**EC cultures.** Primary BMEC cultures were generated from *Cdh5(PAC)-creERT2*; *Mapk[fl/fl]*, as follows. Femurs and tibiae were gently crushed using a mortar and pestle and digested with Digestion buffer for 15 min at $37\,^\circ C$, filtered (40 μm; Corning 352340), and washed in MACS buffer. WBM was depleted of terminally differentiated hematopoietic cells using a murine Lineage Cell Depletion Kit (Miltenyi Biotec 130-090-858) according to the manufacturer's recommendations. BM endothelium was immunopurified from cell suspensions using sheep anti-rat IgG Dynabeads (ThermoFisher Scientific 11035) pre-captured with a CD31 antibody (MEC13.3; Biolegend) in MACS buffer according to the manufacturer's suggestions. CD31 selected BMECs were cultured in endothelial growth media and transduced with $10^4$ pg *myrAkt1* lentivirus per $3 \times 10^4$ ECs/cm². *Akt*-transduced BMECs were selected for 7 days in serum- and cytokine-free StemSpan SFEM (StemCell Technologies, Inc. 09650) media. BMEC lines were stained with antibodies against VECAD (BV13; Biolegend), CD31 (390; Biolegend), and CD45 (30-F11; Biolegend) and FACS sorted for purity (BMEC defined as CD45− CD31+ VEcadherin+). Established BMEC lines were transduced either with GFP lentivirus (Control BMECs) or GFP-Cre lentivirus (*CDH5-MAPK* BMECs) and the resultant GFP+ cells were FACS sorted to purity. Control and *CDH5-MAPK* BMECs were subsequently transduced with either Puro-empty or Puro-IKB-SS lentivirus to generate, respectively, Control, *IkB*, *CDH5-MAPK*, and *CDH5-MAPK::IkB* cell lines. Transduced cell lines were selected with 2 μg/mL puromycin for 5 days. Cells were cultured in endothelial growth medium at $37\,^\circ C$, 5% $CO_2$, and 20% $O_2$ in 70% relative humidity. Growth media was changed every 2 days and cells were passaged 1:2 at 95% confluency with Accutase Cell Detachment Solution (Biolegend 423201) according to the manufacturer's suggestions.

**Immunoblots.** Established BMEC lines were serum starved in low-glucose DMEM for 36 h prior to preparing cell lysates. Cultured cells were washed with ice-cold PBS (pH 7.2) and lysed in RIPA buffer (150 mM NaCl, 1% IGEPAL CA-630, 0.5% deoxycholate, 0.1% sodium dodecyl sulfate (SDS), and 50 mM Tris-HCl, pH 8.0) with PhosStop Phosphatase Inhibitor (Roche 04906845001) and Complete EDTA-free Protease Inhibitor Cocktail (Roche 11836170001) for 20 min at $4\,^\circ C$, sonicated, and centrifuged for 10 min at $21,000 \times g$ at $4\,^\circ C$ to remove insoluble debris. Supernatants were stored at $-80\,^\circ C$. Protein concentrations were determined using the DC Protein Assay (BioRad 5000111) and 5 μg total protein was denatured for 3 min at $95\,^\circ C$ in Laemmli Buffer, resolved on 12.5% SDS-acrylamide gels, and electroblotted to nitrocellulose. Transferred blots were blocked for 1 h in 5% non-fat dry milk in PBS (pH 7.2) with 0.05% IGEPAL CA-630 (Sigma-Aldrich I8896) and incubated overnight at $4\,^\circ C$ in 5% non-fat dry milk in PBS (pH 7.2) with 0.05% IGEPAL CA-630 with primary antibodies raised against phospho-p65 (Ser536) at 1:1000 (Cell Signaling 3033), p65 at 1:1000 (Cell Signaling 4764), phospho-ERK1/2 (Thr202/Tyr204) at 1:2000 (Cell Signaling 4370), ERK1/2 at 1:1000 (Cell Signaling 9102), Total IκBα at 1:1000 (Cell Signaling 4814), and Tubulin at 1:1000 (Cell Signaling 2146). Blots were washed $3 \times 10$ min in PBS (pH 7.2) with 0.05% IGEPAL CA-630 at room temperature and incubated in 5% non-fat dry milk in PBS (pH

7.2) with 0.05% IGEPAL CA-630 and anti-rabbit or anti-mouse IgG (H + L) horseradish peroxidase (Jackson ImmunoResearch Laboratories) secondary antibodies at a dilution of 1:10,000 for 1 h at room temperature. Blots were washed $3 \times 10$ min in PBS (pH 7.2) with 0.05% IGEPAL CA-630 at room temperature and developed using Amersham ECL Prime Western Blotting Detection Reagent (GE Healthcare RPN2232), according to the manufacturer's suggestions. All blots were developed using Carestream Kodak BioMax Light Film (Sigma-Aldrich).

**Quantification of nuclear p65.** For assessment of nuclear p65, BMECs were plated in endothelial growth medium in chamber slides (Nunc Lab-Tek II CC² Chamber Slide; Catalog # 154941). At ~70% confluency, cells were serum starved in low-glucose DMEM for 36 h following which cells were washed in PBS (pH 7.2) and fixed in 4% PFA in PBS (pH 7.2) for 15 min at room temperature. Cells were then permeabilized with 5% normal goat serum containing 1% Triton X-100 in PBS (pH 7.2) for 30 min. Cells were then stained with p65 antibody (C22B4; Cell Signaling, 1:100 dilution) in antibody dilution buffer (1% Triton™ X-100 in PBS containing 1% BSA) for 1 h at room temperature. Cells were washed three times with PBS and stained with goat anti-rabbit Alexa Fluor 647 (Thermo Scientific #A-21245, 1:250 dilution) in antibody dilution buffer for 30 min at room temperature. Cells were washed three times with PBS and counterstained with DAPI at 1 μg/mL and mounted using Prolong Gold anti-fade solution (Life Technologies). To determine the effect of SCGF on nuclear p65 levels, BMECs derived from *CDH5-MAPK* mice were incubated with 0.4 μg/mL SCGF (or vehicle control) for 36 h in low-glucose DMEM. Concentration-matched isotype control antibody (Cell Signaling #3900) was used for establishing background fluorescence. Images were acquired on a Nikon C2 confocal laser scanning microscope. Nuclear p65 levels were quantified using Image J by measuring mean fluorescence intensity of p65 in every nucleus demarcated by DAPI staining[36].

**Gene expression analysis.** WBM was flushed from femurs and tibiae using a 26G×1/2 needle with MACS buffer and depleted of red blood cells using 1× RBC Lysis Buffer (Biolegend 420301) according to the manufacturer's recommendations. Total RNA was isolated from $4 \times 10^6$ RBC-lysed WBM cells using RNeasy plus Mini Kit (Qiagen 74134) according to the manufacturer's instructions. Briefly, cells were lysed in 600 μL of Buffer RLT and homogenized using QIAshredder columns (Qiagen 79654). For RNA isolation from CD45 cells, stromal cells, Lepr+ cells, and osteoblasts, cells were directly sorted into Trizol LS using FACS and RNA was purified using the manufacturer's recommendations. RNA was isolated from 100,000 CD45+ cells per sample and 1000 cells per sample for stromal cells, Lepr+ cells, and osteoblasts. For WBM and CD45 cells, total RNA was reverse transcribed using RT2 First Strand Kit (Qiagen 330401). cDNA generated from 100 ng total RNA was subsequently loaded on RT2 PCR profiler arrays to evaluate gene expression of NF-kB signaling targets (Qiagen PAMM-225ZC). For stromal cells, Lepr+ cells, and osteoblasts, cDNA was generated and amplified using the Ovation Pico WTA System V2 (Nugen) according to the manufacturer's suggested protocol and 100 ng amplified cDNA was utilized for the qPCR arrays. qPCR was performed using RT2 SYBR Green qPCR Mastermix (Qiagen 330522) in a ViiA 7 qPCR system (Applied Biosystems) with recommended cycling parameters. Qiagen's online data analysis tool was utilized to calculate fold changes, generate unsupervised hierarchical clustering, and gene expression heatmaps (https://www.qiagen.com/in/shop/genes-and-pathways/data-analysis-center-overview-page/). Reference genes for normalization were selected from a panel of five housekeeping genes (*Actb, B2m, Gapdh, Gusb,* and *Hsp90a1b*) using "automatic selection of housekeeping genes" in the Qiagen online tool which selects the most stable reference gene for each condition. Fold changes were calculated using the $2^{-\Delta\Delta CT}$ method. Confirmation of *Il1b* and *Csf1* expression was performed by RT-qPCR using primers obtained from Qiagen (Cat # PPM03109F-200 and PPM03116C-200). Primers for RT-qPCR analysis are listed in Supplementary Table 6. For evaluation of *cre* and *IkB-SS* transgene expression in hematopoietic cells, RNA was isolated from 100,000 CD45+ cells per sample. Total RNA was reverse transcribed using RT2 First Strand Kit (Qiagen 330401). cDNA equivalent to RNA content of 2000 cells was utilized for RT-PCR analysis. Primers for RT-PCR analysis are listed in Supplementary Table 6.

**Peripheral hematopoietic recovery.** Mice were irradiated with sublethal irradiation (650 rad) from a $^{137}Cs$ source for evaluating hematopoietic recovery following myelosuppressive injury. Peripheral blood was collected using 75 mm heparinized glass capillary tubes (Kimble-Chase) via retro-orbital sinus bleeds at indicated time points. WBC, RBC, and platelet populations were analyzed using an Advia120 (Bayer Healthcare).

**ROS estimation.** To examine ROS, mice were intravitally labeled for 10 min with 25 μg Alexa Fluor 647-conjugated CD144/VE-Cadherin antibody (BV13; Biolegend) via retro-orbital injection. Mice were sacrificed and femurs were either flushed (for HSPC analysis) or gently crushed and enzymatically disassociated (for BMEC and stromal cell analysis) in Digestion buffer for 15 min at $37\,^\circ C$ with gentle agitation. Cell suspensions were filtered (40 μm) and washed in MACS buffer followed by surface staining using the indicated antibodies for 20 min at $4\,^\circ C$. Stained cell suspensions were washed in MACS buffer and then incubated with 5

µM CellROX Orange (ThermoFisher Scientific) in Stemspan SFEM (StemCell Technologies) at 37 °C for 30 min, washed with MACS buffer, and resuspended in PBS containing 2 mM EDTA. ROS levels in the indicated cell types were estimated using flow cytometry.

**Hypoxyprobe**. To evaluate bone marrow oxygenation status, mice were co-injected with 100 mg/kg of Pimonidazole HCl (hypoxyprobe-1; Hypoxyprobe, Inc.) and 25 µg Alexa Fluor 647-conjugated CD144/VE-Cadherin (BV13; Biolegend) via retro-orbital injection. Following 20 min, mice were euthanized and femurs were isolated. Femurs were either flushed (for HSPC analysis) or gently crushed and enzymatically disassociated (for BMEC and stromal cell analysis) in Digestion buffer for 15 min at 37 °C with gentle agitation. Following surface staining, cells were fixed and permeabilized using the BD Cytofix/Cytoperm Kit (BD Biosciences) and stained with a monoclonal antibody raised against hypoxyprobe-1 at a 1:100 dilution (HP-Red549; Hypoxyprobe, Inc.) according to the manufacturer's suggestions. Hypoxyprobe levels in the indicated cell types were estimated using flow cytometry.

**Cell cycle and apoptosis**. For cell-cycle analysis of BMECs, stromal cells, and HSPCs, cells were surface stained, fixed, and permeabilized using the BD Cytofix/Cytoperm Kit (BD Biosciences) as described in the preceding section following which the cells were stained with an antibody raised against Ki67 (B56, BD 561284) and counterstained with Hoechst 33342 (BD Biosciences), according to the manufacturer's recommendations. For cell-cycle analysis of HSCs, WBM cells were first depleted of lineage-positive cells using a lineage cell depletion kit (Miltenyi Biotec # 130-110-470) prior to surface staining. Cell were analyzed using flow cytometry with a low acquisition rate (350 events/s). Cell-cycle status was classified as follows: G0 (Ki-67$^{negative}$; 2N DNA content), G1 (Ki-67+; 2N DNA content), and S/G2/M (Ki-67+; >2N DNA content). Percentage of singlet cells in the sub-G0/G1 area were classified as apoptotic.

**Plasma ELISA**. For aptamer-based sandwich ELISAs, streptavidin-coated 96-well plates (ThermoFisher Scientific 15124) were incubated at 4 °C overnight with 20 nM biotinylated αClec11a aptamer (SomaLogic; Boulder, CO Cat# B-4500-50_2) in SBT buffer (40 mM HEPES pH 7.5, 120 mM NaCl, 5 mM KCl, 5 mM MgCl$_2$, and 0.05% Tween-20) according to the manufacturer's recommendations. Plates were washed three times with SBT buffer and blocked with 100 µM biotin (Sigma-Aldrich B4501) in SBT buffer for 10 min at room temperature. Plates were washed three times at 100 RPM for 1 min with SBT buffer and blocked with 3% BSA in SBT buffer for 30 min at room temperature. Plates were washed three times at 100 RPM for 1 min with SBT buffer. Mouse plasma biological replicates were diluted 1:80 in SBT buffer (optimal plasma dilution range determined in-house) and incubated at 450 RPM for 2 h at 37 °C. SBT alone was used to determine background signal. Plates were then washed three times at 100 RPM for 1 min in TBST buffer (Tris-HCl pH 7.6, 150 mM NaCl, 0.05% Tween-20) and incubated at 450 RPM for 1 h at room temperature with 1 µg/mL αClec11a polyclonal antibody (R&D Systems AF3729) in 3% BSA in TBST buffer. Plates were washed three times at 100 RPM for 1 min in TBST buffer and incubated at 450 RPM for 30 min at room temperature with 8 ng/mL peroxidase-conjugated donkey-αgoat secondary antibody (Jackson ImmunoResearch 705-035-147) in 3% BSA in TBST buffer (optimal secondary antibody dilution range determined in-house). Plates were washed three times at 100 RPM for 1 min in TBST buffer and incubated in 1-Step Ultra TMB substrate (ThermoFisher Scientific 34028) for 12 min at room temperature. Sulfuric acid was added to a final concentration of 1 M to stop the reaction and absorbance was read at 450 nm.

Direct ELISAs were performed as follows. Mouse plasma was diluted 1:50 in PBS (pH 7.2) (optimal plasma dilution range determined in-house to achieve a maximum signal-to-noise ratio) and coated on Corning 1 × 8 Stripwell 96-well plates (Sigma-Aldrich CLS2592) and incubated overnight at 4 °C. PBS (pH 7.2) alone was used to determine background signal. Plates were washed three times in PBS (pH 7.2) with 0.01% Tween-20 and blocked for 2 h at room temperature with ELISA Blocker blocking buffer (ThermoFisher Scientific N502). Plates were washed three times in PBS (pH 7.2) with 0.01% Tween-20 and incubated for 2 h at room temperature with 1 µg/mL αClec11a polyclonal antibody (R&D Systems AF3729) in PBS (pH 7.2) with 0.01% Tween-20. Following primary antibody incubation, plates were washed three times in PBS (pH 7.2) with 0.01% Tween-20 and incubated for 1 h at room temperature with 8 ng/mL peroxidase-conjugated donkey-αgoat secondary antibody (Jackson ImmunoResearch 705-035-147) in PBS (pH 7.2) with 0.01% Tween-20 (optimal secondary antibody dilution range determined in-house). Plates were then washed three times in PBS (pH 7.2) with 0.01% Tween-20 and incubated in 1-Step Ultra TMB substrate (ThermoFisher Scientific 34028) for 12 min at room temperature. Sulfuric acid was added to a final concentration of 1 M to stop the reaction and absorbance was read at 450 nm. Recombinant murine Clec11a protein (R&D Systems 3729-SC/CF) was used to establish standard curves and determine protein concentration.

**Clec11a/SCGF infusion**. Recombinant murine Clec11a (SCGF) protein (R&D Systems 3729-SC/CF) was resuspended in PBS (pH 7.2) to 100 µg/mL and stored at −80 °C as single-use aliquots. For steady-state analysis, 100 µL of either 4 µg SCGF in PBS (pH 7.2) or PBS alone was injected subcutaneously on five consecutive days

prior to analysis. Total SCGF protein dosing in steady-state animals was adapted from a previous report[62]. For regeneration analysis, 100 µL of either 2 µg SCGF in PBS (pH 7.2) or PBS alone was injected subcutaneously on days 1, 3, 5, 7, 9, 11, and 13 post-TBI (650 rads). SCGF dosing following myelosuppressive injury was determined by a dose–response experiment (Supplementary Fig. 10).

**µCT analysis**. Femurs were isolated, fixed in 4% PFA for 24 h at 4 °C, and stored in 70% ethanol at 4 °C. A Scanco Medical µCT 35 system with an isotropic voxel size of 7 µm was used to image the distal femur. Scans used an X-ray tube potential of 55 kVp, an X-ray intensity of 0.145 mA, and an integration time of 600 ms. For trabecular bone analysis, an upper 2.1-mm region beginning 280 µm proximal to the growth plate was contoured. For cortical bone analysis, a region 0.6 mm in length centered on the mid-shaft was used. Trabecular and cortical bones were thresholded at 211 and 350 per mg HA/cm$^3$, respectively. 3D images were obtained from contoured 2D images by methods based on distance transformation of the binarized images.

**Statistics**. Sample sizes for phenotypic and functional analysis of mouse hematopoietic parameters were determined based on prior estimates of variance and effect sizes observed in previous experiments. Number of animals needed were calculated based on the ability to detect a two-fold change in the mean with 80% power, with the threshold for significance (α) set at 0.05. All experimental findings were confirmed in at least two independent cohorts of mice and the data presented in the manuscript represent pooled data from independent experiments. Statistical comparisons between two groups were performed using two-tailed Student's $t$-test. Multiple comparisons were made using a one-way ANOVA analysis with a Tukey's correction. Data are presented as the mean ± standard error of the mean (SEM), unless otherwise noted. Statistical significance is indicated as *($P < 0.05$), **($P < 0.01$), ***($P < 0.001$), and n.s. (not significant). Statistical analysis was performed using Prism 6 (GraphPad Software). HSC frequency and 95% confidence intervals were determined using ELDA software (http://bioinf.wehi.edu.au/software/elda/)[83].

**Reporting summary**. Further information on research design is available in the Nature Research Reporting Summary linked to this article.

## Data availability
The authors declare that data supporting the findings of this study are available within the manuscript and its Supplementary Information files. Source data for Figs. 1–9 and Supplementary Figs. 1–10 are provided with the paper as a Source Data File. Proteomics raw data are included in Supplementary Data 1. Proteomics Raw Data are included in the Source Data File.

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

## Acknowledgements

We would like to thank the Starr Foundation Tri-Institutional Core Facilities. Our work is supported by the Tri-Institutional Stem Cell Initiative (2014-004), American Society of Hematology Scholar Award, American Federation for Aging Research, National Institutes of Health (1R01CA204308 and 1R01HL133021), and the Leukemia and Lymphoma Society Quest for Cures. Jason M. Butler is a Scholar of The Leukemia and Lymphoma Society.

## Author contributions

P.R. designed the experiments; analyzed hematopoietic parameters including transplantations and myelosuppressive studies; performed proteomic and transcriptional profiling and analysis; performed direct ELISAs and immunoblots; analyzed all data and helped prepare the manuscript. E.L. and M.G.P. helped design the experiments; analyzed vascular leakiness and performed aptamer-based Sandwich ELISAs; helped with functional assays; helped analyze the data and prepare the manuscript. D.L. helped design the experiments; performed SCGF infusion studies; helped analyze hematopoietic parameters; helped analyze the data and prepare the manuscript. C.C.K. helped design the experiments and executed steady-state hematopoietic assays. M.J.C. analyzed SomaLogic proteomic analysis and conducted Ingenuity Pathway Analysis. M.C.G. and L.K. conducted immunofluorescence analysis, helped with functional assays and maintained and generated all animal models. M.B.G. conducted bone μCT analysis. M.C.G., L.K., A.G.F., and C.Y.P. helped with the functional assays. P.R. and J. M.B. designed the experiments, was responsible for the overall project strategy and management, helped execute all phenotypic and functional hematopoietic assays, and prepared the manuscript.

## Competing interests

The authors declare no competing interests.
