## [Peer Review File · Nature Communications]

Reviewers' comments:

Reviewer #1, expert in inflammatory signaling in hematopoiesis (Remarks to the Author):

In their manuscript, Ramalingam et al. describe the consequences of deregulated MAPK signaling in the maintenance of endothelial niche. In particular, they suggest that defective MAPK signaling leads to activation of the canonical NF-KB pathway mediated inflammatory signals and that these signals are responsible for defective HSC functions and hematopoiesis in their mouse model.

Overall, these studies are fairly well performed and there are some interesting observations reported in their manuscript. However, there are many serious (both conceptual and technical) concerns that need to be addressed. A few of them are highlighted below;

1. The novelty and significance of their findings are unclear. As pointed out by the authors in pages 3 & 4, NFkB and MAPK are well known candidates to control inflammatory responses in endothelial cells; a previous report (Poulos, Nature communications, 2016) from the same group established (through a loss of function approach) that Endothelial specific inhibition of NF-KB led to increased HSC functions due to a decrease in pro-inflammatory cytokines; and another report by the same group already demonstrated that endothelial MAPK activation drives myeloid-biased differentiation of HSCs, at the expense of self-renewal possibly due to inflammation. The current study essentially recapitulates all their previous findings, although with a slightly different approach and model system.

2. The authors repeatedly discuss about BM niche/endothelial niche and HSC maintenance/functions, but surprisingly they do not show any documentation on the actual "BM/HSC niche status" in the CDH5-MAPK mice. They should perform sections on the BM and perform microscopy to demonstrate that the vascular niche and HSC interactions are disturbed in their model.

3. The HSPC characterization in figure 1 is far from complete. It is surprising that they make large claims without providing the immunophenotype of the "true" HSCs. They show highly reduced HSCPs (LSK cells), but they have not provided data on distinct HSPC subsets; LT-HSC, ST-HSCs, MPP2, MPP3 & MPP4. Are there specific changes on these distinct subsets? are there changes in relative numbers vs. absolute numbers of these fractions. Without all these critical information, it is very difficult to make any claims on HSCs.

4. The western blot data shown in Figure 2 are not convincing. Have they performed any densitometry quantifications of these images? Is this statistically significant? Also, it is quite difficult to conceive that only a modest increase of phospho-ERK caused all these changes. Do the authors believe that ERK is the only MAPK target in their model, have they looked at other MAPK members?

5. Similarly, the phospho- p65 data are not convincing. As mentioned above, no quantification/statistical data were given. Moreover, a critical analysis of p65 blots suggests that the phosphorylation of p65 may not be increased in CDH5-MAPK mice. This is because the total levels of pan-p65 were increased in the CDH5-MAPK mice (or even total protein levels were increased, as tubulin levels were slightly higher). Moreover, the authors make a strong claim that increased (canonical) NFkB activation was observed in their model. However, all they show to justify their claims was a simple western blot on p65, which is questionable. To substantiate their claim that canonical NFkB is hyperactivated, the authors should really show Phospho-IkB and IkB degradation assays and p65 translocation to the nucleus. Again, without the statistical data and with all these indicated caveats, it is impossible to make proper sense of these presented data.

6. In figure 4a-d, the authors show data on ROS levels, quiescence and apoptosis. Again, all these experiments were performed with mixed bag of (LSK) cells. It is unclear if any of these inferences can be attributed to the "true" HSCs, especially in view of the fact that LT-HSCs are only ~10 % of LSK. Moreover, it is of a major concern that they show that ~ 90% of the LSK cells of control and IkB mice are in G0 phase. This is totally unacceptable and certainly there are some obvious technical issues

associated with this analysis. According to the current understanding on HSC biology, on an average only 20-30% of the LSK cells will remain quiescent under steady state conditions (Wilson, Cell, 2008). Among the LSK cells the true quiescent LT-HSCs (lin⁻Sca1+cKit+CD150+CD48⁻CD34⁻) are the only fraction that remains quiescent (~ 70%), but this fraction is < 5 % of the LSK. So, their data on quiescence cannot be considered.

7. The manuscript lacks any molecular insights. While they do provide many isolated observations on hematopoiesis, NFkB signals, hypoxia and the role of SCGF in the model. However, no attempt was made to connect all these observations and make sense of them. In particular, they talk about hypoxia but they have not provided insights on the role of hypoxia (if any) in the HSC phenotype. Does hypoxia affect any physiological properties of HSCs, in other words what are the functional consequences of hypoxia? Moreover, what are the actual or proposed mechanisms through which SCGF influences the phenotype?

8. Along the lines mentioned above, the gating strategy for CMPs, GMPs and MEPs are inappropriate. The field has convincingly shown by many groups and different technologies, that true common myeloid progenitors cannot be sorted accurately by Akashi et al. 2000 Nature. The described CMP population is a mixture of pre-GMPs and pre-MEPs that are already restricted to GM or MekE. The authors should adopt the accepted immunophenotyping strategy (Pronk et al. Cell Stem Cell 2007 and Rieger et al. 2008 Brit J Haematol.).

9. The authors performed their studies based on VE-Cadherin-Cre and Tie2-Cre mouse strains. Even though cre expression in these strains is largely specific to endothelial lineage cells, there are reports that both these strains may have "leakiness" in the hematopoietic lineage cells. It is unclear if the authors have taken these observations into consideration while interpreting their data. Are the authors convinced that the Cre deleters used in their studies are perfect? Did they cross their cre lines with Rosa26 reporter mice and study the deletion efficiencies in endothelial vs. hematopoietic lineages. In the absence of such data it is unclear if their phenotype is exclusively caused by HSC extrinsic mechanisms. They should have performed reciprocal BMT (injecting WT bone marrow cells into CDH5 MAPK mice) studies to prove that the phenotype is exclusively mediated by endothelial cells.

Reviewer #2, expert in HSC niche biology (Remarks to the Author):

This is an interesting report from Ramalingam et al. in which the authors show that activation of MAPK signaling in BM ECs induces inflammatory pathway activation in the hematopoietic compartment and suppression of NFkB signaling in endothelial cells restores niche integrity and hematopoietic function. The authors further identify stem cell growth factor alpha as a soluble factor that suppresses BM inflammation and restores hematopoietic function. The manuscript includes some interesting findings that would be of interest to the field, but I have some concerns regarding the experimental approach and conclusions that, if addressed, can improve the manuscript:

1. The authors utilized tamoxifen - impregnated feed to induce MAPK activation in endothelial cells and then describe the effects of this intervention on bone marrow cellularity, etc. Did the authors verify the increase in MAPK signaling in VEcadherin+ ECs? This should be shown.

2. Figure 1. In the competitive repopulation assay, the cell doses in the donor and competitor grafts should be stated in the main text or figure legend. Was this a 1:1 dilution? Were secondary transplants performed to assess long-term HSC defects?

In Figure 1e, the data appear to show the contribution of donor cells within each lineage population,

as opposed to the absolute percentages of donor+myeloid cells, donor+ T cells, donor+ B cells, etc. in recipient mice. Is the myeloid biased engraftment also evident when the absolute percentages of donor lineage+ cells are compared? Also, if the CDH5-MAPK mouse produces myeloid – biased engraftment after transplantation, why don't the CFCs from the BM of CDH5-MAPK mice (Figure 1c) show increased myeloid progenitor cells compared to littermate controls?

Figure 1g. A p value of 0.0479 would appear to approximate $p = 0.05$ or non-significance. Does this mean that the difference in the limiting dilution analysis is relatively weak or requires more replicates to confirm a statistically significant difference? Usually, limiting dilution assays to estimate HSC frequency are evaluated using Poisson Statistical Analysis. Was this approach utilized here?

Figure 1k. The immunofluorescent images of the BM vasculature suggest an impressive difference in response to enforced MAPK signaling in ECs. What is the time point of analysis of the mice following MAPK induction in ECs? Is this change in BM vascular appearance persistent over time or does it resolve or worsen over time?

3. Figure 2. In Figure 2d, the authors describe crossing CDH5-MAPK mice with Tie2.IkB-SS mice to generate CDH5-MAPK::IkB mice to evaluate whether suppression of endothelial NF-kB signaling downstream of endothelial MAPK activation could temporize BM inflammation observed in CDH5-MAPK mice. Tie2 is expressed in the majority of hematopoietic cell lineages (Genesis 2010;48:563-7) so the Tie2.IkB-SS mice would suppress NF-kB signaling in Tie2+ hematopoietic cells and endothelial cells. How does this cross allow for specific conclusions to be drawn about suppression of endothelial-specific NF-kB signaling suppression?

4. Figure 3. As stated above, the authors should explain how the CDH5-MAPK mice show decreased myeloid colony formation, whereas CDH5-MAPK::IkB mice show restoration of myeloid colony formation (Figure 3c); following competitive transplantation, the opposite effect is observed, with increased myeloid repopulation in the recipients of CDH5-MAPK donor cells and restoration of normal lineage distribution in recipients of CDH5-MAPK::IkB mice (Figure 3e). These results appear to be at odds with each other.

Figure 3i. The differences in hematologic parameters shown in this panel are interesting, but it would be helpful to know whether neutrophil counts are also significantly different in the CDH5-MAPK mice versus controls and corrected with CDH5-MAPK::IkB mice. This would be much more relevant than RBC counts which are largely irrelevant clinically.

My larger concern, as stated above, relates to the Tie2 IkB mouse model and whether it can allow discrimination of effects on endothelial cells versus hematopoietic cell effects. Would it be possible to validate the findings from the CDH5-MAPK::IkB mice using an endothelial – specific model in which NFkB signaling is suppressed?

5. Figure 4. Could the absence of differences in ROS levels in BM ECs be a function of the possibility that Tie2 – driven suppression of NFkB signaling is primarily occurring in hematopoietic cells, rather than in the vascular compartment in these mice?

In Figure 4, the authors transition into analysis of BM stromal cells for evidence of alteration in ROS levels, cell cycling and apoptosis. It is unclear or not well justified or explained why the authors chose to focus additionally on BM stromal cells as well as endothelial cells and HSPCs. Do the authors postulate that suppression of NFkB signaling in ECs alters cross-talk between ECs and stromal cells such that BM stromal cell function is altered? Do the authors have any insight mechanistically as to how this communication between ECs and stromal cells is occurring?

In Figure 4n-p, the authors show that expression of Il1b and Csf1 is decreased in CD45+ cells, but only Il1b is decreased in BM ECs and stromal cells. If Il1b and Csf1 are both involved in mediating the

effects of MAPK activation in BM ECs on hematopoiesis, how do the authors explain these differences?

6. Figure 5. The authors show that delivery of SCGF reverses the hematopoietic and BM vascular defects in CDH5-MAPK mice by suppression of BM inflammation driven by MAPK activation in BM ECs. Are the differences in gene expression of target genes shown in Figure 5i significant? Were these analyses repeated more than once to validate the original results? Significant differences should be indicated clearly in the Figure.

It is unclear mechanistically exactly how SCGF suppresses BM inflammation driven by BM EC MAPK activation. How does increased NF κ B signaling in ECs downregulate SCGF expression in BM stromal cells since BM stromal cells are the primary source of SCGF in the BM (extended Figure 7)? Does SCGF act directly on HSPCs and also directly on BM ECs and/or stromal cells or do indirect effects occur on these populations in response to SCGF infusion?

7. Figure 6. Interesting results are shown to indicate that infusion of SCGF improves hematologic recovery following irradiation. Did SCGF accelerate recovery neutrophils in irradiated mice? This would be the most clinically interesting result, since neutrophil counts dictate need for hospitalization and risk of bacterial infections.

If SCGF infusions do not increase numbers of phenotypic HSPCs in wild type mice following irradiation (6d), how do the authors explain the significant enhancement in recovery of cells capable of competitive repopulation as shown in Figure 6e?

Minor Concern

Page 14, bottom paragraph: Several other publications have characterized the effects of myelosuppressive injury on the BM vasculature and BM vascular function beyond Hooper et al., and these references should be appropriately cited.

Reviewer #3, expert in HSC repopulation (Remarks to the Author):

Endothelial MAPK activation disrupts hematopoiesis by including NF κ B dependent inflammatory stress by Ramalingam et al.

The role of chronic inflammatory signals in the bone marrow niche for hematopoiesis is still controversially discussed. The authors demonstrate that constitutive expression of MAPK in endothelial cells perturbs hematopoiesis. This correlates with an overall inflammatory signature in stroma as well as hematopoietic cells (NF κ B) and inflammatory signatures. Genetic inhibition of NF κ B signaling restores normal function in the model system. Mechanistically, that also restores hypoxia in the bone marrow to normal levels of hypoxia, but not ROS, as ROS is not induced by elevated MAPK in endothelial cells. The authors further identify SCGF as a target cytokine that mediates the negative effect of MAPK activation as well as myelosuppressive treatment in endothelial cells on hematopoiesis. This is an overall interesting study with a number of very novel findings including mechanisms that will contribute to the field.

There are though a couple of concerns that should/need to be addressed:

Major:

The authors need to confirm the level of activation of MAPK in endothelial cells in their animal model system. That information is not provided. All cells? Level of activation etc.?

The Tie2 promoter can be leaky and also be expressed in HSCs/hematopoietic cells.

The authors need to test the level of expression of the transgene in hematopoietic cells and especially HSCs. It looks like that has not been tested yet, also not in the previous publications with that mouse. That needs to be provided, as worst case, the data would require a strong re-interpretation of the

results.

The link between activation of MAPK in endothelial cells and the overall strong inflammatory signature in the whole bone marrow, including hematopoietic cells, is not determined to a great extent. There should be more emphasis on that point, especially as 2d, e and g show very similar overall activation patterns.

Minor:

Hypoxia in bone marrow is notoriously difficult to determine. The authors need to provide gating information etc. on these analyses, especially as the data are central for the conclusions.

The hematopoietic phenotype is complex, and due to the overall broad changes in the bone marrow, multiple cells type might be the target of the changes and contribute to the observed changes. For example, a 5 day treatment that results in such a large overall changes in hematopoiesis needs to target a broad range of distinct type of hematopoietic cells. That should be addressed/discussed in more detail.

Page11: The data does not show that the defects in hematopoiesis are a direct effect of inflammation-induced alterations in ROS and hypoxia. That is a correlation.

NCOMMS-19-11331-T: Endothelial MAPK activation disrupts hematopoiesis by inducing NF-kB-dependent inflammatory stress

Reviewer #1

General Comments: *In their manuscript, Ramalingam et al. describe the consequences of deregulated MAPK signaling in the maintenance of endothelial niche. In particular, they suggest that defective MAPK signaling leads to activation of the canonical NF-KB pathway mediated inflammatory signals and that these signals are responsible for defective HSC functions and hematopoiesis in their mouse model. Overall, these studies are fairly well performed and there are some interesting observations reported in their manuscript.*

Response: We would like to thank the reviewer for critically reviewing our manuscript and for their insightful comments. We have now included new and exciting data providing more mechanistic insights, comprehensively addressed all the technical concerns raised and have clarified many points that strengthens the overall conclusions of the manuscript.

1. The novelty and significance of their findings are unclear. As pointed out by the authors in pages 3 & 4, NFkB and MAPK are well known candidates to control inflammatory responses in endothelial cells; a previous report (Poulos, Nature communications, 2016) from the same group established (through a loss of function approach) that Endothelial specific inhibition of NF-KB led to increased HSC functions due to a decrease in pro-inflammatory cytokines; and another report by the same group already demonstrated that endothelial MAPK activation drives myeloid-biased differentiation of HSCs, at the expense of self-renewal possibly due to inflammation. The current study essentially recapitulates all their previous findings, although with a slightly different approach and model system.

Response: Yes, it is true that NFkB and p38 MAPK are known candidates that can promote inflammatory responses; however, the role of ERK1/2 MAPK pathway and its cross-talk with NFkB during chronic inflammation have not been examined to date. More importantly, the signaling pathways mediating chronic niche-driven inflammation and their impact on *in vivo* niche-HSC interactions, remain unexplored due to the lack of appropriate genetic models. Our *CDH5-MAPK* mouse model represents a significant advance over Kobayashi et al. Nature Cell Biology 2010: PMID: 20972423 which utilized an *in vitro* co-culture system to show that endothelial MAPK activation impacts HSPC function *ex vivo*, thereby laying the groundwork and justifying our approach in the present study to test this hypothesis *in vivo*. The *CDH5-MAPK* mouse model has several advantages for studying the impact of chronic vascular inflammation on HSC function and hematopoiesis. It is well established that chronic inflammation is a systemic process affecting the vasculature in all organs. *CDH5-MAPK* manifest both systemic as well as BM localized inflammation making it physiologically relevant for studying chronic inflammation. Most of these changes including increased vascular leakiness, increased ROS levels and hypoxia have been shown to impact HSC function and are known to be directly regulated by inflammation. In short, the *CDH5-MAPK* mouse model is able to faithfully recapitulate a diverse array of hematopoietic and vascular changes observed during chronic inflammation making it a relevant model for studying the impact of chronic inflammation on HSC-niche interactions during myelosuppressive injuries, aging and leukemias and ***will serve as a useful tool for the scientific community.***

As regards to the previous publication mentioned (Poulos, Nature communications, 2016), the findings from that study demonstrated that suppression of basal NF-kB signaling during *steady state homeostasis* is sufficient to enhance HSC function. Contrarily, the *CDH5-MAPK* mice in the present study represent a mouse model of *chronic inflammation and stress hematopoiesis*. In the present study, we utilize the *Tie2::IkB.SS* mice to evaluate the interactions between ERK and NF-kB signaling pathways and confirm that suppression of endothelial NF-kB signaling is able to genetically override endothelial

MAPK activation. Although interactions of p38 MAPK and NF- κ B signaling have been well studied, this is the *first study to directly demonstrate genetic interactions of ERK and NF- κ B signaling pathways* within the BM vascular niche. Importantly, the complete rescue provided by endothelial NF- κ B inhibition makes it an even more valuable tool for the field to study niche biology in the context of chronic inflammation. ***Most importantly, the identification of SCGF as a novel factor that enhances vascular and hematopoietic regeneration from the models used in this study attests to their strength and also adds a significant advancement from our previous findings making this study novel and impactful.*** We have included these observations on the novelty and impact of the study in the relevant sections of manuscript.

2. The authors repeatedly discuss about BM niche/endothelial niche and HSC maintenance/functions, but surprisingly they do not show any documentation on the actual “BM/HSC niche status” in the CDH5-MAPK mice. They should perform sections on the BM and perform microscopy to demonstrate that the vascular niche and HSC interactions are disturbed in their model.

Response: We have updated our previous Figure 3 (**New Figure 4**) to include whole mount immunofluorescence images to demonstrate vascular niche-HSC interactions. We have determined that CD150+ (white cells) CD48-Lineage- (blue cells not shown for better resolution of HSC interactions with the vasculature) HSCs are farther away from the BM vasculature in *CDH5-MAPK* mice and that simultaneous endothelial-specific NF- κ B inhibition restores the proximity of HSCs to the BM endothelial niche. These new findings demonstrate that endothelial MAPK activation disrupts HSC-niche interactions that are resolved upon simultaneous inhibition of endothelial NF- κ B signaling.

3. The HSPC characterization in figure 1 is far from complete. It is surprising that they make large claims without providing the immunophenotype of the “true” HSCs. They show highly reduced HSCPs (LSK cells), but they have not provided data on distinct HSPC subsets; LT-HSC, ST-HSCs, MPP2, MPP3 & MPP4. Are there specific changes on these distinct subsets? are there changes in relative numbers vs. absolute numbers of these fractions. Without all these critical information, it is very difficult to make any claims on HSCs.

Response: We respectfully disagree with the reviewer. Although immunophenotyping by Flow cytometry is certainly useful, any claims on ‘true HSC’ can only be inferred from gold-standard functional assays such as limiting dilution and competitive repopulation transplantation assays, which we had included in the original manuscript. Phenotypic frequency of HSCs, MPPs etc. estimated by Flow cytometry does not provide any information regarding the functional potential of the HSC. For instance, the seminal paper by Rossi et al., (PMID 15967997) demonstrated that there is ~4-fold increase of phenotypic HSC frequency in aged mice as compared to young mice. However, transplantation of aged HSCs results in decreased engraftment, as compared to young HSCs indicating a *decline of functional potential* of aged HSCs. This has also been shown to be relevant during inflammation-induced myelosuppression wherein HSC numbers are not significantly altered phenotypically but display defects in their functional potential (Zhang H et al., PMID 27264973). Hence, the only way to assess HSC function is by evaluating long-term engraftment potential of the HSC. Since all of our conclusions in the original version of the manuscript were based on functional transplantation assays, our claims on HSCs in this study remain absolutely valid (New Figures 1F-I, 4D-G, 8D-G and 9G-J).

Nonetheless, we would like to clarify that the data in original Figure 1B represented the frequency of immunophenotypically defined HSCs (cKIT⁺Lineage^{Neg} SCA1⁺ CD150⁺CD48^{Neg}) and not LSK cells, as we had clearly described and defined in the original manuscript text pertaining to Figure 1B. Given that all immunophenotypes used to enrich for “true” HSCs never give rise to 100% long-term, multilineage engraftment, we believe they are better served as being labeled as ‘HSPC’ since this is what they actually are based on functional assays. However, we acknowledge the reviewer’s point regarding nomenclature, as it raises the possibility that our HSC data might be misinterpreted as ‘KLS cells’ by the readers. To address this, we have updated all references in the revised manuscript to explicitly state ‘HSCs’ in all instances where cKIT⁺Lineage^{Neg} SCA1⁺ CD150⁺CD48^{Neg} SLAM HSCs were quantified. We have also

included representative contour plots (New Figure 1C) to demonstrate the flow cytometry gating strategy for analyzing HSC and HSPC subsets.

Regarding the reviewers' concern about immunophenotyping of HSPC subsets, we would like to point out that our previous Extended Figure 2 (New Figure 5) demonstrated a comprehensive analysis of hematopoietic precursors (MPP, CMP, CLP, GMP, MEP and B cell progenitors) in the BM and spleen from all four cohorts (control, *CDH5-MAPK*, *IκB*, and *CDH5-MAPK::IκB* mice), as well as complete analysis of the peripheral blood (blood counts, lineage distribution and HSPC frequency) and show that *CDH5-MAPK* mice manifested defects in all of these populations and that *CDH5-MAPK::IκB* mice restored all these defects. However, to address the Reviewer's concerns, we have now included the frequencies as well as absolute numbers of phenotypic HSCs as well as HSPC subsets including KLS cells, MPPs, HPC-1 and HPC-2 all of which demonstrate a striking reduction of both frequencies as well as absolute numbers in *CDH5-MAPK* mice (New Figure 1D; please see above, and New Supp. Figure 1A). Notably, these HSPC defects are resolved upon endothelial NF-κB inhibition (New Supp. Figure

2A) or SCGF infusion (**New Supp. Figure 8C**). It's important to note that the immunophenotypes were based on published data (i.e. PMID 23434755). Taken together, the immunophenotypic data and the functional transplantations demonstrate that activation of MAPK specifically in endothelial cells have deleterious effects to HSCs and are resolved upon endothelial NF-kB inhibition or SCGF infusion.

4/5. The western blot data shown in Figure 2 are not convincing. Have they performed any densitometry quantifications of these images? Is this statistically significant? Also, it is quite difficult to conceive that only a modest increase of phospho-ERK caused all these changes. Do the authors believe that ERK is the only MAPK target in their model, have they looked at other MAPK members?

Similarly, the phospho-p65 data are not convincing. As mentioned above, no quantification/statistical data were given. Moreover, a critical analysis of p65 blots suggests that the phosphorylation of p65 may not be increased in CDH5-MAPK mice. This is because the total levels of pan-p65 were increased in the CDH5-MAPK mice (or even total protein levels were increased, as tubulin levels were slightly higher). Moreover, the authors make a strong claim that increased (canonical) NFkB activation was observed in their model. However, all they show to justify their claims was a simple western blot on p65, which is questionable. To substantiate their claim that canonical NFkB is hyperactivated, the authors should really show Phospho-IkB and IkB degradation assays and p65 translocation to the nucleus. Again, without the statistical data and with all these indicated caveats, it is impossible to make proper sense of these presented data.

Response: The Reviewer is absolutely correct in their observation that p65 phosphorylation changes are not striking (**quantification of the blot in Original Figure 2C shown below in Reviewer Figure 1a**).

This is an important observation which highlights the differential magnitude of phosphorylation changes observed during acute versus sustained inflammation. Acute stimulation with TNF α or LPS (~15-30 minutes) results in a profound increase in S536 phosphorylation along with degradation of IkB α . However, sustained stimulation with pro-inflammatory cytokines results in activation of endogenous feedback mechanisms that downregulate p65 phosphorylation and increase IkB α synthesis, as demonstrated previously (Pradère J-P et al., PMID: 27555662, Yang F et al., PMID 12759443, Brown K et al., PMID 8460169, and Wu C et al., PMID 12791687). The CDH5-MAPK model represents a constitutive MEK1DD driven ERK1/2 phosphorylation leading to sustained activation of NF-kB signaling. Hence the magnitude of p65 phosphorylation changes in this scenario would be expected to be more reflective of chronic stimulation of NF-kB signaling. We have included new data (**New Figure 2**) that confirms the modest but consistent increase in p65 phosphorylation in BMECs derived from CDH5-MAPK mice (n=3 biological replicates/cohort per experiment, n=2 independent experiments). Total IkB α

levels were not altered and we did not observe any detectable phosphorylation of I κ B α . These features are consistent with chronic activation of NF- κ B signaling (Pradère J-P et al., PMID: 27555662, Yang F et al., PMID 12759443, Brown K et al., PMID 8460169, and Wu C et al., PMID 12791687). Importantly, as per the reviewer's suggestion, we confirmed activation of NF- κ B signaling in *CDH5-MAPK* BMECs by evaluating nuclear translocation of p65, the gold-standard assay for NF- κ B activation (Wessel A et al., PMID 25736764). Quantification of nuclear p65 levels confirmed that *CDH5-MAPK* BMECs display increased nuclear p65 translocation. Collectively, these findings demonstrate that nuclear p65

New Figure 2

New Figure 2. *CDH5-MAPK* mice display systemic and BM-localized inflammation. **g, h)**

Immunoblot analysis of BMECs isolated from *CDH5-MAPK* mice demonstrating that MEK1DD expression in BMECs results in an increase in ERK1/2 and p65 phosphorylation. **i, j)** Representative immunofluorescence images and quantification demonstrating increased levels of nuclear p65 in BMECs derived from *CDH5-MAPK* mice as compared to controls. Control BMECs treated with TNF α (10 ng/mL for 15 minutes) were used as a positive control for the assay. Each dot within the bar graph represents nuclear p65 staining intensity per individual cell. Error bars represent sample mean \pm SEM. Statistical significance was determined using two-tailed unpaired Student's t-test. * $P < 0.05$; ** $P < 0.01$; *** $P < 0.001$.

translocation is a better assay to evaluate NF- κ B activation in the context of chronic inflammation as compared to S536 phosphorylation of p65. We believe these findings represent important observations for the field studying chronic inflammation and have included these new observations in the manuscript.

Also, we do not observe any changes in p38 phosphorylation confirming the specificity of MEK1DD dependent ERK1/2 phosphorylation (**Reviewer Only Figure 1b**).

6. In figure 4a-d, the authors show data on ROS levels, quiescence and apoptosis. Again, all these experiments were performed with mixed bag of (LSK) cells. It is unclear if any of these inferences can be attributed to the "true" HSCs, especially in view of the fact that LT-HSCs are only ~10 % of LSK. Moreover, it is of a major concern that they show that ~ 90% of the LSK cells of control and I κ B mice are in G0 phase. This is totally unacceptable and certainly there are some obvious technical issues associated

with this analysis. According to the current understanding on HSC biology, on an average only 20-30% of the LSK cells will remain quiescent under steady state conditions (Wilson, Cell, 2008). Among the LSK cells the true quiescent LT-HSCs (lin-Sca1+cKit+CD150+CD48-CD34-) are the only fraction that remains quiescent (~ 70%), but this fraction is < 5 % of the LSK. So, their data on quiescence cannot be considered.

Response: We did not encounter any technical issues and our flow cytometry-based analysis is performed using appropriate single-stained controls for compensation and concentration matched isotype controls and FMO controls for establishing gates. However, in light of the concerns raised by the reviewer, we have confirmed our original analysis of the cell-cycle experiments using LSK cells to validate our observations and the data confirms that endothelial MAPK activation causes increased cycling of LSK cells and are restored upon simultaneous inhibition of endothelial NF-kB (**New Figure 7C**). Moreover, to clarify flow cytometry gating strategy for readers and reviewers alike, we have now included representative contour plots to demonstrate gating strategy utilized for all of our technical analyses throughout the manuscript including cell-cycle, apoptosis, ROS and hypoxia along with gating strategy for analysis of various BM cellular subsets including HSCs, hematopoietic stem and progenitors including KLS cells, MPPs, HPC-1, HPC-2, CLP, CMP, GMP, MEP & B cell progenitors, BM niche cells including BMECs, BM stromal cells, Lepr+ cells and osteoblasts as well as lineage composition analysis of hematopoietic cells.

Although LSK are a mixed population, they are still highly enriched for HSC activity (1 in 6 to 1 in 10, respectively) and usually changes in LSK cell-cycle status are reflected within the HSC population as well, although the converse might not always hold true. Nonetheless, to confirm that in our experimental model system whether the ‘true HSCs’ are indeed undergoing increased cycling and apoptosis, we performed cell-cycle analysis on HSCs (**New Figure 1 and New Supplementary Figure 1**) which demonstrated that all sub-populations within the LSK fraction including HSCs, MPPs, HPC-1 and HPC-2 manifest a loss of quiescence along with increased apoptosis in *CDH5-MAPK* mice. Notably, the % of cells in G0 phase in control mice (~90% of HSCs and MPPs, and ~50% of KLS, HPC-1 and HPC-2) is in agreement with established parameters of cell-cycle based on SLAM markers (Oguro et al., 2013 PMID 23827712) validating our analysis.

7a. The manuscript lacks any molecular insights. While they do provide many isolated observations on hematopoiesis, NFkB signals, hypoxia and the role of SCGF in the model. However, no attempt was made to connect all these observations and make sense of them.

Response: We strongly disagree with the reviewer. In our original submission, we utilized our model systems and demonstrated that endothelial MAPK activation results in significant defects in the functional output of HSCs. These phenotypes are, in part, driven by vascular leakiness and local bone marrow and systemic inflammation and a decrease in SCGF. Simultaneous inhibition of endothelial NF-kB signaling completely reverses all of the phenotypes and restores SCGF expression. Additionally, exogenous infusion of SCGF restores the hematopoietic defects, much like endothelial NF-kB inhibition, by suppressing bone marrow inflammation, restoring vascular integrity and resolving the phenotypic numbers of SLAM HSCs and their functional output based on transplant assays. We have now included new data which confirms that expression of I κ B α -super suppressor in BMECs derived from *CDH5-MAPK* mice is able to block nuclear p65 translocation (**New Figure 3**). More importantly, we demonstrate that treatment of *CDH5-MAPK* BMECs with SCGF significantly reduces their nuclear p65 levels (**New Figure 8**).

These data come full circle and connect the dots. Collectively, our findings demonstrate that chronic endothelial inflammation leads to loss of vascular integrity, increased expression of pro-inflammatory cytokines and alterations of hypoxia and ROS levels within the BM which adversely impact HSC activity. Suppression of endothelial inflammation by either inhibition of NF-kB signaling or infusion of SCGF

resolves BM inflammation, restores vascular integrity and HSC function. The schematic in **New Figure 10** summarize these observations. This new data, coupled with our previous rigorous functional data, uncovers a new role for SCGF in directly regulating NF- κ B dependent endothelial inflammation.

7b. In particular, they talk about hypoxia but they have not provided insights on the role of hypoxia (if any) in the HSC phenotype. Does hypoxia affect any physiological properties of HSCs, in other words what are the functional consequences of hypoxia? Moreover, what are the actual or proposed mechanisms through which SCGF influences the phenotype?

Response: It is well established that hypoxic conditions increases ROS production. In fact, HSCs that are high in ROS levels have defects in their long-term repopulation and survival (Tsvee Lapidot's Laboratory, Nature, PMID: 27074509). We demonstrate for the first time that endothelial MAPK activation can promote these deleterious intrinsic defects in HSPCs that not only includes increased hypoxia and ROS production (detrimental for HSPCs and progenitors; see Lapidot paper), but also involves an increase in proliferation and apoptosis. Since endothelial NF- κ B inhibition rescues all of these phenotypes and also rescues the long-term repopulation activity of the HSCs, our data demonstrates that the increase in hypoxia is due to endothelial NF- κ B dependent inflammatory stress within the BM. As mentioned in the Response to **7a**, exogenous infusion of SCGF suppressed BM inflammation, restored vascular integrity and resolved the phenotypic numbers of SLAM HSCs and their functional output based on gold standard transplant assay. Our new data (**New Figure 8**) suggests that these effects are mediated by the effect of

SCGF on endothelial inflammation by decreasing nuclear p65 translocation. We have clarified the role of hypoxia in our model system within the manuscript.

8. Along the lines mentioned above, the gating strategy for CMPs, GMPs and MEPs are inappropriate. The field has convincingly shown by many groups and different technologies, that true common myeloid progenitors cannot be sorted accurately by Akashi et al. 2000 Nature. The described CMP population is a mixture of pre-GMPs and pre-MEPs that are already restricted to GM or MekE. The authors should adopt the accepted immunophenotyping strategy (Pronk et al. Cell Stem Cell 2007 and Rieger et al. 2008 Brit J Haematol.).

Response: Our antibody panels and gating strategies were based on published data from Sean Morrison's group (PMID 23434755), Irv Weissman's group (PMID: 20890962), Camilla Forsberg's group (PMID 21726834) and Emmanuelle Passegue's group (PMID: 28355185). These are leaders in the field and have systematically described all of these populations and cell surface markers utilized in this manuscript. With the utmost respect to the reviewer, to call our approach inappropriate or an unaccepted strategy is in of itself inappropriate. With this said, we do not take anything away from the gating strategies developed and utilized by David Bryder and Timm Schroeder (leaders in the field as well), as their studies added important information to the complexity of the cell surface markers of stem and downstream progenitors and have been adopted and further expanded by preeminent scientists such as Andreas Trumpp.

We do agree with the reviewer that with advent of single cell technologies, the immunophenotyping strategies are bound to become more refined and yield better enrichment for hematopoietic subsets. However, as mentioned previously in our response to Reviewers' comment 3 regarding HSC immunophenotyping, we again reiterate the importance of functional assays. In our original manuscript, we demonstrated that the decreased frequency of immunophenotypically defined hematopoietic progenitors in *CDH5-MAPK* mice are manifested as an overall decrease in functional progenitor activity of erythroid (BFU-E), myeloid (CFU-G, CFU-M) and megakaryocytic (CFU-GEMM) lineages using methylcellulose-based colony forming assays (New Figure 1E). Furthermore, suppression of endothelial NF- κ B signaling (New Figure 4C) or infusion of SCGF (New Figure 8C) is able to significantly restore colony forming ability in *CDH5-MAPK* mice demonstrating a functional restoration of progenitor activity. These data robustly demonstrate the progenitor defects in *CDH5-MAPK* mice.

9. The authors performed their studies based on VE-Cadherin-Cre and Tie2-Cre mouse strains. Even though cre expression in these strains is largely specific to endothelial lineage cells, there are reports that both these strains may have "leakiness" in the hematopoietic lineage cells. It is unclear if the authors have taken these observations into consideration while interpreting their data. Are the authors convinced that the Cre deleters used in their studies are perfect? Did they cross their cre lines with Rosa26 reporter mice and study the deletion efficiencies in endothelial vs. hematopoietic lineages. In the absence of such data it is unclear if their phenotype is exclusively caused by HSC extrinsic mechanisms. The should have performed reciprocal BMT (injecting WT bone marrow cells into CDH5 MAPK mice) studies to prove that the phenotype is exclusively mediated by endothelial cells.

Response: There are two major clarifications that will help address these concerns:

1. We do not use a Tie2-Cre deleter system. Our I κ B mouse model is a transgenic model wherein a Tie2 promoter/enhancer element drives the expression of the I κ Ba Super Suppressor transgene (**New Figure 6**). Although Tie2 is expressed in hematopoietic cells during embryonic development (observed in Tie2-cre mouse models), its expression is restricted to the endothelium in adult mice as demonstrated by Tie2-cre/ERT2 models (Reviewed in Pyane S et al., PMID 30354251). Utilizing *Tie2-Cre; Rosa26REYFP*, Tang et al demonstrated that Tie2 promoter is no longer active in hematopoietic cells beyond E12.5 (Tang Y et al., PMID 20645309). The endothelial-restricted expression of Tie2 driven transgenes in adult

C57BL6 mice has also been confirmed by several independent groups (Forde A et al., PMID 12203917, Gareus R et al., PMID 19046569, Korhonen H et al., PMID 19171764). With regard to Tie2 expression in HSCs of adult mice, in our Nature Communications manuscript (Poulos et al., PMID: 28000664) we confirmed that Tie2 was not expressed on phenotypic HSCs of adult C57BL6 mice. More importantly, we demonstrated that HSCs from *Tie2::IkB.SS* mice, unlike their BMEC counterparts, responded to TNF α stimulation confirming the absence of transgene expression within HSCs of adult *Tie2.IkB-SS* mice. We also confirmed these findings by performing reciprocal transplantations showing that enhanced HSC activity of *Tie2.IkB-SS* mice was exclusively mediated by inhibition of endothelial NF- κ B signaling. Collectively, these observations and experiments clearly demonstrate that the *IkB-SS* transgene in *Tie2::IkB-SS* mice is not expressed in HSCs and does not interfere with the activation of canonical NF- κ B signaling in HSCs. Nonetheless, in the revised manuscript we confirm that the decreased expression of NF- κ B target genes within hematopoietic cells of *CDH5-MAPK* mice upon crossing to *Tie2.IkB-SS* mice (New Supp. Fig. 3a) is not due to leaky expression of the *IkB-SS* transgene within hematopoietic cells by performing RT PCR analysis and detecting the expression of *IkB-SS* transgene from FACS sorted BMECs and CD45⁺ hematopoietic cells. Our analysis confirmed that the *IkB-SS* transgene was only expressed in BMECs and not in CD45⁺ cells (**New Figure 6**).

2. The *Cdh5(PAC)-creERT2* system utilized in this study was developed by Ralf Adams and expresses a tamoxifen inducible CreERT2 transgene specifically in endothelial cells of adult mice (Reviewed in Pyane S et al., PMID 30354251). Using a Rosa26-mT/mG reporter line (as suggested by the Reviewer), Adam's group have previously demonstrated that transgene expression in adult mice is exclusive to endothelial cells and not expressed in hematopoietic cells (Langen UH et al., PMID 28218908). Kilani et al recently utilized the *Cdh5(PAC)-creERT2:mT/mG* reporter system and confirmed that CreERT2 expression is restricted to endothelial cells of adult mice and not detected in hematopoietic cells (Kilani et

al., PMID 30801958). Furthermore, they use an innovative JAK2V617F mouse model to detect CreERT2 expression in rare subsets of adult HSCs and show that *Cdh5(PAC)-creERT2* is not expressed in HSCs of adult mice. Nonetheless, we performed additional experiments to confirm that *creERT2* in our *CDH5-MAPK* mouse model is restricted to endothelial cells (**New Figure 6**). The MAPK mouse model utilized in this study utilizes a mutant form of MAPKK1 rendered constitutively active by two serine->aspartic acid substitutions (S218D/S222D) within the catalytic domain. This followed by an IRES-GFP that enables tracking of *creERT2* expression *in vivo* by flow cytometry. Utilizing the GFP reporter system, we observed that GFP+ cells represented a very small fraction of the WBM cells (~0.024%) and were exclusively detected in the CD45-Ter119- non-hematopoietic fraction. Furthermore, virtually all of the

GFP+ cells demonstrated surface expression of endothelial markers (VEcadherin+CD31+ CD45-/Ter119-). We also confirmed *in vivo* activation of phospho-ERK1/2 and show that GFP expression correlates with increased phospho ERK1/2 expression. We observed that ~71% of BMECs (defined as VEcadherin+CD31+ CD45-Ter119-) were GFP+ (which represents the recombination efficiency) and that BM stromal cells were negative for GFP expression (**New Figure 6**). We further characterized the hematopoietic compartment and demonstrated that phenotypic SLAM HSCs and terminally differentiated

cells were negative for GFP expression (**New Supplemental Figure 4**). Notably, we did not detect GFP expression in any other cell type analyzed other than endothelial cells. Additionally, RT PCR analysis in FACS sorted BMECs and hematopoietic cells confirmed that *creERT2* expression is restricted to endothelial cells of *CDH5-MAPK* mice. Collectively, the new data (**New Figure 6**) confirms fidelity of both *CDH5-MAPK* as well as *Tie2.IkB-SS* mice utilized in this study, further strengthening the conclusions of the manuscript.

Reviewer #2, expert in HSC niche biology (Remarks to the Author):

This is an interesting report from Ramalingam et al. in which the authors show that activation of MAPK signaling in BM ECs induces inflammatory pathway activation in the hematopoietic compartment and suppression of NFkB signaling in endothelial cells restores niche integrity and hematopoietic function. The authors further identify stem cell growth factor alpha as a soluble factor that suppresses BM inflammation and restores hematopoietic function. The manuscript includes some interesting findings that would be of interest to the field, but I have some concerns regarding the experimental approach and conclusions that, if addressed, can improve the manuscript:

Response: We would like to thank the reviewer for reviewing our manuscript and for their insightful comments. We have now included new data that clarifies our experimental approach and addresses the concerns raised by the reviewer. We have also performed a thorough and comprehensive characterization of our mouse models to confirm that transgene expression in our model systems are expressed exclusively in endothelial cells. We have also included new and exciting data providing more mechanistic insights that strengthens the overall conclusions of the manuscript.

1. The authors utilized tamoxifen – impregnated feed to induce MAPK activation in endothelial cells and then describe the effects of this intervention on bone marrow cellularity, etc. Did the authors verify the increase in MAPK signaling in VEcadherin+ ECs? This should be shown.

Response: We have confirmed the increase in MAPK signaling in VE cadherin+ BMECs of CDH5-MAPK mice after Tamoxifen administration (**New Figure 6**).

2a. Figure 1. In the competitive repopulation assay, the cell doses in the donor and competitor grafts should be stated in the main text or figure legend. Was this a 1:1 dilution?

Response: Yes. This was a 1:1 ratio competitive transplant wherein 5×10^5 WBM cells (CD45.2) were transplanted with 5×10^5 competitor WBM cells (CD45.1) via retro-orbital sinus injections into pre-conditioned CD45.1 recipient mice (New Figures 1F-G, 4D-E and 8D-E). We have included these details in the corresponding figure legends.

2b. Were secondary transplants performed to assess long-term HSC defects?

Response: We have included the secondary transplant data which demonstrates that *CDH5-MAPK* mice manifest long-term HSC defects that are resolved upon endothelial NF- κ B inhibition (New **Supplementary Figure 2c**). We have also confirmed that SCGF infusion is able to rescue long-term HSC defects in *CDH5-MAPK* mice both during steady state (New **Figure 8f-g**), as well as following myelosuppressive injury (New **Figure 9i-j**).

2c. In Figure 1e, the data appear to show the contribution of donor cells within each lineage population, as opposed to the absolute percentages of donor+myeloid cells, donor+ T cells, donor+ B cells, etc. in recipient mice. Is the myeloid biased engraftment also evident when the absolute percentages of donor lineage+ cells are compared?

Response: We would like to clarify that the data in original Figure 1E (New **Figure 1G**) represents % of each of the lineage+ cell within the long-term engrafted donor CD45.2 cells. The gating strategy is based on Weissman's group study describing myeloid-biased engraftment of HSCs from old mice (Rossi et al., PMID 15967997, Gating strategy described in Supplementary Figures). To convey this information in an unambiguous manner, we have included representative contour-plots to demonstrate the gating strategy (New **Supplemental Figure 2B**).

2d. Also, if the *CDH5-MAPK* mouse produces myeloid – biased engraftment after transplantation, why don't the CFCs from the BM of *CDH5-MAPK* mice (Figure 1c) show increased myeloid progenitor cells compared to littermate controls?

Response: The long-term myeloid-biased engraftment is not due to increased numbers of myeloid progenitor cells in the donor WBM. Myeloid-biased engraftment after transplantation (≥ 16 weeks) is a readout for the differentiation potential of long-term engrafted HSCs, since most of the transplanted progenitors are exhausted by 16 weeks. Donor HSCs within the *CDH5-MAPK* WBM are exposed to inflammatory signals in their native environment which manifests as myeloid-biased engraftment after transplantation. The decreased frequency of HSCs within donor WBM of *CDH5-MAPK* mice is reflected as decreased long-term CD45.2 engraftment post-transplant.

The CFC assay is a measure of progenitor activity and is used to assay for number of functional progenitors (i.e. CMP, GMP, MEP etc) with a fixed number of WBM cells. CFC assays do not reflect HSC activity or their differentiation potential (Reviewed in Purton LE et al., PMID: 18371361). In Supplementary Figure 2A-B of our original manuscript (New Figure 5A), we demonstrated that *CDH5-MAPK* mice show a significant *decrease* of all the hematopoietic progenitors including CMP, GMP and MEP which correlates perfectly in the CFC assay as decreased numbers of BFU-E, CFU-G, CFU-GEMM etc. In summary, the decrease in *phenotypic* progenitor frequency observed by flow cytometry is supported by a *functional* decrease in colony forming ability by the CFC assay. We have included these clarifications in our revised manuscript.

2e. Figure 1g. A p value of 0.0479 would appear to approximate $p = 0.05$ or non-significance. Does this mean that the difference in the limiting dilution analysis is relatively weak or requires more replicates to confirm a statistically significant difference?

Response: Although it has become a common practice (including the current manuscript) to denote statistical significance as $P < 0.05$, $P < 0.01$ and $P < 0.001$ as “*”, “**” and “***”, it has unfortunately given rise to a common misconception that smaller P-values indicate a higher likelihood that the observed result is true or ‘strong’. These have been nicely reviewed in Cohen HW ‘*P values: use and misuse in medical literature*’ (PMID: 20966898) and Goodman S ‘*A dirty dozen: twelve p-value misconceptions*’ (PMID 18582619). P-values and the associated statistical significance do not give any information on the magnitude of the biological effect being studied (‘effect size’) or its biological significance. We decide the number of animals needed for each experiment *a priori* by performing a power calculation analysis utilizing data generated from preliminary experiments performed in our lab or available in the scientific literature. We routinely observe an effect size of ≥ 2 for most of the continuous variables that we measure in our experiments. Using this data, we performed sample size estimation using GPOWER software (Faul F et al., PMID 17695343) for a two-tailed t-test (or ANOVA for multiple comparisons) with an effect size of 2 (α -error probability = 0.05, power = 0.8, allocation ratio = 1) which revealed that we need $\sim n=5$ animals per experimental group to achieve our desired statistical power. Our estimates for the minimum number of mice needed for limiting dilution analysis ($\sim n=8$ mice/cohort/cell dose) is in agreement with established guidelines (Reviewed in Purton LE et al., PMID: 18371361). These numbers and our lab-specific experiment guidelines give us confidence in our statistical analysis.

Having said that, the P value in original Figure 1G ($P=0.0479$) is statistically significant based on the significance level ($\alpha=0.05$) that was decided before the experiment. Although the limiting dilution analysis is statistically significant, we strongly believe that the overall conclusions must not rely solely on P-values of any single assay. Equal importance must be given to the effect size observed, its biological significance, and reproducibility of the results and whether the findings from multiple assays support the overall conclusions. We would like to highlight the consistency of HSC defects observed in multiple analyses including a decrease in phenotypic HSC frequency along with a loss of HSC quiescence and increased apoptosis (New Figure 1). These phenotypic defects correlate nicely with the loss of long-term engraftment potential observed in competitive repopulation assays (New Figure 1F) as well as decrease in HSC numbers by limiting dilution analysis (New Figure 1H). These HSC defects are also evident in the limiting dilution analysis (New Figure 4F-G) as well as competitive transplantation assays (New Figure 4D-E). Along with engraftment defects, we consistently observe decreased colony counts in CFC assays, drop in peripheral blood counts and lineage-skewing in the peripheral blood of *CDH5-MAPK* mice. Taken together, these findings collectively reflect HSC defects in *CDH5-MAPK* mice that are restored upon endothelial NF- κ B inhibition. We hope this clarification further strengthens the overall conclusions of our manuscript.

2f. Usually, limiting dilution assays to estimate HSC frequency are evaluated using Poisson Statistical Analysis. Was this approach utilized here?

Response: We utilize Extreme Limiting Dilution Analysis (ELDA) for estimation of HSC frequency, which is currently the gold-standard method for assessing stem cell frequency by limiting dilution (Hu Y & Smyth GK., PMID 19567251). The merits of ELDA over Poisson statistics are discussed in the above referenced paper. A few of the recent articles from established investigators in the field of HSC biology utilizing ELDA are as follows: PMIDs 22281595, 26416744, 22385655, 25594182, 25395663, 27111842 and 25326802).

2g. Figure 1k. The immunofluorescent images of the BM vasculature suggest an impressive difference in response to enforced MAPK signaling in ECs. What is the time point of analysis of the mice following MAPK induction in ECs? Is this change in BM vascular appearance persistent over time or does it resolve or worsen over time?

Response: After the Tamoxifen regimen, mice are maintained on regular diet for 4 weeks prior to analysis. The immunofluorescence images represent mice that were started on Tamoxifen at 8 weeks of age, taken off tamoxifen at 12 weeks of age and their femurs harvested for immunofluorescence analysis at 16 weeks of age. We have included a new figure (**New Figure 6**) describing the Tamoxifen regimen and experimental analysis. The vascular appearance and hematopoietic defects are persistent for at least 3 months after Tamoxifen administration and does not worsen or improve over time. We are currently in the process of aging these mice for investigating the effects of chronic endothelial MAPK activation on hematopoietic aging.

3. Figure 2. In Figure 2d, the authors describe crossing CDH5-MAPK mice with Tie2.IkB-SS mice to generate CDH5-MAPK::IkB mice to evaluate whether suppression of endothelial NF-kB signaling downstream of endothelial MAPK activation could temporize BM inflammation observed in CDH5-MAPK mice. Tie2 is expressed in the majority of hematopoietic cell lineages (Genesis 2010;48:563-7) so the Tie2.IkB-SS mice would suppress NF-kB signaling in Tie2+ hematopoietic cells and endothelial cells. How does this cross allow for specific conclusions to be drawn about suppression of endothelial-specific NF-kB signaling suppression?

Response: The *Tie2.IkB-SS* mouse model is a transgenic model wherein a Tie2 promoter/enhancer element drives the expression of the IkBa Super Suppressor transgene (**New Figure 6**). Although Tie2 is expressed in hematopoietic cells during embryonic development (observed in Tie2-cre mouse models), its expression is restricted to the endothelium in adult mice as demonstrated by Tie2-cre/ERT2 models (Reviewed in Pyane S et al., PMID 30354251). In the *Genesis* article cited by the reviewer (Genesis 2010;48:563-7), utilizing *Tie2-Cre;Rosa26REYFP*, Tang et al nicely demonstrated that Tie2 promoter is no longer active in hematopoietic cells beyond E12.5 (Tang Y et al., PMID 20645309). The endothelial-restricted expression of Tie2 driven transgenes in adult C57BL6 mice has also been confirmed by several independent groups (Forde A et al., PMID 12203917, Gareus R et al., PMID 19046569, Korhonen H et al., PMID 19171764). With regard to Tie2 expression in HSCs of adult mice, in our Nature Communications manuscript (Poulos et al., PMID: 28000664) we confirmed that Tie2 was not expressed on HSCs of adult C57BL6 mice. More importantly, we demonstrated that HSCs from *Tie2::IkB.SS* mice, unlike their BMEC counterparts, responded to TNF α stimulation confirming the absence of transgene expression within HSCs of adult *Tie2.IkB-SS* mice. We also confirmed these findings by performing reciprocal transplantations showing that enhanced HSC activity of *Tie2.IkB-SS* mice was exclusively mediated by inhibition of endothelial NF-kB signaling. Collectively, these observations and experiments clearly demonstrate that the *IkB-SS* transgene in *Tie2::IkB-SS* mice is not expressed in adult HSCs or hematopoietic cells and does not interfere with the activation of canonical NF-kB signaling in HSCs.

Additionally, to address the reviewer's concerns and further confirm that the decreased expression of NF-kB target genes within hematopoietic cells of *CDH5-MAPK* mice upon crossing to *Tie2.IkB-SS* mice (New Supp. Fig. 3a) is not due to leaky expression of the IkB-SS transgene within hematopoietic cells, we performed RT PCR analysis for detecting the expression of IkB-SS transgene from FACS sorted BMECs and CD45+ hematopoietic cells. Our analysis confirmed that the IkB-SS transgene was only expressed in BMECs and not in CD45+ cells, further strengthening the conclusions of our findings (**New Figure 6**). Furthermore, we have also performed a thorough characterization of the *Cdh5(PAC)-creERT2* mouse model and have confirmed that *creERT2* expression is restricted to endothelial cells and does not occur in hematopoietic cells including HSCs.

New Figure 6

e

f

g

New Figure 6. *CDH5-MAPK* and *Tie2.IkB-SS* mice demonstrate endothelial-specific expression of transgenes. e) Endothelial cells within the BM of *CDH5-MAPK* mice (defined as CD45- Ter119- CD31+ VECadherin+) demonstrate *cre*-mediated recombination whereas stromal cells (defined as CD45- Ter119- CD31- VECadherin-) do not (n=4 mice per cohort). f) Schematic describing the *Tie2.IkB-SS* mouse model. g) Agarose gel electrophoresis image of RT-PCR amplicons for the indicated genes using RNA isolated from FACS sorted endothelial cells and CD45+ hematopoietic cells in the indicated genotypes (n=3 mice per cohort). Note that IkB-SS transgene is expressed in endothelial cells and shows no detectable expression in hematopoietic cells. Also note the expression of *cre* transgene in endothelial cells of *CDH5-MAPK* mice and no detectable expression in hematopoietic cells. NTC denotes 'No Template Control'. Sort purity was confirmed using expression of *Cdh5* (for endothelial cells) and *Ptpcr* (for hematopoietic cells).

4a. Figure 3. As stated above, the authors should explain how the CDH5-MAPK mice show decreased myeloid colony formation, whereas CDH5-MAPK::I κ B mice show restoration of myeloid colony formation (Figure 3c); following competitive transplantation, the opposite effect is observed, with increased myeloid repopulation in the recipients of CDH5-MAPK donor cells and restoration of normal lineage distribution in recipients of CDH5-MAPK::I κ B mice (Figure 3e). These results appear to be at odds with each other.

Response: Please refer to the Response provided for the Reviewer's comment **2d** regarding colony assays.

4b. Figure 3i. The differences in hematologic parameters shown in this panel are interesting, but it would be helpful to know whether neutrophil counts are also significantly different in the CDH5-MAPK mice versus controls and corrected with CDH5-MAPK::I κ B mice. This would be much more relevant than RBC counts which are largely irrelevant clinically.

Response: We have now included the Neutrophil recovery data (**New Figure 4J**). Please note that although *CDH5-MAPK* mice do not display a statistically significant decrease in neutrophil recovery, they generally tend to have slower neutrophil recovery as compared to control mice. Importantly, both *Tie2.I κ B-SS* and *CDH5-MAPK::I κ B* mice display faster neutrophil recovery as compared to control and *CDH5-MAPK* mice.

4c. My larger concern, as stated above, relates to the Tie2 I κ B mouse model and whether it can allow discrimination of effects on endothelial cells versus hematopoietic cell effects. Would it be possible to validate the findings from the CDH5-MAPK::I κ B mice using an endothelial – specific model in which NF κ B signaling is suppressed?

Response: Please refer to the Response provided for the Reviewer's comment regarding **Point 3**, Figure 2D. Based on our comprehensive characterization of the *Tie2.I κ B-SS* mice which shows that the transgene expression is restricted to endothelial cells, we believe that our conclusions are valid.

5a. Figure 4. Could the absence of differences in ROS levels in BM ECs be a function of the possibility that Tie2 – driven suppression of NF κ B signaling is primarily occurring in hematopoietic cells, rather than in the vascular compartment in these mice?

Response: Same response as above. The lack of expression of the I κ B-SS transgene within the hematopoietic cells of *CDH5-MAPK::I κ B* mice confirms that the vascular and hematopoietic recovery is mediated by endothelial NF- κ B suppression. However, the Reviewer raises an interesting point. It is likely that HSPCs are more susceptible to inflammation-induced ROS production, as compared to niche cells. In support of this idea, we consistently observe higher ROS levels in KLS HSPCs of control mice as compared to their niche cells. We also observe a similar trend in Hypoxyprobe binding wherein KLS cells demonstrate a higher Hypoxyprobe binding as compared to niche cells, which potentially explains their higher ROS levels. Alternatively, recent studies indicate that niche cells in the microenvironment might uptake ROS from HSPCs to protect them stress, suggesting that niche cells might have alternate a robust ROS detoxification system as compared to HSPCs (Reviewed in Ludin A et al., PMID: 24762207). We believe that our *CDH5-MAPK* mouse model will offer a valuable tool to the scientific community to explore the role of niche cells in ROS detoxification.

5b. In Figure 4, the authors transition into analysis of BM stromal cells for evidence of alteration in ROS levels, cell cycling and apoptosis. It is unclear or not well justified or explained why the authors chose to focus additionally on BM stromal cells as well as endothelial cells and HSPCs. Do the authors postulate that suppression of NF κ B signaling in ECs alters cross-talk between ECs and stromal cells such that BM

stromal cell function is altered? Do the authors have any insight mechanistically as to how this communication between ECs and stromal cells is occurring?

Response: Cross-talk between the endothelium and stromal cells have already been well-established (Reviewed in CCW Hughes, PMID 18391786). In addition to BMECs, BM stromal cells (including Lepr+ cells and Osteoblasts) have been shown to regulate HSC and hematopoietic progenitor function. We have recently demonstrated that ablation of stromal cells leads to vascular defects in the bone marrow, indicating an intimate cross-talk between BMECs and stroma (Ramalingam P et al., PMID 28594660). In the Figure 2 of our original manuscript (Heatmaps for NF-kB target genes), in addition to BMECs, we surprisingly observed an increased expression of inflammatory gene targets in stromal cells which indicated that endothelial inflammation affected stromal cells. To determine whether inflammation affected stromal cells, we analyzed their ROS levels, hypoxia and cell-cycle status. The objective of this analysis was to comprehensively characterize the impact of inflammation not only on HSPCs but also the niche cells. Our analysis revealed that while ROS levels were unchanged in stromal cells of *CDH5-MAPK mice*, they did manifest changes in hypoxia and increased apoptosis. However, these defects were rescued upon endothelial NF-kB inhibition which confirms that stromal defects in *CDH5-MAPK mice* arise as a direct result of inflammation in the endothelial niche. We have included the possible causes of unchanged ROS levels within niche cells in our revised manuscript and these findings are summarized in **New Figure 10** of the manuscript.

5c. In Figure 4n-p, the authors show that expression of Il1b and Csf1 is decreased in CD45+ cells, but only Il1b is decreased in BM ECs and stromal cells. If Il1b and Csf1 are both involved in mediating the effects of MAPK activation in BM ECs on hematopoiesis, how do the authors explain these differences?

Response: We would like to clarify that we did not claim in our original manuscript that IL1b and Csf1 were the only mediators of inflammation induced HSC dysfunction. In fact, the data in the Heatmaps of NF-kB target genes demonstrate that a diverse array of inflammatory genes are upregulated in BMECs of *CDH5-MAPK mice* that are suppressed upon crossing with *Tie2.IkB-SS mice*. We decided to evaluate IL1b and Csf1 because they appeared to show increased expression in all the cellular subsets of the BM and have previously been shown to directly affect HSC function. Although Csf1 expression is not decreased in BMECs of *CDH5-MAPK::IkB mice*, the decreased expression of IL1b in BMECs appears to be sufficient to decrease the inflammation in CD45 + cells, stromal cells and the WBM microenvironment, indicating that IL1b might be one of the key inflammatory cytokines mediating the overall BM inflammation. However, this is only a correlation and it's likely that the outcomes of chronic inflammation involves a balance of multiple pro- and anti- inflammatory molecules, and the discovery of SCGF in this study is supportive of this hypothesis. We have clarified these statements in the revised manuscript.

6a. Figure 5. The authors show that delivery of SCGF reverses the hematopoietic and BM vascular defects in *CDH5-MAPK mice* by suppression of BM inflammation driven by MAPK activation in BM ECs. Are the differences in gene expression of target genes shown in Figure 5i significant? Were these analyses repeated more than once to validate the original results? Significant differences should be indicated clearly in the Figure.

Response: All experimental findings were confirmed in at least two independent experiments to ensure reproducibility. The qPCR array analysis was performed to determine whether there was an overall decrease in expression of NF-kB target genes within the WBM of *CDH5-MAPK mice* after SCGF infusion. We have included the statistical analysis as suggested by the Reviewer (**New Figure 8J**). Although some genes are statistically significant and others are not, the results suggest that chronic inflammation is a dynamic process leading to continuous transcriptional changes. This can be observed in

the increased variance of inflammatory gene expression specifically in cellular subsets of *CDH5-MAPK* mice (New Figures 3F, 8J and Supp. Figures 3 and 6F,I). However, we do observe a generalized decrease in NF- κ B target genes in *CDH5-MAPK* mice after SCGF infusion. Importantly, the phenotypic decrease in BM inflammation observed by qPCR after SCGF infusion is functionally supported by restored vascular integrity and rejuvenation of HSC function.

6b. It is unclear mechanistically exactly how SCGF suppresses BM inflammation driven by BM EC MAPK activation.

Response: We have now included new and exciting data which confirms that expression of I κ B α -super suppressor in BMECs derived from *CDH5-MAPK* mice is able to block nuclear p65 translocation (New **Figure 3**). More importantly, we demonstrate that treatment of *CDH5-MAPK* BMECs with SCGF significantly reduces their nuclear p65 translocation (New **Figure 8**). This new data, coupled with our previous rigorous functional data, uncovers a new role for SCGF in directly regulating NF- κ B dependent endothelial inflammation.

New Figure 3 and New Figure 8

Figure 3

Figure 8

New Figure 3. c, d) Representative immunofluorescence images and quantification demonstrating increased levels of nuclear p65 in BMECs derived from *CDH5-MAPK* mice as compared to controls. Note that expression of I κ B-SS in BMECs isolated from *CDH5-MAPK* mice (*CDH5-MAPK::Ikb*) decreases nuclear p65 levels. Each dot within the bar graph represents nuclear p65 staining intensity per individual cell.

New Figure 8. k, l) Representative immunofluorescence images and quantification demonstrating decreased levels of nuclear p65 in BMECs derived from *CDH5-MAPK* mice treated with SCGF.

6c. How does increased NFkB signaling in ECs downregulate SCGF expression in BM stromal cells since BM stromal cells are the primary source of SCGF in the BM (extended Figure 7)?

Response: As the reviewer correctly indicates, analysis of SCGF expression in total stromal cells, Lepr⁺ cells, and osteoblasts from BM of control, *Tie2-IkB-SS*, *CDH5-MAPK*, and *CDH5-MAPK::IkB* mice, revealed no significant changes in mRNA expression (**New Supp. Fig. 9f**) indicating that decreased plasma SCGF in *CDH5-MAPK* mice is not due to transcriptional alterations. However, given that SCGF is a secreted protein and appears to regulate inflammatory responses, it is likely that SCGF expression might be subject to translational regulation. It has been demonstrated that cytokines mediating inflammatory responses are regulated at the translational level (Mazumder B et al., PMID 20304832). Interestingly, a recent report demonstrated that I11b regulates the secretory response of chondrocytes by regulating translation (McDermott BT et al., PMID 31300557). Given that SCGF is predominantly expressed by BM stromal cells, it is likely that it might be subject to translational regulation by I11b. These possibilities, although exciting, will require a thorough biochemical characterization of translational regulation of SCGF in diverse BM stromal cellular subtypes and are beyond the scope of our manuscript. However, the most likely explanation for decreased plasma SCGF based on our existing data appears to be due to the overall decrease in BM stromal cell numbers (the cells producing plasma SCGF) in *CDH5-MAPK* mice due to their increased apoptosis. (**Figure 7h, k**). We have included these observations in the revised manuscript.

6d. Does SCGF act directly on HSPCs and also directly on BM ECs and/or stromal cells or do indirect effects occur on these populations in response to SCGF infusion?

Response: SCGF clearly acts on stromal cells as it improves bone formation in *CDH5-MAPK* mice (**New Supp. Figure 9a-d**). Given that SCGF knockout mice display bone loss along with normal hematopoietic parameters (Yue R et al., PMID 27976999) and the lack of discernible effects on hematopoiesis in SCGF infused control mice in our study (**New Figures. 8a-g**), it is likely that the hematopoietic recovery observed in SCGF infused *CDH5-MAPK* mice is mediated by its effect on BMECs. Our new data supports this hypothesis wherein treatment of *CDH5-MAPK* BMECs with SCGF decreases their nuclear p65 translocation (**New Figure 8k-l**). We have included these observations in our revised manuscript.

7a. Figure 6. Interesting results are shown to indicate that infusion of SCGF improves hematologic recovery following irradiation. Did SCGF accelerate recovery neutrophils in irradiated mice? This would be the most clinically interesting result, since neutrophil counts dictate need for hospitalization and risk of bacterial infections.

Response: We would like to thank the reviewer for appreciating the clinical relevance of SCGF infusion on hematologic recovery following myelosuppressive injury. We believe this is the most important finding of this study. We have included new data demonstrating that SCGF infusion enhances neutrophil recovery not only in *CDH5-MAPK* mice, but control mice as well (**New Figure 9 A, B**).

7b. If SCGF infusions do not increase numbers of phenotypic HSPCs in wild type mice following irradiation (6d), how do the authors explain the significant enhancement in recovery of cells capable of competitive repopulation as shown in Figure 6e?

Response: Although phenotypic HSC frequency estimated by Flow cytometry is certainly informative, it does not provide any information regarding the functional potential of the HSC. For instance, the seminal paper by Rossi et al., (PMID 15967997) demonstrated that there is ~4-fold increase of *phenotypic* HSC frequency in aged mice as compared to young mice. However, transplantation of aged HSCs results in decreased engraftment, as compared to young HSCs indicating a *decline of functional potential* of aged HSCs. This has also been shown to be relevant during inflammation-induced myelosuppression wherein

HSC numbers are not significantly altered phenotypically but display defects in their functional potential (Zhang H et al., PMID 27264973). Hence, the only way to assess HSC function is by evaluating hematopoietic parameters including blood counts as well long-term engraftment potential of the HSC. Our data indicates that SCGF preserves HSC functionality as indicated by data in New Figure 9. We have included these observations in the manuscript.

Minor Concern

Page 14, bottom paragraph: Several other publications have characterized the effects of myelosuppressive injury on the BM vasculature and BM vascular function beyond Hooper et al., and these references should be appropriately cited.

Response: Hooper et al., was the first study to genetically demonstrate that hematopoietic recovery following myelosuppression is contingent upon vascular recovery. However, we have included the appropriate citation describing effect of myelosuppressive injury on the vasculature. We would welcome the input of the reviewer in case we have missed any key references.

Reviewer #3, expert in HSC repopulation (Remarks to the Author):

Endothelial MAPK activation disrupts hematopoiesis by including NFkappaB dependent inflammatory stress by Ramalingam et al. The role of chronic inflammatory signals in the bone marrow niche for hematopoiesis is still controversially discussed. The authors demonstrate that constitutive expression of MAPK in endothelial cells perturbs hematopoiesis. This correlates with an overall inflammatory signature in stroma as well as hematopoietic cells (NFkappaB) and inflammatory signatures. Genetic inhibition of NFkappaB signaling restores normal function in the model system. Mechanistically, that also restores hypoxia in the bone marrow to normal levels of hypoxia, but not ROS, as ROS is not induced by elevated MAPK in endothelial cells. The authors further identify SCGF as a target cytokine that mediates the negative effect of MAPK activation as well as myelosuppressive treatment in endothelial cells on hematopoiesis. This is an overall interesting study with a number of very novel findings including mechanisms that will contribute to the field.

Response: We would like to thank the Reviewer for reviewing our manuscript and providing insightful comments. We would also like to thank the Reviewer for acknowledging the mechanistic insights that our study provides to the field of inflammation and stress hematopoiesis. We have now included new data that clarifies our experimental approach and carefully addresses the concerns raised by the reviewer. We have also performed a comprehensive characterization of our mouse models to confirm that transgene expression in our model systems are expressed exclusively in endothelial cells. We have also included new and exciting data providing more mechanistic insights that strengthens the overall conclusions of the manuscript.

There are though a couple of concerns that should/need to be addressed:

Major Concerns:

1. The authors need to confirm the level of activation of MAPK in endothelial cells in their animal model system. That information is not provided. All cells? Level of activation etc.?

Response: We have included new data that demonstrates the level of MAPK activation in BMECs of *CDH5-MAPK* mice (**New Figure 6**). Using a GFP reporter system, we confirm that MAPK activation occurs exclusively within the endothelium with no off-target expression in any other cell types including hematopoietic cells and HSCs.

2a. The Tie2 promoter can be leaky and also be expressed in HSCs/hematopoietic cells.

The authors need to test the level of expression of the transgene in hematopoietic cells and especially HSCs. It looks like that has not been tested yet, also not in the previous publications with that mouse. That needs to be provided, as worst case, the data would require a strong re-interpretation of the results.

Response: We would like to clarify the mouse model used in our study. The *Tie2.IkB-SS* mouse model is a transgenic model wherein a Tie2 promoter/enhancer element drives the expression of the IkBa Super Suppressor transgene (**New Figure 6**). Although Tie2 is expressed in hematopoietic cells during embryonic development (observed in *Tie2-cre* mouse models), its expression is restricted to the endothelium in adult mice as demonstrated by *Tie2-cre/ERT2* models (Reviewed in Pyane S et al., PMID 30354251). Utilizing *Tie2-Cre;Rosa26REYFP*, Tang et al demonstrated that Tie2 promoter is no longer active in hematopoietic cells beyond E12.5 (PMID 20645309). The endothelial-restricted expression of Tie2 driven transgenes in adult C57BL6 mice has also been confirmed by several independent groups (Forde A et al., PMID 12203917, Gareus R et al., PMID 19046569 and Korhonen H et al., PMID 19171764). However, as per the Reviewer's suggestion, to further confirm that the decreased expression of NF-kB target genes within hematopoietic cells of *CDH5-MAPK* mice upon crossing to *Tie2.IkB-SS*

mice (**New Supplemental Fig. 3a**) is not due to leaky expression of the I κ B-SS transgene within hematopoietic cells, we performed RT PCR analysis for detecting the expression of I κ B-SS transgene from FACS sorted BMECs and CD45⁺ hematopoietic cells. Our analysis confirmed that the I κ B-SS transgene was only expressed in BMECs and not in CD45⁺ cells, further strengthening the conclusions of our findings (**New Figure 6**).

New Figure 6

New Figure 6. **c)** Analysis of phospho-ERK1/2 expression by flow cytometry confirms *in vivo* activation of MAPK pathway in BMECs of *CDH5-MAPK* mice following Tamoxifen administration (n=3 mice per cohort). **d)** GFP⁺ BMECs as compared to GFP⁻ BMECs within *CDH5-MAPK* mice demonstrate increased phospho-ERK1/2 expression by flow cytometry, confirming the fidelity of GFP reporter for tracking *cre*-mediated recombination *in vivo*. (n=3 mice per cohort). **e)** Endothelial cells within the BM of *CDH5-MAPK* mice (defined as CD45⁻ Ter119⁻ CD31⁺ VECadherin⁺) demonstrate *cre*-mediated recombination whereas stromal cells (defined as CD45⁻ Ter119⁻ CD31⁻ VECadherin⁻) do not (n=4 mice per cohort). **f)** Schematic describing the *Tie2.I κ B-SS* mouse model. **g)** Agarose gel electrophoresis image of RT-PCR amplicons for the indicated genes using RNA isolated from FACS sorted endothelial cells and CD45⁺ hematopoietic cells in the indicated genotypes (n=3 mice per cohort). Note that I κ B-SS transgene is expressed in endothelial cells and shows no detectable expression in hematopoietic cells. Also note the expression of *cre* transgene in endothelial cells of *CDH5-MAPK* mice and no detectable expression in hematopoietic cells. NTC denotes 'No Template Control'. Sort purity was confirmed using expression of *Cdh5* (for endothelial cells) and *Ptprc* (for hematopoietic cells).

2b. The Tie2 promoter can be leaky and also be expressed in HSCs/hematopoietic cells. The authors need to test the level of expression of the transgene in hematopoietic cells and especially HSCs. *It looks like that has not been tested yet, also not in the previous publications with that mouse.*

Response: In our Nature Communications manuscript (Poulos et al., PMID: 28000664) we confirmed that Tie2 was not expressed on HSCs of adult C57BL6 mice. More importantly, we demonstrated that HSCs from *Tie2::IkB.SS* mice, unlike their BMEC counterparts, responded to TNF α stimulation confirming the absence of transgene expression within HSCs of adult *Tie2::IkB-SS* mice. We also confirmed these findings by performing reciprocal transplantations showing that enhanced HSC activity of *Tie2.IkB-SS* mice was exclusively mediated by inhibition of endothelial NF-kB signaling. Collectively, these observations and experiments clearly demonstrate that the *IkB-SS* transgene in *Tie2::IkB-SS* mice is not expressed in adult HSCs or hematopoietic cells and does not interfere with the activation of canonical NF-kB signaling in HSCs.

3. The link between activation of MAPK in endothelial cells and the overall strong inflammatory signature in the whole bone marrow, including hematopoietic cells, is not determined to a great extent. There should be more emphasis on that point, especially as 2d, e and g show very similar overall activation patterns.

Response: Our new data confirming that MAPK activation and NF-kB inhibition occurs exclusively within the endothelial cells demonstrates that endothelial inflammation can have profound effects on hematopoietic cells within the bone marrow, as reflected by the Heatmaps of inflammatory signature. We have included these observations in the revised manuscript to emphasize this point, as per the Reviewer's suggestion.

Minor Concerns:

1. Hypoxia in bone marrow is notoriously difficult to determine. The authors need to provide gating information etc. on these analyses, especially as the data are central for the conclusions.

Response: We have included the gating information for hypoxia in our revised manuscript (New Supplementary Figure 5A). We have also included gating information for all the cellular sub-types analyzed as well as functional assays including ROS detection, cell-cycle analysis and apoptosis, in appropriate sections of the manuscript. Notably, we have previously utilized Hypoxyprobe to confirm that aging results in increased hypoxia within the BM vascular niche (Poulos et al., PMID 29035282).

2. The hematopoietic phenotype is complex, and due to the overall broad changes in the bone marrow, multiple cells type might be the target of the changes and contribute to the observed changes. For example, a 5 day treatment that results in such a large overall changes in hematopoiesis needs to target a broad range of distinct type of hematopoietic cells. That should be addressed/discussed in more detail.

Response: We agree with the Reviewer. SCGF clearly acts on stromal cells as it improves bone formation in *CDH5-MAPK* mice (**New Supp. Figure 9A-D**). However, given that SCGF knockout mice display bone loss along with normal hematopoietic parameters (Yue R et al., PMID 27976999) and the lack of discernible effects on hematopoiesis in SCGF infused control mice in our study (**New Figures. 8A-G**), it is likely that the hematopoietic recovery observed in SCGF infused *CDH5-MAPK* mice is mediated by its effect on BMECs and the vasculature. Our new data supports this hypothesis wherein treatment of *CDH5-MAPK* BMECs with SCGF decreases their nuclear p65 translocation (**New Figure 8K-L**). However, these observations do not rule out the possibility that SCGF could also have a direct effect on hematopoietic cells. We have included these observations in our revised manuscript.

3. Page11: The data does not show that the defects in hematopoiesis are a direct effect of inflammation-induced alterations in ROS and hypoxia. That is a correlation.

Response: It is indeed a correlation and we have changed the text in our revised manuscript accordingly.

REVIEWERS' COMMENTS:

Reviewer #1 (Remarks to the Author):

With a great enthusiasm I read the revised manuscript and the response to reviewers' critiques of Ramalingam et al. While, undoubtedly, the authors made a serious attempt to revise the manuscript according to the reviewers suggestions, I am disappointed to see that the major concerns of reviewer # 1 have not been appropriately addressed with new data, as requested (see below).

1. It still remains unclear if the justifications provided (such as "useful tool for scientific community" and "SCGF as a novel factor that enhances vascular and hematopoietic regeneration") regarding the novelty of their study is convincing enough to publish in Nature communications.

2. Reviewer # 1 requested the authors to provide data on distinct HSPC subsets; LT-HSC, ST-HSC, MPP2, MPP3 & MPP4, as these would be critical in understanding the physiological consequences of MAPK signaling defects in HSCs. However, the authors provided data following a HSC immunophenotyping strategy that is not accepted or followed by the field. Unfortunately, the immunophenotyping scheme (HPC-1, HPC-2 & MPP) presented by the authors is not consistent and contradicts with the HSC immunophenotyping strategies established by the pioneering labs, including Passegue lab (Pietras, Cell Stem Cell, 2015; Yamashita, Cell Stem Cell, 2019) and Trumpp Lab (Wilson, Cell, 2008). For example, the authors show data on MPP without including Flt3 in their panels. This is unacceptable, as the "classical" MPPs established by the Weissman Lab are CD34+Flt3+LSK. Moreover, the authors did not provide data on relative numbers of any of these subsets. Finally, the rigor of the new data on HSPC is questionable, as they have shown data based on only n=5 mice.

3. The rationale provided by the authors regarding Hypoxia is not at all convincing. Simply relying on the observations from some other Laboratory (of Tsvee Lapidot) is unacceptable, because, as we all know each experimental setting is different and data obtained from different mouse models can be completely different. The authors should have put some efforts in describing the downstream consequences of ROS in their model.

4. Reviewer # 1 requested the authors to provide data on myeloid progenitors with a more refined immunophenotyping strategy (based on the strong published reports). Instead of showing these data with new experiments, the authors justify their immunophenotyping scheme by citing others (who still follow a controversial immunophenotyping scheme for defining myeloid progenitors), even though there are enough evidences to prove that this approach is incorrect. I do not believe that they provided a scientific justification for not showing the requested data.

Reviewer #2 (Remarks to the Author):

I commend the authors for the additional detailed analyses performed to address my concerns. Their revised manuscript has satisfactorily addressed the vast majority of my concerns.

Reviewer #3 (Remarks to the Author):

Endothelial MAPK activation disrupts hematopoiesis by including NFkappaB dependent inflammatory stress by Ramalingam et al.

The authors addressed the concerns of this reviewer and almost all concerns of the other two reviewers sufficiently and convincingly, also with a large amount of novel data.

NCOMMS-19-11331A-Z:

Ramalingam et al., “Endothelial MAPK activation disrupts hematopoiesis by inducing NF-kB-dependent inflammatory stress”

Reviewer #2 (Remarks to the Author)

I commend the authors for the additional detailed analyses performed to address my concerns. Their revised manuscript has satisfactorily addressed the vast majority of my concerns.

Reviewer #3 (Remarks to the Author)

Endothelial MAPK activation disrupts hematopoiesis by inducing NFkappaB dependent inflammatory stress by Ramalingam et al. The authors addressed the concerns of this reviewer and almost all concerns of the other two reviewers sufficiently and convincingly, also with a large amount of novel data.

Response: We thank Reviewers 2 and 3 for their time and efforts in reviewing our manuscript, their positive feedback and their acknowledgement of the inclusion of the large amount of novel data. We are pleased to know that we have satisfactorily addressed the majority of their comments. We also thank all the Reviewers for their comments and suggestions, which we believe have significantly strengthened the overall conclusions of the manuscript and led to the identification of novel mechanistic insights into the role of SCGF in suppressing inflammation.

Reviewer #1 (Remarks to the Author)

With a great enthusiasm I read the revised manuscript and the response to reviewers' critiques of Ramalingam et al. While, undoubtedly, the authors made a serious attempt to revise the manuscript according to the reviewer's suggestions, I am disappointed to see that the major concerns of reviewer # 1 have not been appropriately addressed with new data, as requested (see below).

Response: We would like to thank Reviewer 1 for their time and effort in reviewing our manuscript, and for their acknowledgement of our efforts in addressing their concerns. We would like to address the reviewer's concerns with further clarifications.

1. It still remains unclear if the justifications provided (such as “useful tool for scientific community” and “SCGF as a novel factor that enhances vascular and hematopoietic regeneration”) regarding the novelty of their study is convincing enough to publish in Nature communications.

Response: We believe that our discovery of the genetic interaction between ERK and NF-kB signaling pathways within the vascular endothelium in mediating chronic BM inflammation is novel and impactful. Furthermore, the identification of novel functions of SCGF as a cytokine that suppresses bone marrow inflammation and enhances neutrophil recovery following myelosuppressive injury has the potential for tremendous clinical implications.

2a. Reviewer # 1 requested the authors to provide data on distinct HSPC subsets; LT-HSC, ST-HSC, MPP2, MPP3 & MPP4, as these would be critical in understanding the physiological consequences of MAPK signaling defects in HSCs. However, the authors provided data following a HSC immunophenotyping strategy that is not accepted or followed by the field. Unfortunately, the immunophenotyping scheme (HPC-1, HPC-2 & MPP) presented by the authors is not consistent and contradicts with the HSC immunophenotyping strategies established by the pioneering labs, including Passegue lab (Pietras, Cell Stem Cell, 2015; Yamashita, Cell Stem Cell, 2019) and Trumpp Lab (Wilson,

Cell, 2008). For example, the authors show data on MPP without including Flt3 in their panels. This is unacceptable, as the “classical” MPPs established by the Weissman Lab are CD34+Flt3+LSK.

Response: While we agree with the Reviewer that the refinements to MPP phenotyping strategies developed by many labs, including the Passegue and Trumpp labs, have led to a better understanding of the downstream effector cells of the LT-HSC, we respectfully disagree with the Reviewer that the immunophenotyping utilized in this manuscript (based on Sean Morrison’s 2013 Cell Stem Cell manuscript, PMID: 23827712) is incorrect or not followed by the field. Morrison’s manuscript demonstrated with robust transplantation assays that MPPs (defined as KLS CD150-CD48- and utilized in our manuscript) contain most, if not all, of the subsets of MPPs which include MPP-1, MPP-2, and MPP-3. It is one of the most commonly utilized immunophenotyping strategies in the field to characterize HSPC subsets. We would like to highlight a short list of recent manuscripts that follow this immunophenotyping strategy exclusively within *Nature* group of Journals (Nahrendorf and Scadden’s lab **Nat Med.** 2019 Nov;25(11):1761-1771 PMID 31700184, Jaffredo’s lab **Nat Cell Biol.** 2019 Nov;21(11):1334-1345 PMID: 31685991, Aifantis lab **Nat Immunol.** 2019 Sep;20(9):1196-1207 PMID 31406379, Frenette’s lab **Nat Commun.** 2018 Jun 22;9(1):2449 PMID 29934585, Morrison’s lab **Nature.** 2017 Sep 28;549(7673):476-481 PMID 28825709).

While we acknowledge the Reviewer’s point that ‘refined’ MPP phenotyping certainly adds information in experimental settings wherein there might be subtle changes in absolute or relative numbers of MPP sub-populations indicative of HSC differentiation defects, our data in Figure 1d and Supplemental Figures 1a, 2a and 8c demonstrates a near complete loss of KLS cells and all of the KLS subsets including MPPs in *CDH5-MAPK* mice. In a recent Perspective published in *Nat Rev Immunol* last week (Rodewald HR et al., *Do haematopoietic stem cells age.* Nat Rev Immunol. 2019 Nov 18. PMID 31740804). To quote verbatim from this Perspective, “*In our studies, the long-term repopulating HSCs (LT-HSCs) were CD48-CD150+Lin-SCA1+Kit+ (LSK) cells (ref.56). The ST-HSCs were CD48-CD150-LSK cells; these cells have been designated as MPP1-MPP3 (ref.59) or MPPs (ref.60) in other studies. In addition, we designated CD48+CD150-LSK cells as MPPs (ref. 56), whereas cells of this phenotype have been referred to as haematopoietic progenitor cells fraction 1 (HPC-1) (ref. 59) or MPP3/4 (ref.60) in other reports*”. This Perspective demonstrates that irrespective of the immunophenotyping strategy employed, MPPs are always a subset of the KLS population. Because of the drastic loss of KLS cells and all of its subsets observed in *CDH5-MAPK* mice, it is quite clear that all HSC and MPP subsets (which will fall within one of the 4 quadrant gates of SLAM, whether they are positive or negative for CD34 or Flt3), demonstrate a drastic decline in their numbers. We fail to understand what novel information can be derived by adding additional markers, and how such level of granularity would change the overall conclusions of the current manuscript. In light of these observations, we feel that the gating strategy we utilized is appropriate because of the population’s large overlap with many of the MPP populations that have been described above. Furthermore, our manuscript did not aim to establish that these downstream effector cells were affected in our novel animal models of endothelial-driven inflammation, but rather our objective was to determine whether chronic endothelial inflammation causes HSCs to lose their functional capacity and that inhibiting endothelial inflammation can restore HSC function. Our hypotheses were addressed using vigorous competitive transplantation and limiting dilution assays, leaving little doubt as to the conclusions of the data presented.

2b. Moreover, the authors did not provide data on relative numbers of any of these subsets.

Response: We had included both relative and absolute numbers of these subsets in our revised manuscript, as well as in the rebuttal. Please refer to Figure 1d and Supplemental Figure 1a.

2c. Finally, the rigor of the new data on HSPC is questionable, as they have shown data based on only n=5 mice.

Response: The bar graph denotes representative data from one independent experiment, with n=5 mice per genotype. Sample sizes were determined by performing power calculation analysis, as described in our rebuttal. We have confirmed HSPC defects in *CDH5-MAPK* mice in 5 independent experiments and we believe that our data is rigorous.

3. The rationale provided by the authors regarding Hypoxia is not at all convincing. Simply relying on the observations from some other Laboratory (of Tsvee Lapidot) is unacceptable, because, as we all know each experimental setting is different and data obtained from different mouse models can be completely different. The authors should have put some efforts in describing the downstream consequences of ROS in their model.

Response: The observations on ROS and hypoxia in this study are descriptive in nature, and confirm previous reports of inflammation-induced ROS and hypoxia. Additionally, increased ROS and hypoxia are hallmark features of a vascular niche with disrupted integrity. Therefore, we have essentially described a potential downstream consequence of ROS in our models. Nonetheless, future studies will undoubtedly reveal additional insights into the mechanisms of ROS in this model, but these experiments are beyond the scope of our current manuscript.

4. Reviewer # 1 requested the authors to provide data on myeloid progenitors with a more refined immunophenotyping strategy (based on the strong published reports). Instead of showing these data with new experiments, the authors justify their immunophenotyping scheme by citing others (who still follow a controversial immunophenotyping scheme for defining myeloid progenitors), even though there are enough evidences to prove that this approach is incorrect. I do not believe that they provided a scientific justification for not showing the requested data.

Response: As the Reviewer correctly suggests, immunophenotyping strategies have become ‘controversial’ after the advent of novel technologies. However, our strategy for quantifying myeloid progenitors cannot be called incorrect and is the most widely utilized strategy, including the very recently published inflammation-related articles in the Nature group (Nahrendorf and Scadden’s lab *Nat Med.* 2019 Nov;25(11):1761-1771 PMID 31700184, Aifantis lab *Nat Immunol.* 2019 Sep;20(9):1196-1207 PMID 31406379). We do however acknowledge that the field, as well as journals, needs to acknowledge and address the availability of multiple immunophenotyping strategies to decide on a common strategy acceptable to all researchers.

As for the scientific justification, and as mentioned in our previous rebuttal, we would like to strongly reiterate the importance of functional assays. In our original manuscript, we demonstrated that the decreased frequency of immunophenotypically defined CMPs, GMPs and MEPs in *CDH5-MAPK* mice are manifested as an overall decrease in functional progenitor activity of erythroid (BFU-E), myeloid (CFU-G, CFU-M) and megakaryocytic (CFU-GEMM) lineages using methylcellulose-based colony forming assays (Figure 1E). Furthermore, suppression of endothelial NF-kB signaling (Figure 4C) or infusion of SCGF (Figure 8C) is able to significantly restore colony-forming ability in *CDH5-MAPK* mice demonstrating a functional restoration of progenitor activity. We believe that these functional data robustly demonstrate the myeloid progenitor defects in *CDH5-MAPK* mice.